## Analysis

# Critical transitions in the Amazon forest system

Bernardo M. Flores[1 ✉], Encarni Montoya[2], Boris Sakschewski[3], Nathália Nascimento[4], Arie Staal[5], Richard A. Betts[6,7], Carolina Levis[1], David M. Lapola[8], Adriane Esquível-Muelbert[9,10], Catarina Jakovac[11], Carlos A. Nobre[4], Rafael S. Oliveira[12], Laura S. Borma[13], Da Nian[3], Niklas Boers[3,14], Susanna B. Hecht[15], Hans ter Steege[16,17], Julia Arieira[18], Isabella L. Lucas[19], Erika Berenguer[20], José A. Marengo[21,22,23], Luciana V. Gatti[13], Caio R. C. Mattos[24] & Marina Hirota[1,12,25 ✉]

The possibility that the Amazon forest system could soon reach a tipping point, inducing large-scale collapse, has raised global concern[1–3]. For 65 million years, Amazonian forests remained relatively resilient to climatic variability. Now, the region is increasingly exposed to unprecedented stress from warming temperatures, extreme droughts, deforestation and fires, even in central and remote parts of the system[1]. Long existing feedbacks between the forest and environmental conditions are being replaced by novel feedbacks that modify ecosystem resilience, increasing the risk of critical transition. Here we analyse existing evidence for five major drivers of water stress on Amazonian forests, as well as potential critical thresholds of those drivers that, if crossed, could trigger local, regional or even biome-wide forest collapse. By combining spatial information on various disturbances, we estimate that by 2050, 10% to 47% of Amazonian forests will be exposed to compounding disturbances that may trigger unexpected ecosystem transitions and potentially exacerbate regional climate change. Using examples of disturbed forests across the Amazon, we identify the three most plausible ecosystem trajectories, involving different feedbacks and environmental conditions. We discuss how the inherent complexity of the Amazon adds uncertainty about future dynamics, but also reveals opportunities for action. Keeping the Amazon forest resilient in the Anthropocene will depend on a combination of local efforts to end deforestation and degradation and to expand restoration, with global efforts to stop greenhouse gas emissions.

The Amazon forest is a complex system of interconnected species, ecosystems and human cultures that contributes to the well-being of people globally[1]. The Amazon forest holds more than 10% of Earth's terrestrial biodiversity, stores an amount of carbon equivalent to 15–20 years of global $CO_2$ emissions (150–200 Pg C), and has a net cooling effect (from evapotranspiration) that helps to stabilize the Earth's climate[1–3]. The forest contributes up to 50% of rainfall in the region and is crucial for moisture supply across South America[4], allowing other biomes and economic activities to thrive in regions that would otherwise be more arid, such as the Pantanal wetlands and the La Plata river basin[1]. Large parts of the Amazon forest, however, are projected to experience mass mortality events due to climatic and land use-related disturbances in the coming decades[5,6], potentially accelerating climate change through carbon emissions and feedbacks with the climate system[2,3]. These impacts would also involve irreversible loss of biodiversity, socioeconomic and cultural values[1,7–9]. The Amazon is home to more than 40 million people, including 2.2 million Indigenous peoples of more than 300 ethnicities, as well as afrodescendent and local traditional communities[1]. Indigenous peoples and local communities (IPLCs) would be harmed by forest loss in terms of their livelihoods, lifeways and knowledge systems that inspire societies globally[1,7,9].

[1]Graduate Program in Ecology, Federal University of Santa Catarina, Florianopolis, Brazil. [2]Geosciences Barcelona, Spanish National Research Council, Barcelona, Spain. [3]Potsdam Institute for Climate Impact Research, Member of the Leibniz Association, Potsdam, Germany. [4]Institute of Advanced Studies, University of São Paulo, São Paulo, Brazil. [5]Copernicus Institute of Sustainable Development, Utrecht University, Utrecht, The Netherlands. [6]Met Office Hadley Centre, Exeter, UK. [7]Global Systems Institute, University of Exeter, Exeter, UK. [8]Center for Meteorological and Climatic Research Applied to Agriculture, University of Campinas, Campinas, Brazil. [9]School of Geography, Earth and Environmental Sciences, University of Birmingham, Birmingham, UK. [10]Birmingham Institute of Forest Research, University of Birmingham, Birmingham, UK. [11]Department of Plant Sciences, Federal University of Santa Catarina, Florianopolis, Brazil. [12]Department of Plant Biology, University of Campinas, Campinas, Brazil. [13]Division of Impacts, Adaptation and Vulnerabilities (DIIAV), National Institute for Space Research, São José dos Campos, Brazil. [14]Earth System Modelling, School of Engineering and Design, Technical University of Munich, Munich, Germany. [15]Luskin School for Public Affairs and Institute of the Environment, University of California, Los Angeles, CA, USA. [16]Naturalis Biodiversity Center, Leiden, The Netherlands. [17]Quantitative Biodiversity Dynamics, Utrecht University, Utrecht, The Netherlands. [18]Science Panel for the Amazon (SPA), São José dos Campos, Brazil. [19]Sustainable Development Solutions Network, New York, NY, USA. [20]Environmental Change Institute, University of Oxford, Oxford, UK. [21]Centro Nacional de Monitoramento e Alerta de Desastres Naturais, São José dos Campos, Brazil. [22]Graduate Program in Natural Disasters, UNESP/CEMADEN, São José dos Campos, Brazil. [23]Graduate School of International Studies, Korea University, Seoul, Korea. [24]Program in Atmospheric and Oceanic Sciences, Princeton University, Princeton, NJ, USA. [25]Group IpES, Department of Physics, Federal University of Santa Catarina, Florianopolis, Brazil. ✉e-mail: mflores.bernardo@gmail.com; marinahirota@gmail.com

Understanding the risk of such catastrophic behaviour requires addressing complex factors that shape ecosystem resilience[10]. A major question is whether a large-scale collapse of the Amazon forest system could actually happen within the twenty-first century, and if this would be associated with a particular tipping point. Here we synthesize evidence from paleorecords, observational data and modelling studies of critical drivers of stress on the system. We assess potential thresholds of those drivers and the main feedbacks that could push the Amazon forest towards a tipping point. From examples of disturbed forests across the Amazon, we analyse the most plausible ecosystem trajectories that may lead to alternative stable states[10]. Moreover, inspired by the framework of 'planetary boundaries'[11], we identify climatic and land use boundaries that reveal a safe operating space for the Amazon forest system in the Anthropocene epoch[12].

## Theory and concepts

Over time, environmental conditions fluctuate and may cause stress on ecosystems (for example, lack of water for plants). When stressing conditions intensify, some ecosystems may change their equilibrium state gradually, whereas others may shift abruptly between alternative stable states[10]. A 'tipping point' is the critical threshold value of an environmental stressing condition at which a small disturbance may cause an abrupt shift in the ecosystem state[2,3,13,14], accelerated by positive feedbacks[15] (see Extended Data Table 1). This type of behaviour in which the system gets into a phase of self-reinforcing (runaway) change is often referred to as 'critical transition'[16]. As ecosystems approach a tipping point, they often lose resilience while still remaining close to equilibrium[17]. Thus, monitoring changes in ecosystem resilience and in key environmental conditions may enable societies to manage and avoid critical transitions. We adopt the concept of 'ecological resilience'[18] (hereafter 'resilience'), which refers to the ability of an ecosystem to persist with similar structure, functioning and interactions, despite disturbances that push it to an alternative stable state. The possibility that alternative stable states (or bistability) may exist in a system has important implications, because the crossing of tipping points may be irreversible for the time scales that matter to societies[10]. Tropical terrestrial ecosystems are a well-known case in which critical transitions between alternative stable states may occur (Extended Data Fig. 1).

## Past dynamics

The Amazon system has been mostly covered by forest throughout the Cenozoic era[19] (for 65 million years). Seven million years ago, the Amazon river began to drain the massive wetlands that covered most of the western Amazon, allowing forests to expand over grasslands in that region. More recently, during the drier and cooler conditions of the Last Glacial Maximum[20] (LGM) (around 21,000 years ago) and of the mid-Holocene epoch[21] (around 6,000 years ago), forests persisted even when humans were already present in the landscape[22]. Nonetheless, savannas expanded in peripheral parts of the southern Amazon basin during the LGM and mid-Holocene[23], as well as in the northeastern Amazon during the early Holocene (around 11,000 years ago), probably influenced by drier climatic conditions and fires ignited by humans[24,25]. Throughout the core of the Amazon forest biome, patches of white-sand savanna also expanded in the past 20,000–7,000 years, driven by sediment deposition along ancient rivers[26], and more recently (around 800 years ago) owing to Indigenous fires[27]. However, during the past 3,000 years, forests have been mostly expanding over savanna in the southern Amazon driven by increasingly wet conditions[28].

Although palaeorecords suggest that a large-scale Amazon forest collapse did not occur within the past 65 million years[19], they indicate that savannas expanded locally, particularly in the more seasonal peripheral regions when fires ignited by humans were frequent[23,24]. Patches of white-sand savanna also expanded within the Amazon forest owing to geomorphological dynamics and fires[26,27]. Past drought periods were usually associated with much lower atmospheric $CO_2$ concentrations, which may have reduced water-use efficiency of trees[29] (that is, trees assimilated less carbon during transpiration). However, these periods also coincided with cooler temperatures[20,21], which probably reduced water demand by trees[30]. Past drier climatic conditions were therefore very different from the current climatic conditions, in which observed warming trends may exacerbate drought impacts on the forest by exposing trees to unprecedented levels of water stress[31,32].

## Global change impacts on forest resilience

Satellite observations from across the Amazon suggest that forest resilience has been decreasing since the early 2000s[33], possibly as a result of global changes. In this section, we synthesize three global change impacts that vary spatially and temporally across the Amazon system, affecting forest resilience and the risk of critical transitions.

### Regional climatic conditions

Within the twenty-first century, global warming may cause long-term changes in Amazonian climatic conditions[2]. Human greenhouse gas emissions continue to intensify global warming, but the warming rate also depends on feedbacks in the climate system that remain uncertain[2,3]. Recent climate models of the 6th phase of the Coupled Model Intercomparison Project (CMIP6) agree that in the coming decades, rainfall conditions will become more seasonal in the eastern and southern Amazonian regions, and temperatures will become higher across the entire Amazon[1,2]. By 2050, models project that a significant increase in the number of consecutive dry days by 10–30 days and in annual maximum temperatures by 2–4 °C, depending on the greenhouse gas emission scenario[2]. These climatic conditions could expose the forest to unprecedented levels of vapour pressure deficit[31] and consequently water stress[30].

Satellite observations of climatic variability[31] confirm model projections[2], showing that since the early 1980s, the Amazonian region has been warming significantly at an average rate of 0.27 °C per decade during the dry season, with the highest rates of up to 0.6 °C per decade in the centre and southeast of the biome (Fig. 1a). Only a few small areas in the west of the biome are significantly cooling by around 0.1 °C per decade (Fig. 1a). Dry season mean temperature is now more than 2 °C higher than it was 40 years ago in large parts of the central and southeastern Amazon. If trends continue, these areas could potentially warm by over 4 °C by 2050. Maximum temperatures during the dry season follow a similar trend, rising across most of the biome (Extended Data Fig. 2), exposing the forest[34] and local peoples[35] to potentially unbearable heat. Rising temperatures will increase thermal stress, potentially reducing forest productivity and carbon storage capacity[36] and causing widespread leaf damage[34].

Since the early 1980s, rainfall conditions have also changed[31]. Peripheral and central parts of the Amazon forest are drying significantly, such as in the southern Bolivian Amazon, where annual rainfall reduced by up to 20 mm yr$^{-1}$ (Extended Data Fig. 3a). By contrast, parts of the western and eastern Amazon forest are becoming wetter, with annual rainfall increasing by up to 20 mm yr$^{-1}$. If these trends continue, ecosystem stability (as in Extended Data Fig. 1) will probably change in parts of the Amazon by 2050, reshaping forest resilience to disturbances (Fig. 1b and Extended Data Fig. 3b). For example, 6% of the biome may change from stable forest to a bistable regime in parts of the southern and central Amazon. Another 3% of the biome may pass the critical threshold in annual rainfall into stable savanna in the southern Bolivian Amazon. Bistable areas covering 8% of the biome may turn into stable forest in the western Amazon (Peru and Bolivia), thus becoming more resilient to disturbances. For comparison with

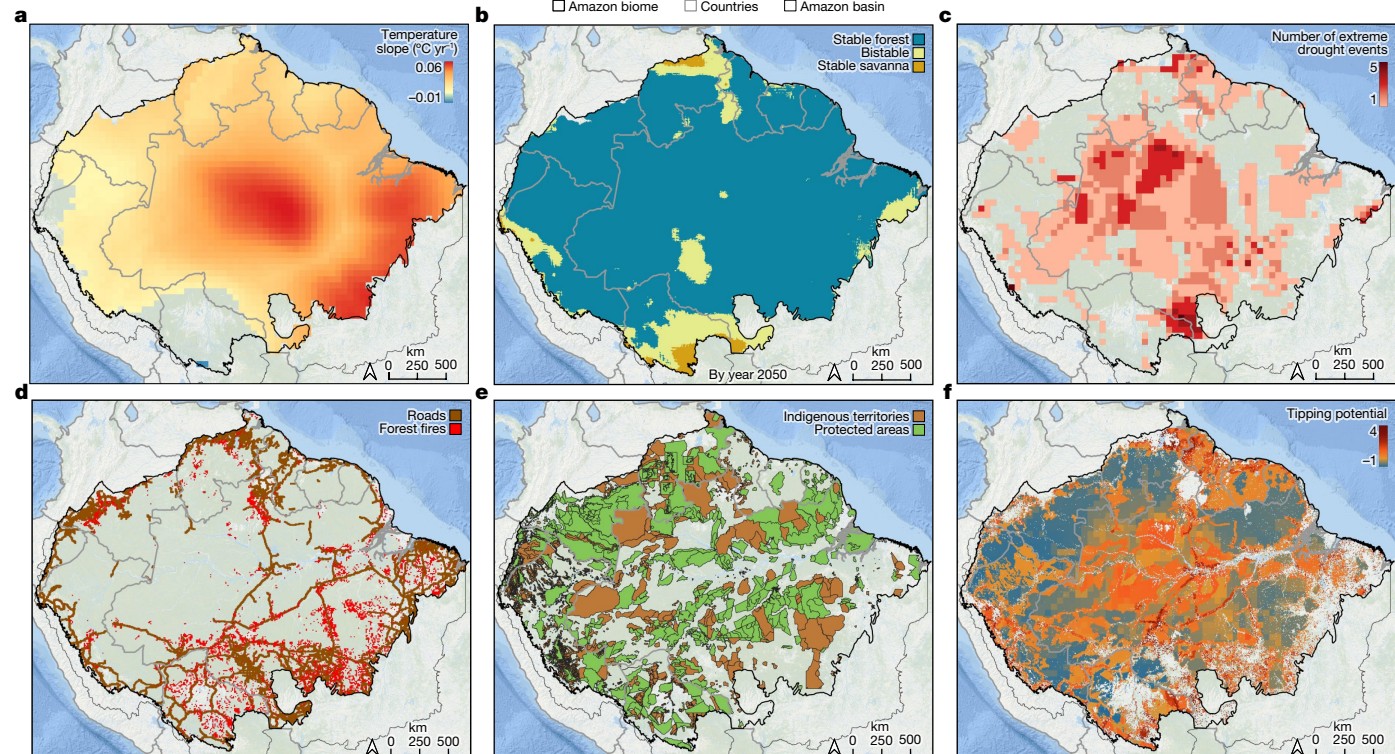

**Fig. 1 | Exploring ecosystem transition potential across the Amazon forest biome as a result of compounding disturbances. a**, Changes in the dry season (July–October) mean temperature reveal widespread warming, estimated using simple regressions between time and temperature observed between 1981 and 2020 (with $P < 0.1$). **b**, Potential ecosystem stability classes estimated for year 2050, adapted from current stability classes (Extended Data Fig. 1b) by considering only areas with significant regression slopes between time and annual rainfall observed from 1981 through 2020 (with $P < 0.1$) (see Extended Data Fig. 3 for areas with significant changes). **c**, Repeated extreme drought events between 2001–2018 (adapted from ref. 39). **d**, Road network from where illegal deforestation and degradation may spread. **e**, Protected areas and Indigenous territories reduce deforestation and fire disturbances. **f**, Ecosystem transition potential (the possibility of forest shifting into an alternative structural or compositional state) across the Amazon biome by year 2050 inferred from compounding disturbances (**a**–**d**) and high-governance areas (**e**). We excluded accumulated deforestation until 2020 and savannas. Transition potential rises with compounding disturbances and varies as follows: less than 0 (in blue) as low; between 1 and 2 as moderate (in yellow); more than 2 as high (orange–red). Transition potential represents the sum of: (1) slopes of dry season mean temperature (as in **a**, multiplied by 10); (2) ecosystem stability classes estimated for year 2050 (as in **b**), with 0 for stable forest, 1 for bistable and 2 for stable savanna; (3) accumulated impacts from extreme drought events, with 0.2 for each event; (4) road proximity as proxy for degrading activities, with 1 for pixels within 10 km from a road; (5) areas with higher governance within protected areas and Indigenous territories, with −1 for pixels inside these areas. For more details, see Methods.

satellite observations, we used projections of ecosystem stability by 2050 based on CMIP6 model ensembles for a low (SSP2–4.5) and a high (SSP5–8.5) greenhouse gas emission scenario (Extended Data Fig. 4 and Supplementary Table 1). An ensemble with the 5 coupled models that include a dynamic vegetation module indicates that 18–27% of the biome may transition from stable forest to bistable and that 2–6% may transition to stable savanna (depending on the scenario), mostly in the northeastern Amazon. However, an ensemble with all 33 models suggests that 35–41% of the biome could become bistable, including large areas of the southern Amazon. The difference between both ensembles is possibly related to the forest–rainfall feedback included in the five coupled models, which increases total annual rainfall and therefore the stable forest area along the southern Amazon, but only when deforestation is not included in the simulations[4,37]. Nonetheless, both model ensembles agree that bistable regions will expand deeper into the Amazon, increasing the risk of critical transitions due to disturbances (as implied by the existence of alternative stable states; Extended Data Fig. 1).

## Disturbance regimes

Within the remaining Amazon forest area, 17% has been degraded by human disturbances[38], such as logging, edge effects and understory fires, but if we consider also the impacts from repeated extreme drought events in the past decades, 38% of the Amazon could be degraded[39]. Increasing rainfall variability is causing extreme drought events to become more widespread and frequent across the Amazon (Fig. 1c), together with extreme wet events and convective storms that result in more windthrow disturbances[40]. Drought regimes are intensifying across the region[41], possibly due to deforestation[42] that continues to expand within the system (Extended Data Fig. 5). As a result, new fire regimes are burning larger forest areas[43], emitting more carbon to the atmosphere[44] and forcing IPLCs to readapt[45]. Road networks (Fig. 1d) facilitate illegal activities, promoting more deforestation, logging and fire spread throughout the core of the Amazon forest[38,39]. The impacts of these pervasive disturbances on biodiversity and on IPLCs will probably affect ecosystem adaptability (Box 1), and consequently forest resilience to global changes.

Currently, 86% of the Amazon biome may be in a stable forest state (Extended Data Fig. 1b), but some of these stable forests are showing signs of fragility[33]. For instance, field evidence from long-term monitoring sites across the Amazon shows that tree mortality rates are increasing in most sites, reducing carbon storage[46], while favouring the replacement by drought-affiliated species[47]. Aircraft measurements of vertical carbon flux between the forest and atmosphere reveal how southeastern forests are already emitting more carbon than they absorb, probably because of deforestation and fire[48].

## Box 1

# Ecosystem adaptability

We define 'ecosystem adaptability' as the capacity of an ecosystem to reorganize and persist in the face of environmental changes. In the past, many internal mechanisms have probably contributed to ecosystem adaptability, allowing Amazonian forests to persist during times of climate change. In this section we synthesize two of these internal mechanisms, which are now being undermined by global change.

### Biodiversity

Amazonian forests are home to more than 15,000 tree species, of which 1% are dominant and the other 99% are mostly rare[107]. A single forest hectare in the central and northwestern Amazon can contain more than 300 tree species (Extended Data Fig. 7a). Such tremendous tree species diversity can increase forest resilience by different mechanisms. Tree species complementarity increases carbon storage, accelerating forest recovery after disturbances[108]. Tree functional diversity increases forest adaptability to climate chance by offering various possibilities of functioning[99]. Rare species provide 'ecological redundancy', increasing opportunities for replacement of lost functions when dominant species disappear[109]. Diverse forests are also more likely to resist severe disturbances owing to 'response diversity'[110]—that is, some species may die, while others persist. For instance, in the rainy western Amazon, drought-resistant species are rare but present within tree communities[111], implying that they could replace the dominant drought-sensitive species in a drier future. Diversity of other organisms, such as frugivores and pollinators, also increases forest resilience by stabilizing ecological networks[15,112]. Considering that half of Amazonian tree species are estimated to become threatened (IUCN Red list) by 2050 owing to climate change, deforestation and degradation[8], biodiversity losses could contribute to further reducing forest resilience.

### Indigenous peoples and local communities

Globally, Indigenous peoples and local communities (IPLCs) have a key role in maintaining ecosystems resilient to global change[113]. Humans have been present in the Amazon for at least 12,000 years[114] and extensively managing landscapes for 6,000 years[22]. Through diverse ecosystem management practices, humans built thousands of earthworks and 'Amazon Dark Earth' sites, and domesticated plants and landscapes across the Amazon forest[115,116]. By creating new cultural niches, humans partly modified the Amazonian flora[117,118], increasing their food security even during times of past climate change[119,120] without the need for large-scale deforestation[117]. Today, IPLCs have diverse ecological knowledge about Amazonian plants, animals and landscapes, which allows them to quickly identify and respond to environmental changes with mitigation and adaptation practices[68,69]. IPLCs defend their territories against illegal deforestation and land use disturbances[49,113], and they also promote forest restoration by expanding diverse agroforestry systems[121,122]. Amazonian regions with the highest linguistic diversity (a proxy for ecological knowledge diversity[123]) are found in peripheral parts of the system, particularly in the north-west (Extended Data Fig. 7b). However, consistent loss of Amazonian languages is causing an irreversible disruption of ecological knowledge systems, mostly driven by road construction[7]. Continued loss of ecological knowledge will undermine the capacity of IPLCs to manage and protect Amazonian forests, further reducing their resilience to global changes[9].

As bistable forests expand deeper into the system (Fig. 1b and Extended Data Fig. 4), the distribution of compounding disturbances may indicate where ecosystem transitions are more likely to occur in the coming decades (Fig. 1f). For this, we combined spatial information on warming and drying trends, repeated extreme drought events, together with road networks, as proxy for future deforestation and degradation[38,39]. We also included protected areas and Indigenous territories as areas with high forest governance, where deforestation and fire regimes are among the lowest within the Amazon[49] (Fig. 1e). This simple additive approach does not consider synergies between compounding disturbances that could trigger unexpected ecosystem transitions. However, by exploring only these factors affecting forest resilience and simplifying the enormous Amazonian complexity, we aimed to produce a simple and comprehensive map that can be useful for guiding future governance. We found that 10% of the Amazon forest biome has a relatively high transition potential (more than 2 disturbance types; Fig. 1f), including bistable forests that could transition into a low tree cover state near savannas of Guyana, Venezuela, Colombia and Peru, as well as stable forests that could transition into alternative compositional states within the central Amazon, such as along the BR319 and Trans-Amazonian highways. Smaller areas with high transition potential were found scattered within deforestation frontiers, where most forests have been carved by roads[50,51]. Moreover, 47% of the biome has a moderate transition potential (more than 1 disturbance type; Fig. 1f), including relatively remote parts of the central Amazon where warming trends and repeated extreme drought events overlap (Fig. 1a,c). By contrast, large remote areas covering 53% of the biome have low transition potential, mostly reflecting the distribution of protected areas and Indigenous territories (Fig. 1e). If these estimates, however, considered projections from CMIP6 models and their relatively broader areas of bistability (Extended Data Fig. 4), the proportion of the Amazon forest that could transition into a low tree cover state would be much larger.

## CO$_2$ fertilization

Rising atmospheric $CO_2$ concentrations are expected to increase the photosynthetic rates of trees, accelerating forest growth and biomass accumulation on a global scale[52]. In addition, $CO_2$ may reduce water stress by increasing tree water-use efficiency[29]. As result, a '$CO_2$ fertilization effect' could increase forest resilience to climatic variability[53,54]. However, observations from across the Amazon[46] suggest that $CO_2$-driven accelerations of tree growth may have contributed to increasing tree mortality rates (trees grow faster but also die earlier), which could eventually neutralize the forest carbon sink in the coming decades[55]. Moreover, increases in tree water-use efficiency may reduce forest transpiration and consequently atmospheric moisture flow across the Amazon[53,56], potentially reducing forest resilience in the southwest of the biome[4,37]. Experimental evidence suggests that $CO_2$ fertilization also depends on soil nutrient availability, particularly nitrogen and phosphorus[57,58]. Thus, it is possible that in the fertile soils of the western Amazon and Várzea floodplains, forests may gain resilience from increasing atmospheric $CO_2$ (depending on how it affects tree mortality rates), whereas on the weathered (nutrient-poor) soils across most of the Amazon basin[59], forests might not respond to atmospheric $CO_2$ increase, particularly on eroded soils within deforestation frontiers[60]. In sum, owing to multiple interacting factors, potential

responses of Amazonian forests to $CO_2$ fertilization are still poorly understood. Forest responses depend on scale, with resilience possibly increasing at the local scale on relatively more fertile soils, but decreasing at the regional scale due to reduced atmospheric moisture flow.

## Local versus systemic transition
### Environmental heterogeneity
Environmental heterogeneity can reduce the risk of systemic transition (large-scale forest collapse) because when stressing conditions intensify (for example, rainfall declines), heterogeneous forests may transition gradually (first the less resilient forest patches, followed by the more resilient ones), compared to homogeneous forests that may transition more abruptly[17] (all forests transition in synchrony). Amazonian forests are heterogeneous in their resilience to disturbances, which may have contributed to buffering large-scale transitions in the past[37,61,62]. At the regional scale, a fundamental heterogeneity factor is rainfall and how it translates into water stress. Northwestern forests rarely experience water stress, which makes them relatively more resilient than southeastern forests that may experience water stress in the dry season, and therefore are more likely to shift into a low tree cover state. As a result of low exposure to water deficit, most northwestern forests have trees with low drought resistance and could suffer massive mortality if suddenly exposed to severe water stress[32]. However, this scenario seems unlikely to occur in the near future (Fig. 1). By contrast, most seasonal forest trees have various strategies to cope with water deficit owing to evolutionary and adaptive responses to historical drought events[32,63]. These strategies may allow seasonal forests to resist current levels of rainfall fluctuations[32], but seasonal forests are also closer to the critical rainfall thresholds (Extended Data Fig. 1) and may experience unprecedented water stress in the coming decades (Fig. 1).

Other key heterogeneity factors (Extended Data Fig. 6) include topography, which determines plant access to groundwater[64], and seasonal flooding, which increases forest vulnerability to wildfires[65]. Future changes in rainfall regimes will probably affect hydrological regimes[66], exposing plateau (hilltop) forests to unprecedented water stress, and floodplain forests to extended floods, droughts and wildfires. Soil fertility is another heterogeneity factor that may affect forest resilience[59], and which may be undermined by disturbances that cause topsoil erosion[60]. Moreover, as human disturbances intensify throughout the Amazon (Fig. 1), the spread of invasive grasses and fires can make the system increasingly homogeneous. Effects of heterogeneity on Amazon forest resilience have been poorly investigated so far (but see refs. 37,61,62) and many questions remain open, such as how much heterogeneity exists in the system and whether it can mitigate a systemic transition.

### Sources of connectivity
Connectivity across Amazonian landscapes and regions can contribute to synchronize forest dynamics, causing different forests to behave more similarly[17]. Depending on the processes involved, connectivity can either increase or decrease the risk of systemic transition[17]. For instance, connectivity may facilitate forest recovery after disturbances through seed dispersal, but also it may spread disturbances, such as fire. In the Amazon, an important source of connectivity enhancing forest resilience is atmospheric moisture flow westward (Fig. 2), partly maintained by forest evapotranspiration[4,37,67]. Another example of connectivity that may increase social-ecological resilience is knowledge exchange among IPLCs about how to adapt to global change[68,69] (see Box 1). However, complex systems such as the Amazon can be particularly vulnerable to sources of connectivity that spread disturbances and increase the risk of systemic transition[70]. For instance, roads carving through the forest are well-known sources of illegal activities, such as logging and burning, which increase forest flammability[38,39].

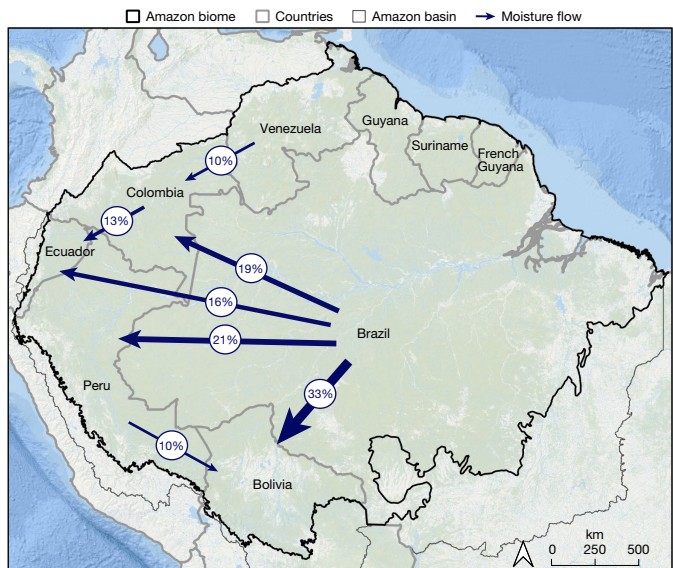

**Fig. 2 | Connectivity between Amazonian countries involving atmospheric moisture flow.** Brazil holds 60% of the Amazon forest biome and has a major responsibility towards its neighbouring countries in the west. Brazil is the largest supplier of rainfall to western Amazonian countries. Up to one-third of the total annual rainfall in Amazonian territories of Bolivia, Peru, Colombia and Ecuador depends on water originating from Brazil's portion of the Amazon forest. This international connectivity illustrates how policies related to deforestation, especially in the Brazilian Amazon, will affect the climate in other countries. Arrow widths are proportional to the percentage of the annual rainfall received by each country within their Amazonian areas. We only show flows with percentages higher than 10% (see Methods for details).

## Five critical drivers of water stress
### Global warming
Most CMIP6 models agree that a large-scale dieback of the Amazon is unlikely in response to global warming above pre-industrial levels[2], but this ecosystem response is based on certain assumptions, such as a large $CO_2$-fertilization effect[53]. Forests across the Amazon are already responding with increasing tree mortality rates that are not simulated by these models[46], possibly because of compounding disturbance regimes (Fig. 1). Nonetheless, a few global climate models[3,14,71–74] indicate a broad range for a potential critical threshold in global warming between 2 and 6 °C (Fig. 3a). These contrasting results can be explained by general differences between numerical models and their representation of the complex Amazonian system. While some models with dynamic vegetation indicate local-scale tipping events in peripheral parts of the Amazon[5,6], other models suggest an increase in biomass and forest cover (for example, in refs. 53,54). For instance, a study found that when considering only climatic variability, a large-scale Amazon forest dieback is unlikely, even under a high greenhouse gas emission scenario[75]. However, most updated CMIP6 models agree that droughts in the Amazon region will increase in length and intensity, and that exceptionally hot droughts will become more common[2], creating conditions that will probably boost other types of disturbances, such as large and destructive forest fires[76,77]. To avoid broad-scale ecosystem transitions due to synergies between climatic and land use disturbances (Fig. 3b), we suggest a safe boundary for the Amazon forest at 1.5 °C for global warming above pre-industrial levels, in concert with the Paris Agreement goals.

### Annual rainfall
Satellite observations of tree cover distributions across tropical South America suggest a critical threshold between 1,000 and 1,250 mm of

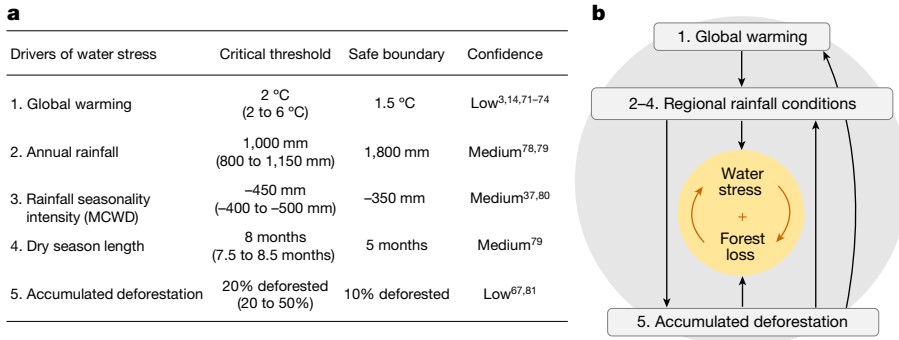

**Fig. 3 | Drivers of water stress on the Amazon forest, their critical thresholds, safe boundaries and interactions. a,** Five critical drivers of water stress on Amazonian forests affect (directly or indirectly) the underlying tipping point of the system. For each driver, we indicate potential critical thresholds and safe boundaries that define a safe operating space for keeping the Amazon forest resilient[11,12]. We followed the precautionary principle and considered the most conservative thresholds within the ranges, when confidence was low. **b,** Conceptual model showing how the five drivers may interact (arrows indicate positive effects) and how these interactions may strengthen a positive feedback between water stress and forest loss. These emerging positive feedback loops could accelerate a systemic transition of the Amazon forest[15]. At global scales,

driver 1 (global warming) intensifies with greenhouse gas emissions, including emissions from deforestation. At local scales, driver 5 (accumulated deforestation) intensifies with land use changes. Drivers 2 to 4 (regional rainfall conditions) intensify in response to drivers 1 and 5. The intensification of these drivers may cause widespread tree mortality for instance because of extreme droughts and fires[76]. Water stress affects vegetation resilience globally[79,104], but other stressors, such as heat stress[34,36], may also have a role. In the coming decades, these five drivers could change at different rates, with some approaching a critical threshold faster than others. Therefore, monitoring them separately can provide vital information to guide mitigation and adaptation strategies.

annual rainfall[78,79]. On the basis of our reanalysis using tree cover data from the Amazon basin (Extended Data Fig. 1a), we confirm a potential threshold at 1,000 mm of annual rainfall (Fig. 3a), below which forests become rare and unstable. Between 1,000 and 1,800 mm of annual rainfall, high and low tree cover ecosystems exist in the Amazon as two alternative stable states (see Extended Data Table 2 for uncertainty ranges). Within the bistability range in annual rainfall conditions, forests are relatively more likely to collapse when severely disturbed, when compared to forests in areas with annual rainfall above 1,800 mm (Extended Data Fig. 1a). For floodplain ecosystems covering 14% of the forest biome, a different critical threshold has been estimated at 1,500 mm of annual rainfall[65], implying that floodplain forests may be the first to collapse in a drier future. To avoid local-scale ecosystem transitions due to compounding disturbances, we suggest a safe boundary in annual rainfall conditions at 1,800 mm.

### Rainfall seasonality intensity
Satellite observations of tree cover distributions across tropical South America suggest a critical threshold in rainfall seasonality intensity at −400 mm of the maximum cumulative water deficit[37,80] (MCWD). Our reanalysis of the Amazon basin (Extended Data Fig. 1c) confirms the critical threshold at approximately −450 mm in the MCWD (Fig. 3a), and suggests a bistability range between approximately −350 and −450 mm (see Extended Data Table 2 for uncertainty ranges), in which forests are more likely to collapse when severely disturbed than forests in areas with MCWD below −350 mm. To avoid local-scale ecosystem transitions due to compounding disturbances, we suggest a safe boundary of MCWD at −350 mm.

### Dry season length
Satellite observations of tree cover distributions across tropical South America suggest a critical threshold at 7 months of dry season length[79] (DSL). Our reanalysis of the Amazon basin (Extended Data Fig. 1d) suggests a critical threshold at eight months of DSL (Fig. 3a), with a bistability range between approximately five and eight months (see Extended Data Table 2 for uncertainty ranges), in which forests are more likely to collapse when severely disturbed than forests in areas with DSL below five months. To avoid local-scale ecosystem transitions due to compounding disturbances, we suggest a safe boundary of DSL at five months.

### Accumulated deforestation
A potential vegetation model[81] found a critical threshold at 20% of accumulated deforestation (Fig. 3a) by simulating Amazon forest responses to different scenarios of accumulated deforestation (with associated fire events) and of greenhouse gas emissions, and by considering a $CO_2$ fertilization effect of 25% of the maximum photosynthetic assimilation rate. Beyond 20% deforestation, forest mortality accelerated, causing large reductions in regional rainfall and consequently an ecosystem transition of 50−60% of the Amazon, depending on the emissions scenario. Another study using a climate-vegetation model found that with accumulated deforestation of 30−50%, rainfall in non-deforested areas downwind would decline[67] by 40% (ref. 67), potentially causing more forest loss[4,37]. Other more recent models incorporating fire disturbances support a potential broad-scale transition of the Amazon forest, simulating a biomass loss of 30−40% under a high-emission scenario[5,82] (SSP5−8.5 at 4 °C). The Amazon biome has already lost 13% of its original forest area due to deforestation[83] (or 15% of the biome if we consider also young secondary forests[83] that provide limited contribution to moisture flow[84]). Among the remaining old-growth forests, at least 38% have been degraded by land use disturbances and repeated extreme droughts[39], with impacts on moisture recycling that are still uncertain. Therefore, to avoid broad-scale ecosystem transitions due to runaway forest loss (Fig. 3b), we suggest a safe boundary of accumulated deforestation of 10% of the original forest biome cover, which requires ending large-scale deforestation and restoring at least 5% of the biome.

### Three alternative ecosystem trajectories
#### Degraded forest
In stable forest regions of the Amazon with annual rainfall above 1,800 mm (Extended Data Fig. 1b), forest cover usually recovers within a few years or decades after disturbances, yet forest composition and functioning may remain degraded for decades or centuries[84–87]. Estimates from across the Amazon indicate that approximately 30% of areas previously deforested are in a secondary forest state[83] (covering 4% of the biome). An additional 38% of the forest biome has been damaged by extreme droughts, fires, logging and edge effects[38,39]. These forests may naturally regrow through forest succession, yet because of

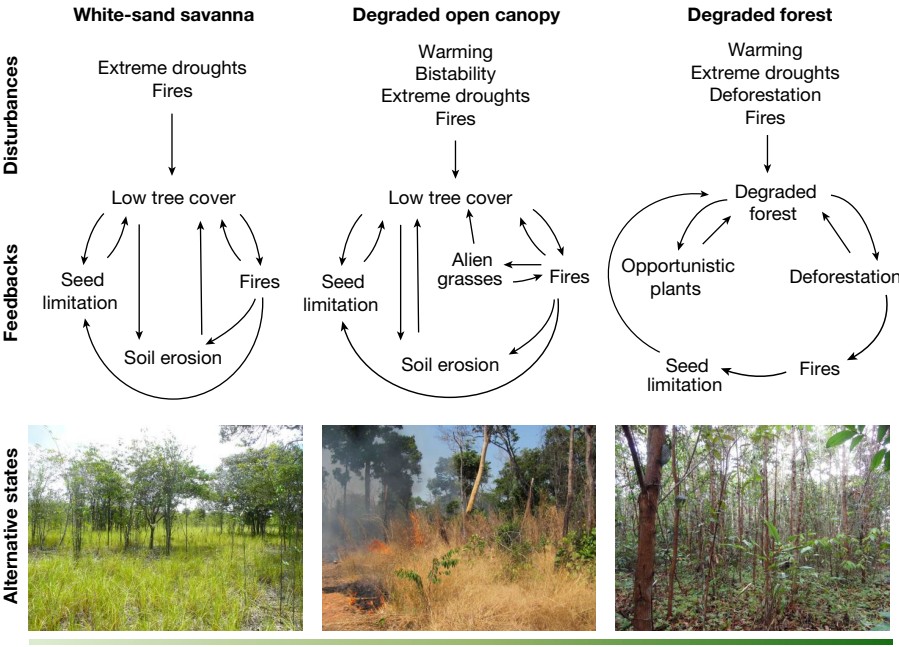

**White-sand savanna**   **Degraded open canopy**   **Degraded forest**

Low tree cover   High tree cover

**Fig. 4 | Alternative ecosystem trajectories for Amazonian forests that transition due to compounding disturbances.** From examples of disturbed forests across the Amazon, we identify the three most plausible ecosystem trajectories related to the types of disturbances, feedbacks and local environmental conditions. These alternative trajectories may be irreversible or transient depending on the strength of the novel interactions[15]. Particular combinations of interactions (arrows show positive effects described in the literature) may form feedback loops[15] that propel the ecosystem through these trajectories. In the 'degraded forest' trajectory, feedbacks often involve competition between trees and other opportunistic plants[85,90,92], as well as interactions between deforestation, fire and seed limitation[84,87,105]. At the landscape scale, secondary forests are more likely to be cleared than mature forests, thus keeping forests persistently young and landscapes fragmented[83].

In the 'degraded open-canopy ecosystem' trajectory, feedbacks involve interactions among low tree cover and fire[97], soil erosion[60], seed limitation[105], invasive grasses and opportunistic plants[96]. At the regional scale, a self-reinforcing feedback between forest loss and reduced atmospheric moisture flow may increase the resilience of these open-canopy degraded ecosystems[42]. In the 'white-sand savanna' trajectory, the main feedbacks result from interactions among low tree cover and fire, soil erosion, and seed limitation[106]. Bottom left, floodplain forest transition to white-sand savanna after repeated fires (photo credit: Bernardo Flores); bottom centre, forest transition to degraded open-canopy ecosystem after repeated fires (photo credit: Paulo Brando); bottom right, forest transition to *Vismia* degraded forest after slash-and-burn agriculture (photo credit: Catarina Jakovac).

feedbacks[15], succession can become arrested, keeping forests persistently degraded (Fig. 4). Different types of degraded forests have been identified in the Amazon, each one associated with a particular group of dominant opportunistic plants. For instance, *Vismia* forests are common in old abandoned pastures managed with fire[85], and are relatively stable, because *Vismia* trees favour recruitment of *Vismia* seedlings in detriment of other tree species[88,89]. Liana forests can also be relatively stable, because lianas self-perpetuate by causing physical damage to trees, allowing lianas to remain at high density[90,91]. Liana forests are expected to expand with increasing aridity, disturbance regimes and $CO_2$ fertilization[90]. *Guadua* bamboo forests are common in the southwestern Amazon[92,93]. Similar to lianas, bamboos self-perpetuate by causing physical damage to trees and have been expanding over burnt forests in the region[92]. Degraded forests are usually dominated by native opportunistic species, and their increasing expansion over disturbed forests could affect Amazonian functioning and resilience in the future.

### White-sand savanna

White-sand savannas are ancient ecosystems that occur in patches within the Amazon forest biome, particularly in seasonally waterlogged or flooded areas[94]. Their origin has been attributed to geomorphological dynamics and past Indigenous fires[26,27,94]. In a remote landscape far from large agricultural frontiers, within a stable forest region of the Amazon (Extended Data Fig. 1b), satellite and field evidence revealed that white-sand savannas are expanding where floodplain forests were

repeatedly disturbed by fires[95]. After fire, the topsoil of burnt forests changes from clayey to sandy, favouring the establishment of savanna trees and native herbaceous plants[95]. Shifts from forest to white-sand savanna (Fig. 4) are probably stable (that is, the ecosystem is unlikely to recover back to forest within centuries), based on the relatively long persistence of these savannas in the landscape[94]. Although these ecosystem transitions have been confirmed only in the Negro river basin (central Amazon), floodplain forests in other parts of the Amazon were shown to be particularly vulnerable to collapse[45,64,65].

### Degraded open-canopy ecosystem

In bistable regions of the Amazon forest with annual rainfall below 1,800 mm (Extended Data Fig. 1b), shifts to degraded open-canopy ecosystems are relatively common after repeated disturbances by fire[45,96]. The ecosystem often becomes dominated by fire-tolerant tree and palm species, together with alien invasive grasses and opportunistic herbaceous plants[96,97], such as vines and ferns. Estimates from the southern Amazon indicate that 5–6% of the landscape has already shifted into degraded open-canopy ecosystems due to deforestation and fires[45,96]. It is still unclear, however, whether degraded open-canopy ecosystems are stable or transient (Fig. 4). Palaeorecords from the northern Amazon[98] show that burnt forests may spend centuries in a degraded open-canopy state before they eventually shift into a savanna. Today, invasion by alien flammable grasses is a novel stabilizing mechanism[96,97], but the long-term persistence of these grasses in the ecosystem is also uncertain.

## Prospects for modelling Amazon forest dynamics

Several aspects of the Amazon forest system may help improve earth system models (ESMs) to more accurately simulate ecosystem dynamics and feedbacks with the climate system. Simulating individual trees can improve the representation of growth and mortality dynamics, which ultimately affect forest dynamics (for example, refs. 61,62,99). Significant effects on simulation results may emerge from increasing plant functional diversity, representation of key physiological trade-offs and other features that determine water stress on plants, and also allowing for community adjustment to environmental heterogeneity and global change[32,55,62,99]. For now, most ESMs do not simulate a dynamic vegetation cover (Supplementary Table 1) and biomes are represented based on few plant functional types, basically simulating monocultures on the biome level. In reality, tree community adaptation to a heterogenous and dynamic environment feeds into the whole-system dynamics, and not covering such aspects makes a true Amazon tipping assessment more challenging.

Our findings also indicate that Amazon forest resilience is affected by compounding disturbances (Fig. 1). ESMs need to include different disturbance scenarios and potential synergies for creating more realistic patterns of disturbance regimes. For instance, logging and edge effects can make a forest patch more flammable[39], but these disturbances are often not captured by ESMs. Improvements in the ability of ESMs to predict future climatic conditions are also required. One way is to identify emergent constraints[100], lowering ESMs variations in their projections of the Amazonian climate. Also, fully coupled ESMs simulations are needed to allow estimates of land-atmosphere feedbacks, which may adjust climatic and ecosystem responses. Another way to improve our understanding of the critical thresholds for Amazonian resilience and how these link to climatic conditions and to greenhouse gas concentrations is through factorial simulations with ESMs. In sum, although our study may not deliver a set of reliable and comprehensive equations to parameterize processes impacting Amazon forest dynamics, required for implementation in ESMs, we highlight many of the missing modelled processes.

## Implications for governance

Forest resilience is changing across the Amazon as disturbance regimes intensify (Fig. 1). Although most recent models agree that a large-scale collapse of the Amazon forest is unlikely within the twenty-first century[2], our findings suggest that interactions and synergies among different disturbances (for example, frequent extreme hot droughts and forest fires) could trigger unexpected ecosystem transitions even in remote and central parts of the system[101]. In 2012, Davidson et al.[102] demonstrated how the Amazon basin was experiencing a transition to a 'disturbance-dominated regime' related to climatic and land use changes, even though at the time, annual deforestation rates were declining owing to new forms of governance[103]. Recent policy and approaches to Amazon development, however, accelerated deforestation that reached 13,000 km$^2$ in the Brazilian Amazon in 2021 (http://terrabrasilis.dpi.inpe.br). The southeastern region has already turned into a source of greenhouse gases to the atmosphere[48]. The consequences of losing the Amazon forest, or even parts of it, imply that we must follow a precautionary approach—that is, we must take actions that contribute to maintain the Amazon forest within safe boundaries[12]. Keeping the Amazon forest resilient depends firstly on humanity's ability to stop greenhouse gas emissions, mitigating the impacts of global warming on regional climatic conditions[2]. At the local scale, two practical and effective actions need to be addressed to reinforce forest–rainfall feedbacks that are crucial for the resilience of the Amazon forest[4,37]: (1) ending deforestation and forest degradation; and (2) promoting forest restoration in degraded areas. Expanding protected areas and Indigenous territories can largely contribute to

these actions. Our findings suggest a list of thresholds, disturbances and feedbacks that, if well managed, can help maintain the Amazon forest within a safe operating space for future generations.

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

# Methods

## Datasets

Our study site was the area of the Amazon basin, considering large areas of tropical savanna biome along the northern portion of the Brazilian Cerrado, the Gran Savana in Venezuela and the Llanos de Moxos in Bolivia, as well as the Orinoco basin to the north, and eastern parts of the Andes to the west. The area includes also high Andean landscapes with puna and paramo ecosystems. We chose this contour to allow better communication with the MapBiomas Amazonian Project (2022; https://amazonia.mapbiomas.org). For specific interpretation of our results, we considered the contour of the current extension of the Amazon forest biome, which excludes surrounding tropical savanna biomes.

We used the Moderate Resolution Imaging Spectroradiometer (MODIS) Vegetation Continuous Fields (VCF) data (MOD44B version 6; https://lpdaac.usgs.gov/products/mod44bv006/) for the year 2001 at 250-m resolution[124] to reanalyse tree cover distributions within the Amazon basin, refining estimates of bistability ranges and critical thresholds in rainfall conditions from previous studies. Although MODIS VCF can contain errors within lower tree cover ranges and should not be used to test for bistability between grasslands and savannas[125], the dataset is relatively robust for assessing bistability within the tree cover range of forests and savannas[126], as also shown by low uncertainty (standard deviation of tree cover estimates) across the Amazon (Extended Data Fig. 8).

We used the Climate Hazards Group InfraRed Precipitation with Station data (CHIRPS; https://www.chc.ucsb.edu/data/chirps)[127] to estimate mean annual rainfall and rainfall seasonality for the present across the Amazon basin, based on monthly means from 1981 to 2020, at a 0.05° spatial resolution.

We used the Climatic Research Unit (CRU; https://www.uea.ac.uk/groups-and-centres/climatic-research-unit)[128] to estimate mean annual temperature for the present across the Amazon basin, based on monthly means from 1981 to 2020, at a 0.5° spatial resolution.

To mask deforested areas until 2020, we used information from the MapBiomas Amazonia Project (2022), collection 3, of Amazonian Annual Land Cover and Land Use Map Series (https://amazonia.mapbiomas.org).

To assess forest fire distribution across the Amazon forest biome and in relation to road networks, we used burnt area fire data obtained from the AQUA sensor onboard the MODIS satellite. Only active fires with a confidence level of 80% or higher were selected. The data are derived from MODIS MCD14ML (collection 6)[129], available in Fire Information for Resource Management System (FIRMS). The data were adjusted to a spatial resolution of 1 km.

## Potential analysis

Using potential analysis[130], an empirical stability landscape was constructed based on spatial distributions of tree cover (excluding areas deforested until 2020; https://amazonia.mapbiomas.org) against mean annual precipitation, MCWD and DSL. Here we followed the methodology of Hirota et al.[104]. For bins of each of the variables, the probability density of tree cover was determined using the MATLAB function ksdensity. Local maxima of the resulting probability density function are considered to be stable equilibria, in which local maxima below a threshold value of 0.005 were ignored. Based on sensitivity tests (see below), we chose the intermediate values of the sensitivity parameter for each analysis, which resulted in the critical thresholds most similar to the ones previously published in the literature.

## Sensitivity tests of the potential analysis

We smoothed the densities of tree cover with the MATLAB kernel smoothing function ksdensity. Following Hirota et al.[104], we used a flexible bandwidth ($h$) according to Silverman's rule of thumb[131]: $h = 1.06\sigma n^{1/5}$, where $\sigma$ is the standard deviation of the tree cover distribution and $n$ is

the number of points. To ignore small bumps in the frequency distributions, we used a dimensionless sensitivity parameter. This parameter filters out weak modes in the distributions such that a higher value implies a stricter criterion to detect a significant mode. In the manuscript, we used a value of 0.005. For different values of this sensitivity parameter, we here test the estimated critical thresholds and bistability ranges (Extended Data Table 2). We inferred stable and unstable states of tree cover (minima and maxima in the potentials) for moving windows of the climatic variables. For mean annual precipitation, we used increments of 10 mm yr⁻¹ between 0 and 3500 mm yr⁻¹. For dry season length, we used increments of 0.1 months between 0 and 12 months. For MCWD, we used increments of 10 mm between −800 mm and 0 mm.

## Transition potential

We quantified a relative ecosystem transition potential across the Amazon forest biome (excluding accumulated deforestation; https://amazonia.mapbiomas.org) to produce a simple spatial measure that can be useful for governance. For this, we combined information per pixel, at 5 km resolution, about different disturbances related to climatic and human disturbances, as well as high-governance areas within protected areas and Indigenous territories. We used values of significant slopes of the dry season (July–October) mean temperature between 1981 and 2020 ($P < 0.1$), estimated using simple linear regressions (at 0.5° resolution from CRU) (Fig. 1a). Ecosystem stability classes (stable forest, bistable and stable savanna as in Extended Data Fig. 1) were estimated using simple linear regression slopes of annual rainfall between 1981 and 2020 ($P < 0.1$) (at 0.05° resolution from CHIRPS), which we extrapolated to 2050 (Fig. 1b and Extended Data Fig. 3). Distribution of areas affected by repeated extreme drought events (Fig. 1c) were defined when the time series (2001–2018) of the MCWD reached two standard deviation anomalies from historical mean. Extreme droughts were obtained from Lapola et al.[39], based on Climatic Research Unit gridded Time Series (CRU TS 4.0) datasets for precipitation and evapotranspiration. The network of roads (paved and unpaved) across the Amazon forest biome (Fig. 1d) was obtained from the Amazon Network of Georeferenced Socio-Environmental Information (RAISG; https://geo2.socioambiental.org/raisg). Protected areas (PAs) and Indigenous territories (Fig. 1e) were also obtained from RAISG, and include both sustainable-use and restricted-use protected areas managed by national or sub-national governments, together with officially recognized and proposed Indigenous territories. We combined these different disturbance layers by adding a value for each layer in the following way: (1) slopes of dry season temperature change (as in Fig. 1a, multiplied by 10, thus between −0.1 and +0.6); (2) ecosystem stability classes estimated for year 2050 (as in Fig. 1b), with 0 for stable forest, +1 for bistable and +2 for stable savanna; (3) accumulated impacts from repeated extreme drought events (from 0 to 5 events), with +0.2 for each event; (4) road-related human impacts, with +1 for pixels within 10 km from a road; and (5) protected areas and Indigenous territories as areas with lower exposure to human (land use) disturbances, such as deforestation and forest fires, with −1 for pixels inside these areas. The sum of these layers revealed relative spatial variation in ecosystem transition potential by 2050 across the Amazon (Fig. 1f), ranging from −1 (low potential) to 4 (very high potential).

## Atmospheric moisture tracking

To determine the atmospheric moisture flows between the Amazonian countries, we use the Lagrangian atmospheric moisture tracking model UTrack[132]. The model tracks the atmospheric trajectories of parcels of moisture, updates their coordinates at each time step of 0.1 h and allocates moisture to a target location in case of precipitation. For each millimetre of evapotranspiration, 100 parcels are released into the atmosphere. Their trajectories are forced with evaporation, precipitation, and wind speed estimates from the ERA5 reanalysis product at 0.25° horizontal resolution for 25 atmospheric layers[133]. Here we use the

runs from Tuinenburg et al.[134], who published monthly climatological mean (2008–2017) moisture flows between each pair of 0.5° grid cells on Earth. We aggregated these monthly flows, resulting in mean annual moisture flows between all Amazonian countries during 2008–2017. For more details of the model runs, we refer to Tuinenburg and Staal[132] and Tuinenburg et al.[134].

## Reporting summary

Further information on research design is available in the Nature Portfolio Reporting Summary linked to this article.

## Data availability

All data supporting the findings of this study are openly available and their sources are presented in the Methods.

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

**Acknowledgements** This work was inspired by the Science Panel for the Amazon (SPA) initiative (https://www.theamazonwewant.org/) that produced the first Amazon Assessment Report (2021). The authors thank C. Smith for providing deforestation rates data used in Extended Data Fig. 5b. B.M.F. and M.H. were supported by Instituto Serrapilheira (Serra-1709-18983) and C.J. (R-2111-40341). A.S. acknowledges funding from the Dutch Research Council (NWO) under the Talent Program Grant VI.Veni.202.170. R.A.B. and D.M.L. were supported by the AmazonFACE programme funded by the UK Foreign, Commonwealth and Development Office (FCDO) and Brazilian Ministry of Science, Technology and Innovation (MCTI). R.A.B. was additionally supported by the Met Office Climate Science for Service Partnership (CSSP) Brazil project funded by the UK Department for Science, Innovation and Technology (DSIT), and D.M.L. was additionally supported by FAPESP (grant no. 2020/08940-6) and CNPq (grant no. 309074/2021-5). C.L. thanks CNPq (proc. 159440/2018-1 and 400369/2021-4) and Brazil LAB (Princeton University) for postdoctoral fellowships. A.E.-M. is supported by the UKRI TreeScapes MEMBRA (NE/V021346/1), the Royal Society (RGS\R1\221115), the ERC TreeMort project (758873) and the CESAB Syntreesys project. R.S.O. received a CNPq productivity scholarship and funding from NERC-FAPESP 2019/07773-1. S.B.H. is supported by the Geneva Graduate Institute research funds, and UCLA's committee on research. J.A.M. is supported by the National Institute of Science and Technology for Climate Change Phase 2 under CNPq grant 465501/2014-1; FAPESP grants 2014/50848-9, the National Coordination for Higher Education and Training (CAPES) grant 88887.136402-00INCT. L.S.B. received FAPESP grant 2013/50531-0. D.N. and N.B. acknowledge funding from the European Union's Horizon 2020 research and innovation programme under grant agreement no. 820970. N.B. has received further funding from the Volkswagen foundation, the European Union's Horizon 2020 research and innovation programme under the Marie Sklodowska-Curie grant agreement no. 956170, as well as from the German Federal Ministry of Education and Research under grant no. 01LS2001A.

**Author contributions** B.M.F. and M.H. conceived the study. B.M.F. reviewed the literature, with inputs from all authors. B.M.F., M.H., N.N., A.S., C.L., D.N, H.t.S. and C.R.C.M. assembled datasets. M.H. analysed temperature and rainfall trends. B.M.F. and N.N. produced the maps in main figures and calculated transition potential. A.S. performed potential analysis and atmospheric moisture tracking. B.M.F. produced the figures and wrote the manuscript, with substantial inputs from all authors. B.S. wrote the first version of the 'Prospects for modelling Amazon forest dynamics' section, with inputs from B.M.F and M.H.

**Competing interests** The authors declare no competing interests.

**Additional information**
**Correspondence and requests for materials** should be addressed to Bernardo M. Flores or Marina Hirota.

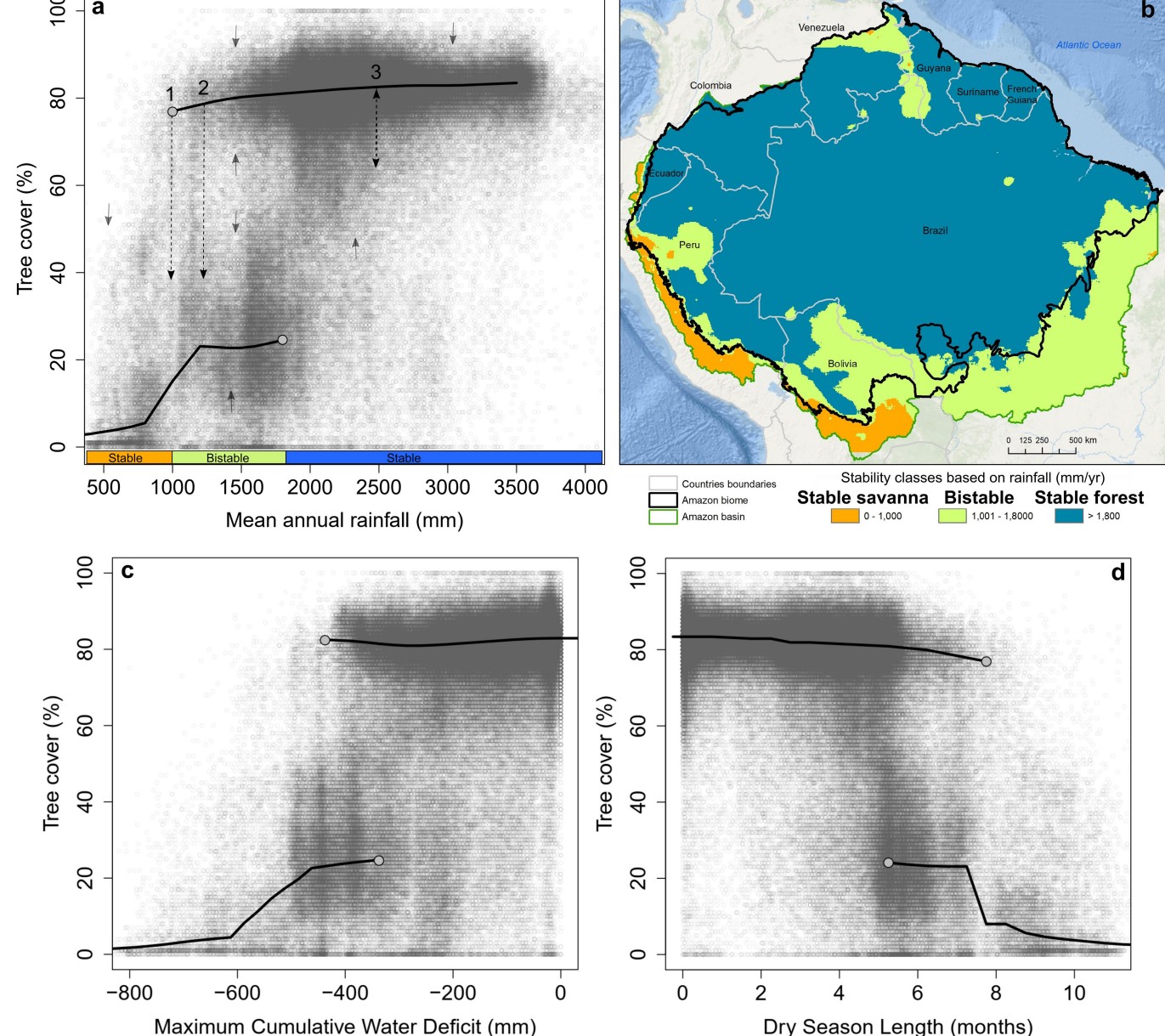

**Extended Data Fig. 1 | Alternative stable states in Amazonian tree cover relative to rainfall conditions.** Potential analysis of tree cover distributions across rainfall gradients in the Amazon basin suggest the existence of critical thresholds and alternative stable states in the system. For this, we excluded accumulated deforestation until 2020 and included large areas of tropical savanna biome in the periphery of the Amazon basin (see Methods). Solid black lines indicate two stable equilibria. Small grey arrows indicate the direction towards equilibrium. (a) The overlap between ~1,000 and 1,800 mm of annual rainfall suggests that two alternative stable states may exist (bistability): a high tree cover state ~ 80 % (forests), and a low tree cover state ~ 20% (savannas). Tree cover around 50 % is rare, indicating an unstable state. Below 1,000 mm of annual rainfall, forests are rare, indicating a potential critical threshold for abrupt forest transition into a low tree cover state[79,104] (arrow 1). Between 1,000 and 1,800 mm of annual rainfall, the existence of alternative stable states implies that forests can shift to a low tree cover stable state in response to disturbances (arrow 2). Above 1,800 mm of annual rainfall, low tree cover becomes rare, indicating a potential critical threshold for an abrupt transition into a high tree cover state. In this stable forest state, forests are expected to always recover after disturbances (arrow 3), although composition may change[47,85]. (b) Currently, the stable savanna state covers 1 % of the Amazon forest biome, bistable areas cover 13 % of the biome (less than previous analysis using broader geographical ranges[78]) and the stable forest state covers 86 % of the biome. Similar analyses using the maximum cumulative water deficit (c) and the dry season length (d) also suggest the existence of critical thresholds and alternative stable states. When combined, these critical thresholds in rainfall conditions could result in a tipping point of the Amazon forest in terms of water stress, but other factors may play a role, such as groundwater availability[64]. MODIS VCF may contain some level of uncertainty for low tree cover values, as shown by the standard deviation of tree cover estimates across the Amazon (Extended Data Fig. 8). However, the dataset is relatively robust for assessing bistability within the tree cover range between forest and savanna[126].

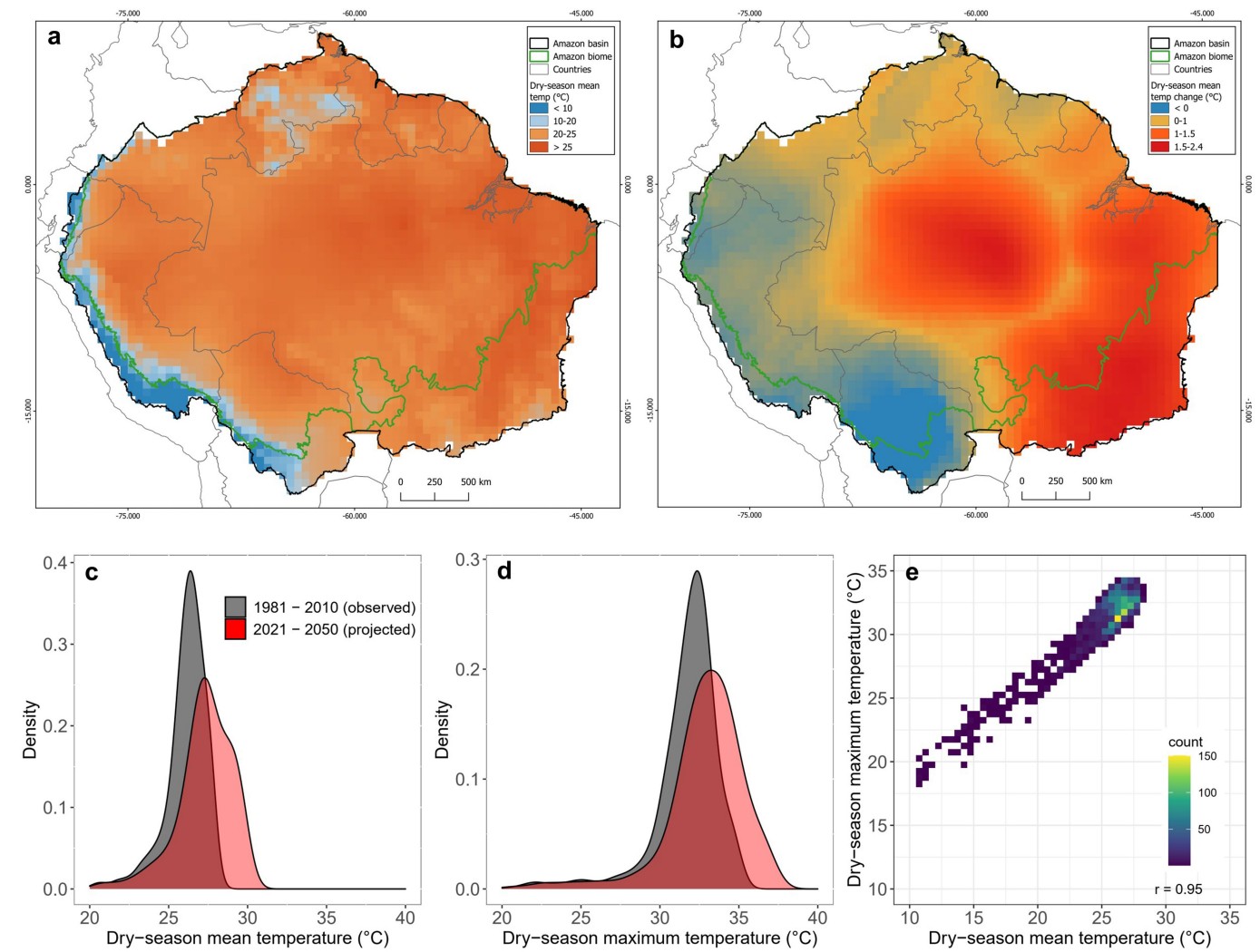

**Extended Data Fig. 2 | Changes in dry-season temperatures across the Amazon basin.** (a) Dry season temperature averaged from mean annual data observed between 1981 and 2010. (b) Changes in dry season mean temperature based on the difference between the projected future (2021–2050) and observed historical (1981–2010) climatologies. Future climatology was obtained from the estimated slopes using historical CRU data[128] (shown in Fig. 1a). (c, d) Changes in the distributions of dry season mean and maximum temperatures for the Amazon basin. (e) Correlation between dry-season mean and maximum temperatures observed (1981–2010) across the Amazon basin ($r$ = 0.95).

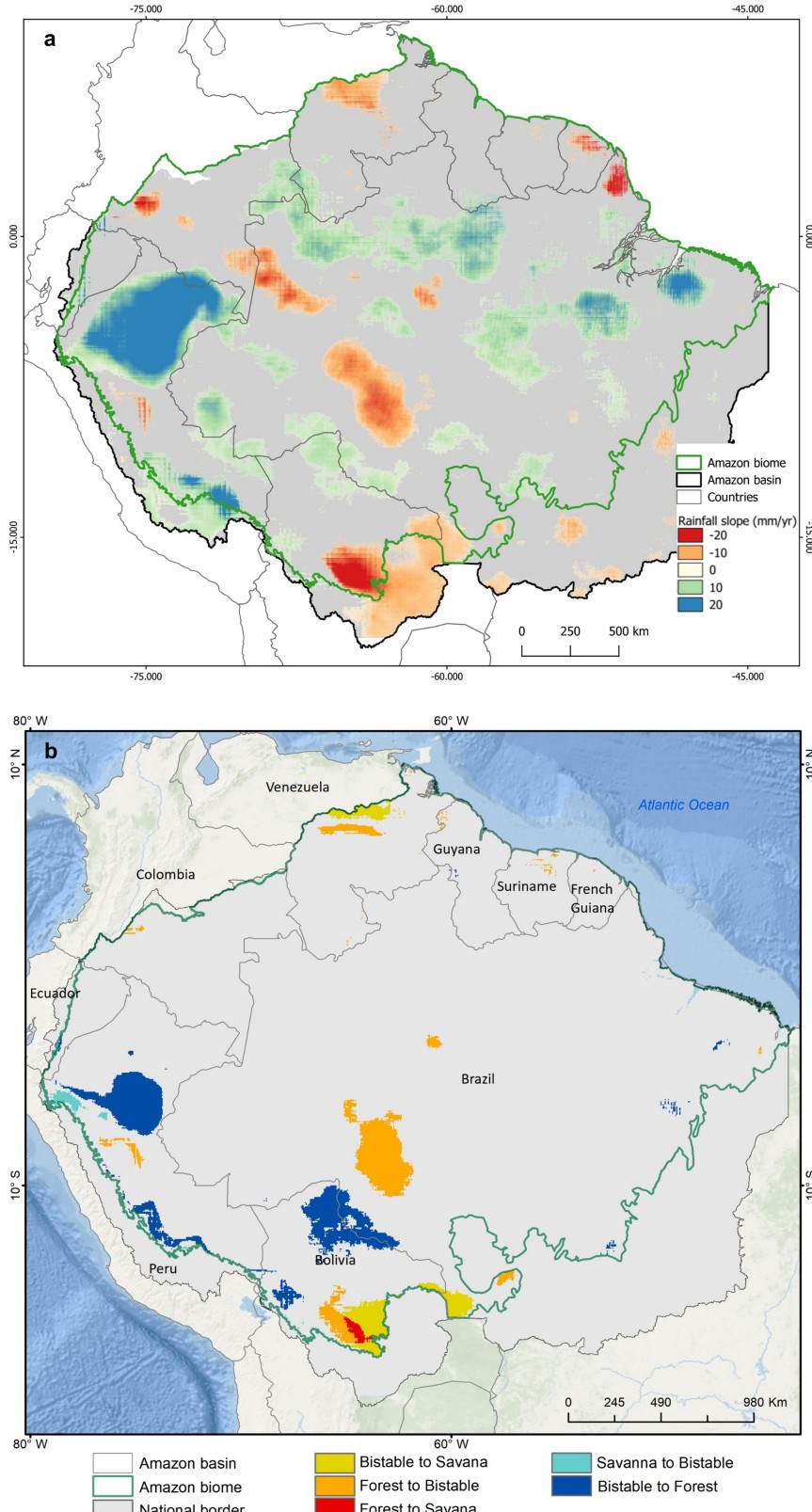

**Extended Data Fig. 3 | Changes in annual precipitation and ecosystem stability across the Amazon forest biome.** (a) Slopes of annual rainfall change between 1981 and 2020 estimated using simple regressions (only areas with significant slopes, $p < 0.1$). (b) Changes in ecosystem stability classes projected for year 2050, based on significant slopes in (a) and critical thresholds in annual rainfall conditions estimated in Extended Data Fig. 1. Data obtained from Climate Hazards Group InfraRed Precipitation with Station data (CHIRPS), at 0.05° spatial resolution[127].

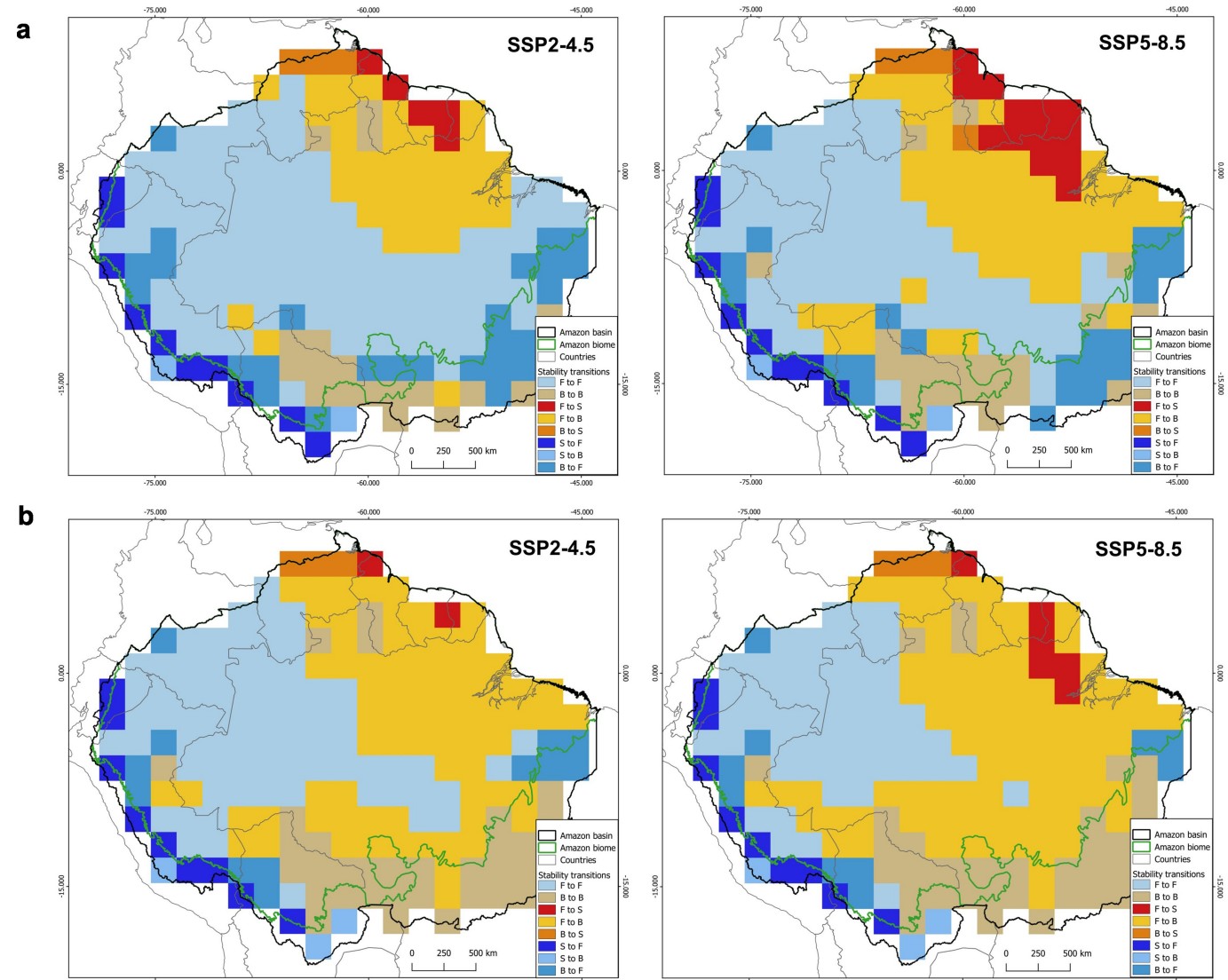

**Extended Data Fig. 4 | Changes in ecosystem stability by 2050 across the Amazon based on annual rainfall projected by CMIP6 models.** (a) Changes in stability classes estimated using an ensemble with the five CMIP6 models that include vegetation modules (coupled for climate-vegetation feedbacks) for two emission scenarios (Shared Socio-economic Pathways - SSPs). (b) Changes in stability classes estimated using an ensemble with all 33 CMIP6 models for the same emission scenarios. Stability changes may occur between stable forest (F), stable savanna (S) and bistable (B) classes, based on the bistability range of 1,000 – 1,800 mm in annual rainfall, estimated from current rainfall conditions (see Extended Data Fig. 1). Projections are based on climate models from the 6th Phase of the Coupled Model Intercomparison Project (CMIP6). SSP2-4.5 is a low-emission scenario of future global warming and SSP5-8.5 is a high-emission scenario. The five coupled models analysed separately in (a) were: EC-Earth3-Veg, GFDL-ESM4, MPI-ESM1-2-LR, TaiESM1 and UKESM1-0-LL (Supplementary Information Table 1).

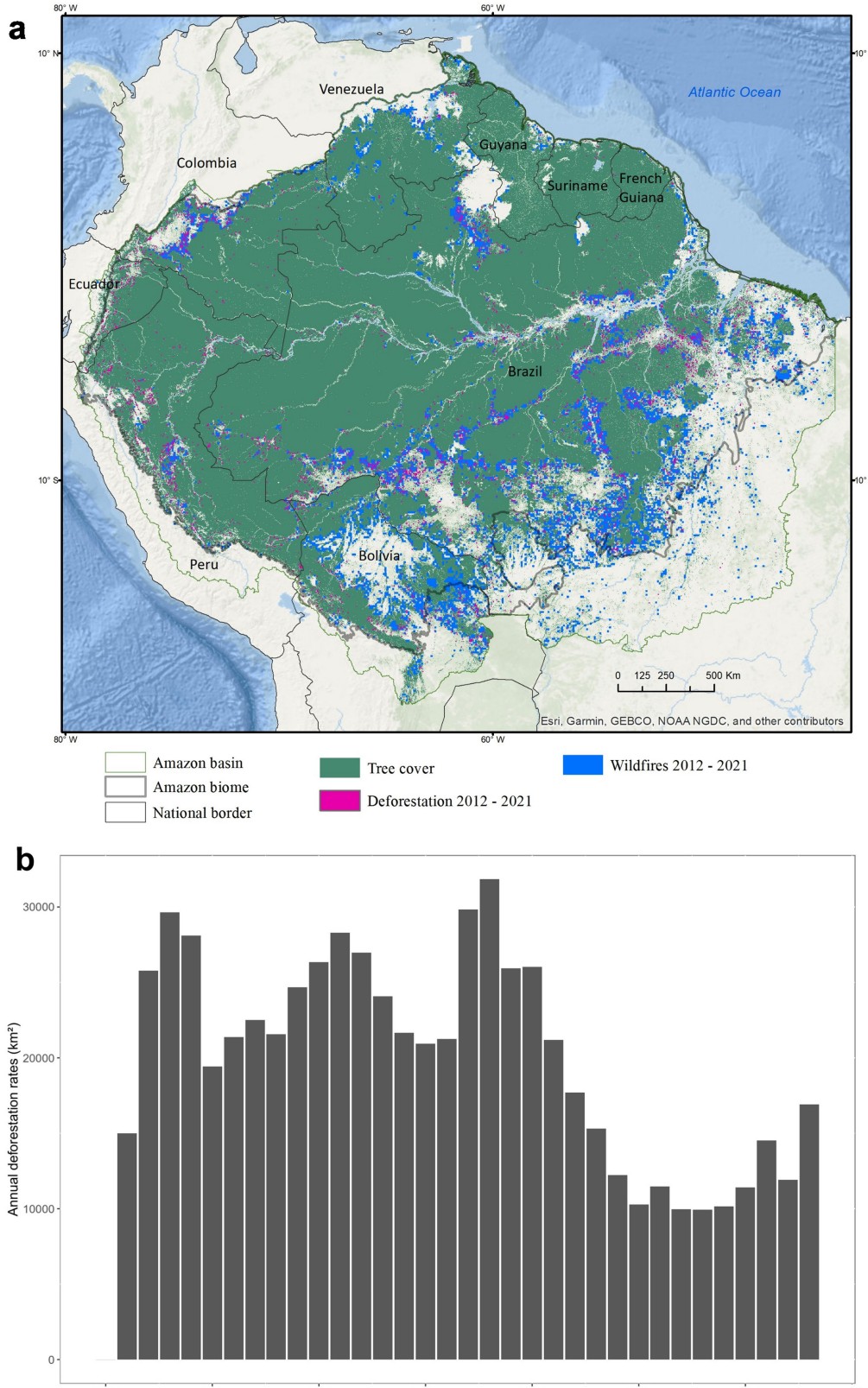

**Extended Data Fig. 5 | Deforestation continues to expand within the Amazon forest system.** (a) Map highlighting deforestation and fire activity between 2012 and 2021, a period when environmental governance began to weaken again, as indicated by increasing rates of annual deforestation in (b). In (b), annual deforestation rates for the entire Amazon biome were adapted with permission from Smith et al.[83].

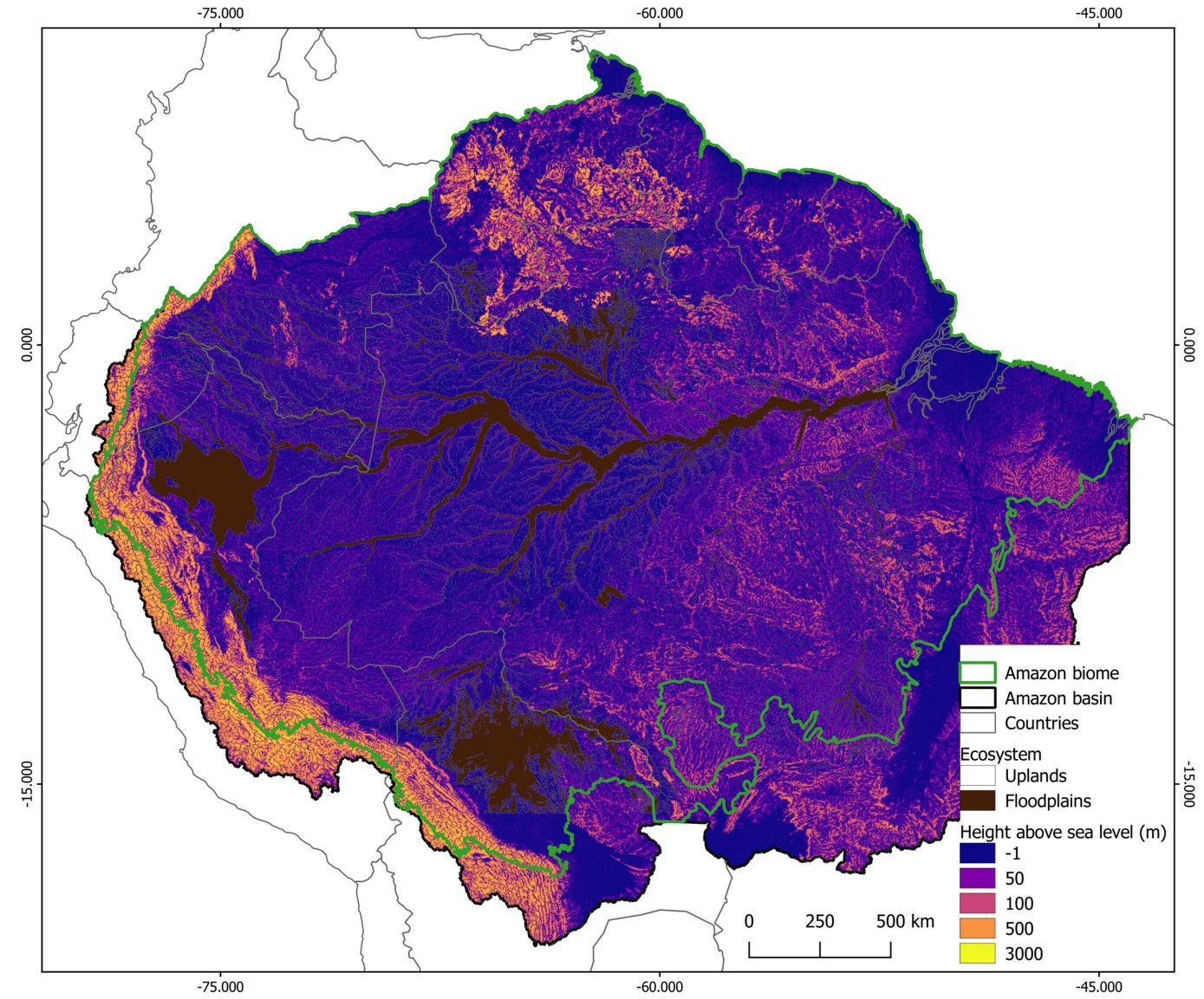

**Extended Data Fig. 6 | Environmental heterogeneity in the Amazon forest system.** Heterogeneity involves myriad factors, but two in particular, related to water availability, were shown to contribute to landscape-scale heterogeneity in forest resilience; topography shapes fine-scale variations of forest drought-tolerance[135,136], and floodplains may reduce forest resilience by increasing vulnerability to wildfires[65]. Datasets: topography is shown by the Shuttle Radar Topography Mission (SRTM; https://earthexplorer.usgs.gov/)[137] at 90 m resolution; floodplains and uplands are separated with the Amazon wetlands mask[138] at 90 m resolution.

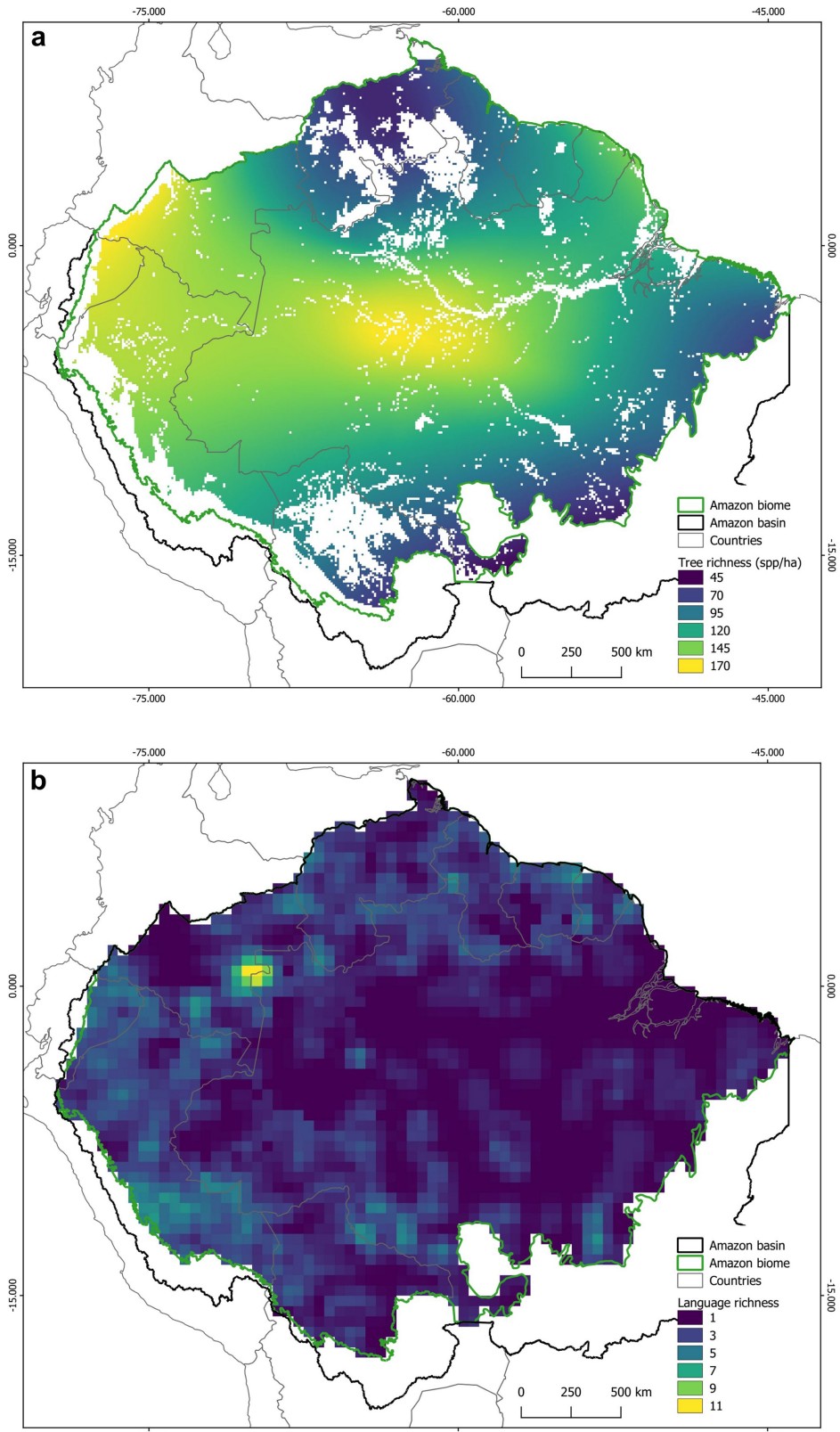

**Extended Data Fig. 7 | The Amazon is biologically and culturally diverse.**
(a) Tree species richness and (b) language richness illustrate how biological and cultural diversity varies across the Amazon. Diverse tree communities and human cultures contribute to increasing forest resilience in various ways that are being undermined by land-use and climatic changes. Datasets: (a) Amazon Tree Diversity Network (ATDN, https://atdn.myspecies.info). (b) World Language Mapping System (WLMS) obtained under license from Ethnologue[139].

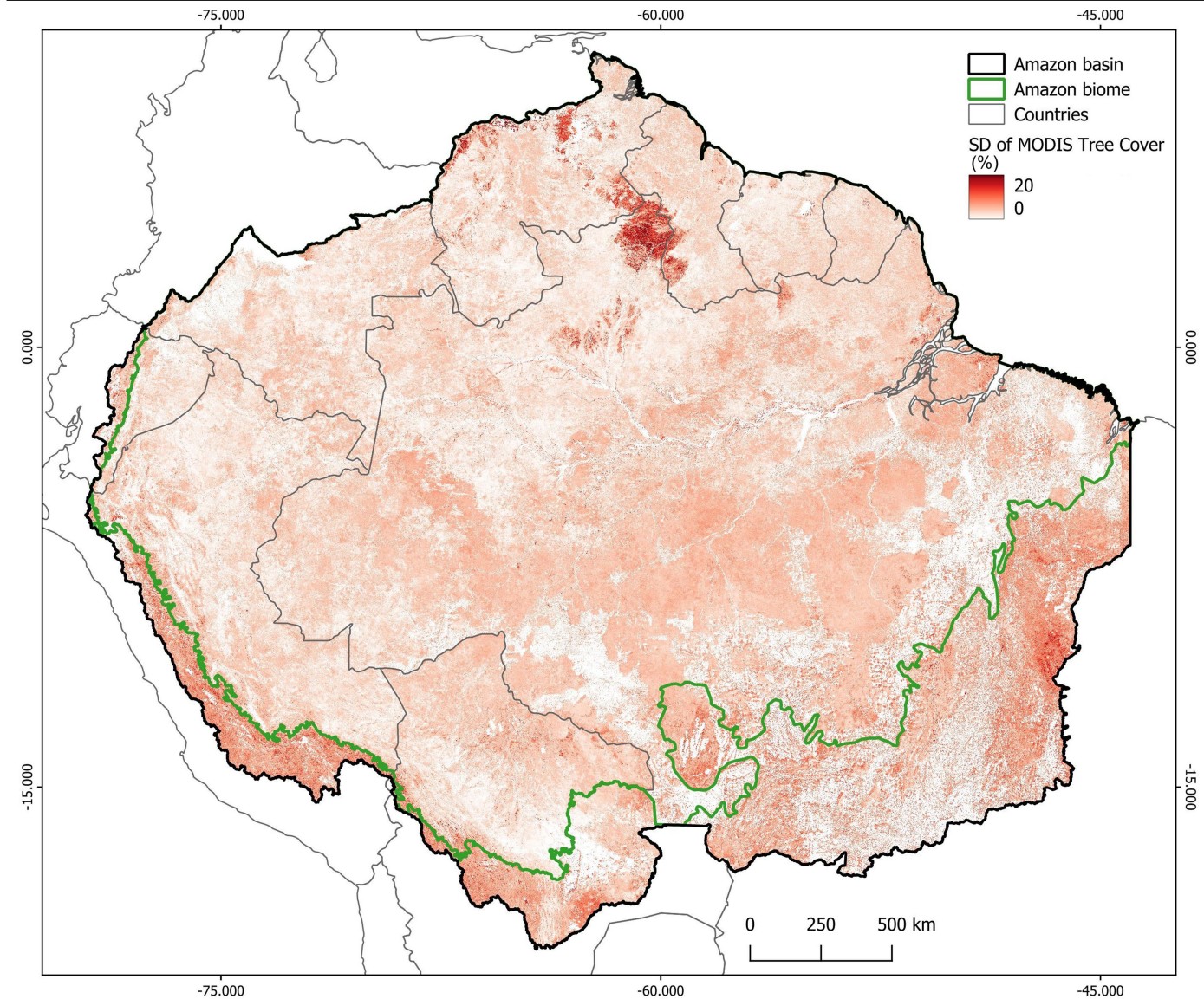

**Extended Data Fig. 8 | Uncertainty of the MODIS VCF dataset across the Amazon basin.** Map shows standard deviation (SD) of tree cover estimates from MODIS VCF[124]. We masked deforested areas until 2020 using the MapBiomas Amazonia Project (2022; https://amazonia.mapbiomas.org).

**Extended Data Table 1 | Examples of positive feedbacks that may affect Amazon forest resilience**

| Positive feedback components | Description | Spatial scale |
|---|---|---|
| Forest - global warming | Forests store carbon, reducing global warming and climatic variability, thus keeping forest carbon stocks more resilient. | Global |
| Forest - rainfall | Forest evapotranspiration by the forest increases atmospheric moisture that flows downwind, increasing rainfall on other forests, thus increasing their resilience. | Regional |
| Drought - deforestation | Intense droughts increase deforestation by facilitating fire use to eliminate dead biomass. Deforestation decreases forest cover and rainfall recycling, increasing drought intensity. | Regional |
| Forest - frugivore animals | Forest trees produce fruits that feed frugivores. Frugivores disperse tree seeds, increasing tree recruitment and forest resilience, including in disturbed sites. | Landscape |
| Forest - fire | Forest cover supresses fire. Fire kills trees, decreasing forest cover and increasing ecosystem flammability. | Landscape |
| Forest - Indigenous Peoples and local communities (IPLCs) | Forest cover increases resources for IPLCs. IPLCs protect the forest and restore degraded areas, increasing forest cover. | Landscape |

Global and regional feedbacks are more likely to propel a large-scale tipping point. Adapted from[15].

**Extended Data Table 2 | Uncertainty ranges in the estimates of critical thesholds for transitions between forest and savanna**

| Sensitivity | MAP (mm/year) | | DSL (months) | | MCWD (mm) | |
|---|---|---|---|---|---|---|
| | F to S | S to F | F to S | S to F | F to S | S to F |
| 0.001 | 780 | 2050 | 8.6 | 4.3 | -590 | -210 |
| 0.002 | 860 | 1930 | 8.4 | 4.7 | -480 | -270 |
| 0.003 | 910 | 1870 | 8.2 | 4.9 | -460 | -310 |
| 0.004 | 950 | 1840 | 8.1 | 5.1 | -450 | -330 |
| **0.005** | **980** | **1810** | **7.9** | **5.2** | **-440** | **-350** |
| 0.006 | 1020 | 1780 | 7.8 | 5.3 | -440 | -360 |
| 0.007 | 1050 | 1750 | 7.6 | 5.4 | -430 | -360 |
| 0.008 | 1080 | 1720 | 7.5 | 5.5 | -430 | -360 |
| 0.009 | 1110 | 1690 | 7.3 | 5.6 | -420 | -370 |
| 0.010 | 1140 | 1660 | 7.6 | 5.7 | -420 | -370 |

A range of possible thresholds results from different values of the sensitivity parameter in the potential analysis, for mean annual precipitation (MAP in mm/year), dry season length (DSL in months) and maximum climatological water deficit (MCWD in mm). F to S means forest to savanna threshold. S to F means savanna to forest threshold.

# Reporting Summary

## Statistics

For all statistical analyses, confirm that the following items are present in the figure legend, table legend, main text, or Methods section.

| n/a | Confirmed | |
|---|---|---|
| ☒ | ☐ | The exact sample size (*n*) for each experimental group/condition, given as a discrete number and unit of measurement |
| ☒ | ☐ | A statement on whether measurements were taken from distinct samples or whether the same sample was measured repeatedly |
| ☒ | ☐ | The statistical test(s) used AND whether they are one- or two-sided *Only common tests should be described solely by name; describe more complex techniques in the Methods section.* |
| ☒ | ☐ | A description of all covariates tested |
| ☒ | ☐ | A description of any assumptions or corrections, such as tests of normality and adjustment for multiple comparisons |
| ☒ | ☐ | A full description of the statistical parameters including central tendency (e.g. means) or other basic estimates (e.g. regression coefficient) AND variation (e.g. standard deviation) or associated estimates of uncertainty (e.g. confidence intervals) |
| ☒ | ☐ | For null hypothesis testing, the test statistic (e.g. $F$, $t$, $r$) with confidence intervals, effect sizes, degrees of freedom and $P$ value noted *Give P values as exact values whenever suitable.* |
| ☒ | ☐ | For Bayesian analysis, information on the choice of priors and Markov chain Monte Carlo settings |
| ☒ | ☐ | For hierarchical and complex designs, identification of the appropriate level for tests and full reporting of outcomes |
| ☒ | ☐ | Estimates of effect sizes (e.g. Cohen's *d*, Pearson's *r*), indicating how they were calculated |

*Our web collection on statistics for biologists contains articles on many of the points above.*

## Software and code

Policy information about availability of computer code

| Data collection | No software was used. |
|---|---|
| Data analysis | Potential analysis - Using potential analysis (Livina et al. 2010), an empirical stability landscape was constructed based on spatial distributions of tree cover against mean annual precipitation (MAP), maximum cumulative water deficit (MCWD) and dry season length (DSL). Here we followed the methodology of Hirota et al. (2011). For bins of each of the variables, the probability density of tree cover was determined using the MATLAB function ksdensity with a bandwidth of 5%. We applied Gaussian weights to the variable with a standard deviation of 0.05 times the range of the variable: 0-3500 mm/yr for MAP, 0-12 months for DSL and -800-0 mm for MCWD. Local maxima of the resulting probability density function are considered to be stable states, in which local maxima below a threshold value of 0.003 were ignored.<br><br>Atmospheric moisture tracking -  To determine the atmospheric moisture flows between the Amazonian countries, we use the Lagrangian atmospheric moisture tracking model UTrack (Tuinenburg & Staal 2020). The model tracks the atmospheric trajectories of parcels of moisture, updates their coordinates at each time step of 0.1 h and allocates moisture to a target location in case of precipitation. For each mm of evapotranspiration, 100 parcels are released into the atmosphere. Their trajectories are forced with evaporation, precipitation, and wind speed estimates from the ERA5 reanalysis product at 0.25º horizontal resolution for 25 atmospheric layers (Hersbach et al. 2020). Here we use the runs from Tuinenburg et al. (2020), who published monthly climatological mean (2008–2017) moisture flows between each pair of 0.5º grid cells on Earth. Here we aggregated these monthly flows, resulting in mean annual moisture flows between all Amazonian countries during 2008–2017. For more details of the model runs, we refer to Tuinenburg & Staal (2020) and Tuinenburg et al. (2020). |

For manuscripts utilizing custom algorithms or software that are central to the research but not yet described in published literature, software must be made available to editors and reviewers. We strongly encourage code deposition in a community repository (e.g. GitHub). See the Nature Portfolio guidelines for submitting code & software for further information.

## Data

Policy information about availability of data

All manuscripts must include a data availability statement. This statement should provide the following information, where applicable:

- Accession codes, unique identifiers, or web links for publicly available datasets
- A description of any restrictions on data availability
- For clinical datasets or third party data, please ensure that the statement adheres to our policy

All datasets used are publicly available, including the land-cover changes and Amazonian contour from MapBiomas Amazonian Project (2022, https://amazonia.mapbiomas.org), tree cover data from the Moderate Resolution Imaging Spectroradiometer (MODIS, https://modis.gsfc.nasa.gov), rainfall data from Climate Hazards Group InfraRed Precipitation with Station data (CHIRPS, https://www.chc.ucsb.edu/data/chirps), temperature data from Climatic Research Unit (CRU, https://www.uea.ac.uk/groups-and-centres/climatic-research-unit), and burnt area also from MODIS (MCD14ML, Collection 6).

## Human research participants

Policy information about studies involving human research participants and Sex and Gender in Research.

| Reporting on sex and gender | This type of information is not part of our study and therefore has not been collected. |
| --- | --- |
| Population characteristics | See above. |
| Recruitment | See above. |
| Ethics oversight | See above. |

Note that full information on the approval of the study protocol must also be provided in the manuscript.

# Field-specific reporting

Please select the one below that is the best fit for your research. If you are not sure, read the appropriate sections before making your selection.

☐ Life sciences    ☐ Behavioural & social sciences    ☒ Ecological, evolutionary & environmental sciences

For a reference copy of the document with all sections, see nature.com/documents/nr-reporting-summary-flat.pdf

# Ecological, evolutionary & environmental sciences study design

All studies must disclose on these points even when the disclosure is negative.

| Study description | Our study is a review of the existing evidence in the literature about the mechanisms that could cause a tipping point in the Amazon forest system, including evidence from modelling studies, paleorecords and observational studies on ecology and climatology. To highlight the most relevant findings, we re-analysed some of the data, focusing on the Amazonian region, using datasets that are publicly available. |
| --- | --- |
| Research sample | Our study site was the area of the Amazon basin, considering parts of the Brazilian Cerrado biome to the south, the Orinoco basin to the north, and eastern parts of the Andes to the west. We chose this contour to allow better communication with the MapBiomas Amazonian Project (2022, https://amazonia.mapbiomas.org). For specific interpretation of our results, we considered the contour of the current extension of the Amazon forest biome, which excludes surrounding tropical savanna biomes.

We used the Vegetation Continuous Fields (VCF) data from the Moderate Resolution Imaging Spectroradiometer (MODIS, https://modis.gsfc.nasa.gov/) for the year 2001 at 250-m resolution (DiMiceli et al. 2011) to analyze tree cover distributions in the Amazon basin.

We used the Climate Hazards Group InfraRed Precipitation with Station data (CHIRPS, https://www.chc.ucsb.edu/data/chirps) (Mitchell and Jones, 2005) to estimate mean annual rainfall and rainfall seasonality for the present across the Amazon basin, based on monthly means from 1981 through 2020, at a 0.05o spatial resolution.

We used the Climatic Research Unit (CRU, https://www.uea.ac.uk/groups-and-centres/climatic-research-unit) (Funk et al. 2015) to estimate mean annual temperature for the present across the Amazon basin, based on monthly means from 1981 through 2020, at a 0.5o spatial resolution.

To mask deforested areas until 2012, we used information from the MapBiomas Amazonia Project (2022), Collection 3, of Amazonian Annual Land Cover & Land Use Map Series, with the link: https://amazonia.mapbiomas.org.

To assess forest fires across the Amazon forest biome, we used burnt area fire data obtained from the AQUA sensor onboard the MODIS satellite. Only active fires with a confidence level of 80% or higher were selected. The data are derived from MODIS |

| | MCD14ML (Collection 6), available in FIRMS – Fire Information for Resource Management System. The data were adjusted to a spatial resolution of 1km, according to the methodology of Silva Junior et al., (2019). |
|---|---|
| Sampling strategy | Our sample sizes are limited by the spatial resolution of the datasets described above and by the area of the Amazon basin. |
| Data collection | Please, see response to "Research sample". |
| Timing and spatial scale | Our temporal and spatial scales are limited by the resolutions of the remote sensing datasets described above. |
| Data exclusions | No data were excluded by our analyses. |
| Reproducibility | We re-analyse previously published and openly available datasets, with focus on the Amazon forest. |
| Randomization | This is not relevant to our study. |
| Blinding | Blinding was not relevant to our study. We analysed all information within our study area. |

Did the study involve field work? ☐ Yes ☒ No

# Reporting for specific materials, systems and methods

We require information from authors about some types of materials, experimental systems and methods used in many studies. Here, indicate whether each material, system or method listed is relevant to your study. If you are not sure if a list item applies to your research, read the appropriate section before selecting a response.

## Materials & experimental systems

| n/a | Involved in the study |
|---|---|
| ☒ | Antibodies |
| ☒ | Eukaryotic cell lines |
| ☒ | Palaeontology and archaeology |
| ☒ | Animals and other organisms |
| ☒ | Clinical data |
| ☒ | Dual use research of concern |

## Methods

| n/a | Involved in the study |
|---|---|
| ☒ | ChIP-seq |
| ☒ | Flow cytometry |
| ☒ | MRI-based neuroimaging |

