## [Peer Review File · Nature]

Manuscript Title: Tipping point in the Amazon forest system

Reviewer Comments & Author Rebuttals

Redactions – Third Party Material

Reviewer Reports on the Initial Version:

Referees' comments:

Referee #1 (Remarks to the Author):

This paper reviews the evidence and immediacy for tipping points being reached in the Amazon, Earth's highest biodiversity ecosystem, and one with climate forcing potentials that make it play an outsized role in the future of the Earth system as a whole. Tipping points have been discussed in the literature for a long time, and became contentious with conclusions at odds from one another. There has been recent work (including this year) on predictions on tipping points from a climatological perspective.

This review goes far beyond climatological perspectives and gives a review of all of the interacting components that conspire to stabilize or destabilize Amazonian forest cover. It is timely, as there has been a lot of work on tipping points and interest in the scientific and general community on the fate of the Amazon, as evidenced in the Science Panel for the Amazon First Assessment Report. It is in large part drawn from information in the SPA report (one would expect this if the SPA report did its job synthesizing information!), and it is the voice of the authors of that same report, and some of the most knowledgeable and well known voices studying the Amazon today. The review is a welcome and important summary, and brings together a large amount of information in one spot and will be of wide interest.

There are areas for improvement and these are discussed in the specific comments below, and a few general comments here.

In the new analyses presented, this paper treats bistability—the forest being able to tip between two states—as exclusively a function of precipitation. This is an oversimplification, and there is a lot of nuance and geography in the cause of that bistability (discussed below) and where it applies. To my mind this is something that needs much better justification and explanation, particularly as the authors' analyses and presentation of bistability extends throughout the Amazon Basin, from the lowlands to high Andes (see Amazon biome vs Amazon basin in figure 2). It also assumes no thermal tolerance changes of trees, or internal migrations/portfolio effects leading to increased resilience, etc., of the trees that make up the forest, even though those could be important sources of climate resilience (though almost certainly not against fire).

Also, early on the paper needs a better discussion of safe operating spaces and the goal of the paper in setting them. This doesn't have to be a lot longer, but it needs to clearly define safe operating space and then talk about the axes that will be discussed. In general, there is jargon in the paper that could be reduced for precision and understanding. For example, biocultural diversity conflates two

very different mechanisms of forest resilience and points of engagement for policy, as the authors discuss. Conflating them in one term hides more than it reveals.

The tipping point potential index and discussion needs to be made more transparent, and I have a larger section below on this. It is an interesting approach, though not without controversy.

Finally, given the importance of deforestation and fire in driving tipping points in Amazonia, I thought the discussion of these factors and their incorporation into the tipping index and prediction was not as thought out or well presented as it could be. The recent Imazon report (<https://imazon.org.br/imprensa/estradas-cortam-ou-se-aproximam-de-41-da-area-de-floresta-na-amazonia-mostra-mapeamento-inedito>) reinforces this. Some kinds of prediction for future disturbance components or drivers like roads and ignition, though fraught, is important, as by 2050 one does not expect them to be the same as today, even if there was not external climatic reinforcement of them. It is possible that we will deforest the Amazon purely for economic motive. (On this line of reasoning, the restoration activities and potential for the SE Amazon is worth expanding as well.). To turn the point around, fire and deforestation are key drivers of the tipping point and decreased resilience, and they should be given at least as deep a treatment as the climatological aspects.

Line-by-line comments:

L53. Does biocultural diversity really enhance adaptability? The Amazon was a rain forest system for ~45 my. Did the addition of human really make the system more adaptable for the last 10-20kyr compared to the previous 44.98 my? I'd rephrase to talk about the changes in human land use and biodiversity rather than conflate them.

L75. Explain acronym as it does not follow from description.

L83. What exactly is meant by safe operating space in this context? Avoid jargon and say precisely. Also, generally, be careful about over-use of jargon.

L150. What is meant by 'sedimentation dynamics' here? Be precise.

L158. Need a section in here about increased CO2 effects. In the past it was cooler during droughts, but CO2 was also about ½ current values for much of the time, creating the potential for high water stress at similar levels of drought. Corlett's (2016) suggestion that current forest drought responses have already been buffered by high CO2, meaning that the past may have been even more severe. It is an open question at this point, but one that needs to be presented when thinking about the past. On further reading there is a section deeper in the paper that treats this, but should be listed up front.

Box 1. This is a beautiful figure of the bistability of tree cover in the Amazon. It suggests a tipping point, but what about soils/underlying geological effects? Shallow, nutrient poor, and/or sandy soils can amplify the effects of drought, or produce the appearance of bistability, but when a second axis

is added, it becomes just stability. Could this be accounted for here? Also, 1500 mm of relatively stable rainfall, either on an annual or interannual timescale, can be very different for trees from a system with high intra- and interannual variability. Does this explain any of the bistability?

More than that, Flores et al. 2017 indicates that much of the bistability is due to floodplain forests—if you separate the forests into varzea and terra firme, the bistability in terra firme largely disappears, being mainly in low-rainfall floodplain systems. Worth addressing here.

Also, in its presentation, the figure is highly misleading—much of the Amazon has been deforested, and that deforestation is concentrated in some of the most interesting places on this map (non-coincidentally). The figure should be modified to show deforested areas and road networks. This will also help with the discussion of anthropogenic fires and tipping-point potential.

L218. Okay, discussion of things raised above here. Needs to be hinted at in the box.

L~230. Feeley et al. 2012 GCB had a discussion of this effect + CO₂ that should be cited somewhere in here in addition to Araujo et al. 2021—former covers the broader Amazon.

L238. Flores et al. 2017 generated a small amount of controversy, but, again, how do the current results compare with those, particularly given the effects of driest quarter, terra firme vs varzea, etc. And, if you were to remove the giant and interesting areas in the upper Madeira (the Bolivian savannas), what inference would we draw from the remaining areas? How much does that one area influence our conclusions on bistability?

L~275. Much of the fire activity in the Amazon, and propagation of fire into natural systems, comes from agricultural land conversion and fire escape. It might be worth stating this explicitly here.

L308. “liana-infested” might be toned down—lianas are as much a part of Amazonia as the trees—it is just when they pass a tipping point that they become problematic. Liana dominated? Liana hyperdominance?

L311. The large extent of bamboo dominated forest (180K km² fide Nelson et al.) is not degraded forest, but a natural feature on the landscape with mammal, bird, reptile, etc., taxa found only there, and its own internal dynamics. It does escape and, like lianas, shifts degraded lands into alternate states, but the presence of bamboo per se is decidedly not pathological in Amazonia, even vast expanses of it.

L347. Another feedback to include here is just human use of the landscape. It opens due to fire but then is grazed or used for agriculture, which is a very strong feedback. I think it is important to distinguish what would happen to the landscape if humans were to disappear tomorrow vs if they are left to business as usual as it informs the scale of possible policy interventions.

L366. I think you can say 55+ million, not 45 thousand. Jaramillo and others have shown that you have rain forest assemblages by the mid-Paleocene, and higher palynological diversity in the South America for rain forest taxa in the Eocene. Obviously no Amazonia per se without the Amazon river,

but, still, 10s of millions of years. The ecological interactions that keep rain forest rain forest are enduring and fundamental consequences of biodiversity.

Figure 2. (a) Color ramp might be reversed, with reddest being hottest. Also, what are the units on the modeled slopes? Text says year, but a low of 0.1/yr still gives almost 4 deg C warming, and a high would give something like 23C! (b) Most of the savanna to bistable is in the high Andes where there now is grassland. Is this worth a few words? Also, it would seem like high elevation fire regimes would prevent this, at least in areas with high enough productivity to carry grass fire. (c) Hard to pick out red for deforestation—different color? (d) can this be stated as a probability?

L406. Intensity  intensify

L429. Great paragraph on CO2 Fertilization—you might make the summary point up front that the potential effects are myriad and poorly understood and poorly constrained, and changeable depending on spatial and temporal scale of inference (what is “good” at the leaf or plant scale may have a negative consequence at the forest scale).

L545. Even though the climate may be mitigated through increasing rainfall, etc. (if that continues), it would be hard to make a statement about forest resilience increasing given the high rates of deforestation, ag conversion, and road building and the fires that they bring. It raises a larger point about resilience and scale. At the tree scale resilience may increase in W Amazonia, but at larger spatial scales anthropogenic clearing and fire could/would swamp that effect outside of large protected areas.

L549. What are the “ancient sources of connectivity”? And, more generally, it is worth talking about fragmentation and land use specifically. There is no mystery about their effects. The bulk of Eastern North America was cleared of forests within a century, and with hand tools.

L550. Here and earlier. Biocultural should be split into “biological” and “cultural” diversity like was done in the very nice section starting L499. The mechanisms for their action are different, and mashing them into one word hides the importance of each.

L558. Reference out of format—should be Hecht 2011?

L567. Citation format.

L568. If GHG levels returned to preindustrial levels would there be no risk of tipping-points in the Amazonian periphery? Human fire after agricultural and other clearing is a powerful force, particularly when coupled with grazing after several rotations of crops. What about the possibility that fragmentation and shifting population/ag practices can create a runaway process, unabated by climate change.

L571. Are -- > is

L1097. Tipping Potential. The time frame for tipping potential is 2050, but some of the indicators makes it seem like the latter, but many are based on past (ending 2020) measurements. An argument could be made that the most recent decade is the best indicator we have of what will happen in the next 30 years, but there are more nuanced ways to look at things like deforestation and fire potential using agricultural and road network trends and predictions. Are those predictions specifically believable—no—but the idea that tipping potential 30 years from now is limited to the current ag clearing and roads doesn't seem to be a valid assumption—there will be much more deforestation. As well, the structure of the tipping potential and its components raises a few questions. First, temperature is taken as dry season temperature, and as the slope of simple linear regressions across the measured time frame. Dry season temperature in the southern Amazon is the coolest time of the year, save for a few months in the late dry season, so the idea that even moderate increases in that per se would be inimical to trees as a growth form is likely not true. It would seem like the interaction with both water stress and clearing practices would be the key harm. Bolivian forests, for example, could take a lot of dry season warming and never come close to the temperatures of a typical central Amazonian forest—the Amazon is huge, and the temperature variability is large. Second, the simple linear extrapolation of rainfall trends to 2050 seems simplistic, particularly when the authors earlier in this paper and other literature have shown the importance of MCWD. Would it make more sense to combine the two and look at actual dry episode predictions? One could think that the linear trends would not be as important as the probability of episodic events, like the megadroughts of 2005 and 2010. The use of deforestation from 2012-2020 as an index of tipping potential through 2050, and the use of current observed fires, seems almost certain to be an underestimate of the impacts of these factors. Road networks will expand, the agricultural network will expand, and ignitions will expand, even if climate were to remain static. At least why not take the same approach of using simple linear extrapolations of road/deforestation/ag activity in any cell through 2050. That would be consistent, at least.

More generally, the authors need to put more transparent justification for the development of these metrics. The paper reviews and discusses a large literature based on model predictions and what they reveal about tipping points. (Several of the authors are authors on these other papers.) The interactions and nonlinearities therein among the factors that lead to forest tipping points are difficult, so the argument could be made for a very simple metric based on the things that factor into the authors' conceptual diagram rather than a more complex model, but that in itself is an interesting statement about how the scientific should use and rely on information in policy, and should be explicitly stated. And, it could be clarified further in the main text that the index is really just the sum of the intensity of five different factors, and that the change from 1-2 may not be equal to the change in 3-4, etc.

Referee #2 (Remarks to the Author):

Review of Flores et al. Tipping points in the Amazon forest system

Overall, the manuscript attempts to assess whether and how much of the Amazon forest may pass a threshold (tipping point) and become a more open system, which would result in important impacts on global and regional climate, local people, and biodiversity.

This is a difficult manuscript to review – it has a bit of new analysis, I think, and the rest is a review. I think it probably works best as a Perspective/Review as there is not enough that is new to call it original research.

I think the new work is estimating the three states of Amazon vegetation related to water availability (in Fig. 1, and ED fig.), some modelling later on (CMIP6 runs), and Fig. 2d, seems to be a new attempt at quantifying where forests may change to more open systems. But in all cases it is not 100% clear what is a new analysis and what is previously published.

The material is written in a series of discrete sections, without much integration across sections. That makes it hard to get clear messages, and leads to some duplication of material. The lack of clear messages starts with the abstract, which contains no quantification or even partial answer to the key question posed.

The topic is very important, and there is lots that I think is useful in the manuscript, but it feels to me that it needs an overhaul in structure and some refinement to be really useful to people.

I have one big and perhaps fundamental disagreement with the authors, as I think there is only actually one threshold, i.e. one tipping point. This threshold is a lack of sufficient water for the maintenance of a closed canopy tropical forest. Three of the five tipping points are annual rainfall, dry season length and water deficit, which are all aspects of the same thing – they are all proxies for a lack of sufficient water for maintenance of the forest. They are also all very highly correlated.

Then, I would argue, the other tipping point, forest loss, is just one of two ways that lack of sufficient water can arrive, as forest loss leads to lower evapotranspiration and water recycling and so lower rainfall downwind. The second way of lacking sufficient water is that the climate changes and leads to dry enough conditions so that a forest cannot persist. I think everything beyond that is a modifier of the forest's ability to resist the change (e.g. disturbance opens the canopy makes droughts more intense) or moves the threshold (e.g. rising CO₂ increases water use efficiency, and so forest can tolerate drier conditions, all else being equal).

Making my argument the other way round, to follow the logic of the authors, we could invent even more tipping points related to different aspects of temperature increases (mean annual temp, or daily max temp), or differing metrics of forest loss. We would end up with huge number of tipping points! I don't think that's scientifically correct, nor very helpful.

There is also a theoretical pure temperature tipping point, but I don't think there is much evidence that under hot wet conditions major die-back is likely.

Lastly, on the major points, some coverage of the papers that show something else, not a tipping point/no tipping point are needed, e.g.

Levine et al 2016. Ecosystem heterogeneity determines the ecological resilience of the Amazon to climate change. PNAS.

There can be major impacts on the forest, without a tipping point per se, as Levine et al. 2016

appear to show. Similarly, mention of the huge impacts for carbon, climate and biodiversity of dramatic changes that fall short of flipping to an open system ought to be mentioned. It is not all or nothing, which some readers could come away with thinking on reading the current manuscript.

One other cross-cutting issue is the treatment of soils. There are essential in terms of modulating the threshold, and shown nicely by the examples in the manuscript of white sands. But soils need more careful inclusion elsewhere, particularly soil texture, as a proxy for soil water holding capacity.

Overall, I wonder if it would be clearer to first examine if there is a tipping point in the Amazon system, and review the evidence. Then with that established, assess where the threshold is for the lack of water, and whether climate change or forest loss can breach it, and if so over how much of the basin. Then deal with the key uncertainties in assessing those thresholds – which can really guide new research -- then finally have a section on what to do and the 'safe operating space'. Feel free to ignore this specific structure, others may work, but I think the current structure isn't working that well.

I have disagreements, but do think an exploration of the Amazon tipping point would be very useful, as it is such a critical topic, and some disagreements among experts are fine.

Section by Section comments.

Introduction

Fig. 1. The authors need to present the most robust choice of rainfall, dry season length and water deficit, and not one arbitrarily. Your data can help. Perhaps fitting a logistic regression to each, and the proxy (rainfall, DSL, MWD) with the steepest slope is the one that best reflects the tipping point.

Supporting Evidence.

I am not sure why there is a section called this. Supporting what?

Paleorecords

To me they show forest resistance and/or resilience in the face of major environmental changes (arrival of humans, glacial to interglacial conditions). But you can see impact of disturbance – some expansion of savanna associated with human-induced fires. This isn't what is currently written.

Line 143, that humans were 'managing' the Amazon 6,000 yrs ago is highly debatable. Their impact was likely v. modest. Suggest rephrasing.

Modelling

Not sure why this comes before observations (or between paleo and contemporary observations). The 'potential' for die-back is not what is interesting, but whether that would happen in the near future.

Observations

Overall, this section is weak, as it is missing data from remote sensing, including the essential work of Gatti et al. 2021 Nature, using CO2 collected from aircraft. Limited data cited from satellites, no mention of the inventory plots data, as relevant data from Brienen et al. 2015, Hubau et al 2020

Nature, nor any disaggregation of the forest responses in intact, logged and burnt forests.

Five tipping points

As above I do not think there are 5 tipping points.

Global temperature rise of 2C

As currently written, this tipping point is really as much part of the water-related tipping point. The reason why there is a risk of breaching a threshold is because droughts increase in frequency and intensity, and I think only secondarily about the heat.

On temperature per se, there is a big literature here, but little is cited.

Saying warming will make it “likely too warm for most Amazon trees” seems speculation, and I doubt it is true. Are you really claiming that warming will be too high for ~8,000 tree species? Needs evidence. There was a push back on the idea of ‘biotic attrition’ some years ago, see Dick et al. 2013 Neogene origins and implied warmth tolerance of Amazon tree species, Ecology and Evolution.

Furthermore, if even some large group of species that continue to form a closed canopy forest can survive a few degrees higher temperatures, then there is no temperature tipping point to savanna, while all else is equal.

Forest Loss

This section should probably should go first as it’s the most immediate.

The relationship of forest loss to a tipping point, is again via water availability, and the lack of moisture recycling. It should be made clear that losing forest is another way of lowering rainfall and reaching the drought tipping point.

Fig.2. The fig 2d is potentially extremely worrying and no doubt controversial, so some details need to be in the main text. I don’t think you can just sum them.

Three alternative ecosystem trajectories

I didn’t understand this section. Alternatives to the tipping points being discussed? I was unclear if these were modifiers of the value of the threshold, or something additional. I didn’t understand why disturbance of a specific soil type was in a grouping with degraded forests per se. If something additional, then perhaps differing terminology would help.

Forest resilience is changing

This is a really useful set of sections. However, I think it would be better to use ‘resistance’ to mean the ability of the system to withstand an environmental change/perturbation, and ‘resilience’ to mean the ability to recover from disturbance/an environmental change, as is common in ecology. For example, higher CO2 levels likely confer resistance, as it increases water use efficiency. Fires likely decrease resilience (ability to recover) and lower resistance (to a second fire).

It seems like each of these sections are, in terms of breaching a tipping point, modifiers, that either

push the system closer to a threshold, or buffer it from reaching the threshold. I wonder if the manuscript works better if you stick to relating it all to tipping points more directly and obviously.

The CO₂ fertilisation section needs more care and more balance, the evidence is much stronger than CO₂ 'may' affect trees. All models assume it; forest theory and contemporary observations are consistent with it, as are very limited experiments. Similarly, on temperature, there is a debate, and some evidence on temperature acclimation that should be noted.

In the Connectivity section, you could also cite Cooper et al. 2020 Regime shifts occur disproportionately faster in larger ecosystems. Nature Comms. As it adds a new-ish worrying development.

Referee #3 (Remarks to the Author):

Thank you for inviting me to review paper: "Tipping Points in the Amazon Forest System" by Flores et al. This has been an enjoyable paper to assess.

It is important to state at the outset that of all the potential iconic tipping points of the climate system, concern about Amazon "dieback" arguably attracts the most concern. So any technical paper or review paper that gets us nearer to determining dangerous levels of climate change (in the context of "die-back") is welcome.

The problem with almost all possible Earth System tipping points is that it is especially difficult to determine the level of global warming at which they will be triggered. Hence, now more than anything else, climate and ecological researchers need to map process understanding onto robust process-based model structures. Such simulation frameworks are usually Earth System Models (ESMs), but there are other illustrative dynamical system structures as well. It is in that context that this paper is reviewed – how can the findings be used to advance ESMs?

So despite the need for ESM improvement, what is noticeable is that this paper is in many ways, quite "model-free". That makes a very refreshing change from the usual reporting of Earth System Model outputs. To say the obvious, more data-led papers provide important benchmarks for future model building. But at the same time, the authors do refer to the climate-ecology-land use couplings as "dynamical systems", hinting that understanding can be readily mapped to numerical models.

In my view, what is missing in the paper is a paragraph (or even small section) on the challenges of modelling, and in particular how the paper findings are expected to be influential. Would a Discussion part work that touches on the following:?

(1) We need to constrain ESMs to tell us more accurately links between future GHG levels (or mean global warming) and the climatological thresholds of Table 1. The current situation of simply regarding all models (e.g. those in Extended Data Table 2) as equally plausible is not satisfactory. Methods exist, of course, to help generate a refined multi-model estimate – Emergent Constraints being one example. Or simple observation trend checking for the contemporary period against ESMs

in the CMIPx ensembles.

(2) Can land surface models be improved to also help turn some of the Confidence levels of Table 1 to “High”? From this study, what missing processes are urgent to get in to land models?

(3) Can we get better direct land use descriptions into ESMs? And can we have factorial ESM simulations to confirm how different scales of spatial heterogeneity aggregate to affect overall dieback risks. What is needed to model the points raised in manuscript section “Environmental heterogeneity?”. To what extent can suppression of land use / deforestation alter the dangerous levels of climate change that determine the overall fate of Amazon tree cover.

(4) Would the authors support recommending developing fast dynamical systems “toy box” models that can explore parameter space and in particular couplings? It would be especially good to quantify what happens if forcings change simultaneously. It is unlikely that the edge of any safe operating space will be approached by just one climate feature. If two or more edges of safe operating space are approached together, then does that potentially bring the tipping point forward? Or make the jump to any new state very fast i.e. highly nonlinear?

(5) Can climate-driven degradation, in conjunction with land use, create an especially dangerous tipping point situation, with strong hysteresis? This means that even with overshooting and attempting to get GHG levels much lower, the system is locked in a new state where the rainforest cannot return.

If the authors like the idea of an extra “roadmap” paragraph, as suggested above, then I think it makes a better ending to the current paper form. In my view, the current paper version ends too abruptly, with a quite specific title “Managing Amazon forest resilience”. At the minimum, there needs to be some mention of next steps required for scientific research, based on the findings shown in this review. Arguably there is a possible disconnect in the paper between the data-led findings and the reported ESM projections, and how they may be combined to refine future estimates of Amazon die-back risk.

I like the use of the UTrack atmospheric moisture tracking dataset to characterise connectivity (Figure 3). As the paper authors will know, there is particular interest in the risk of tipping point “cascades”, but the coupling between effects is poorly understood. Here, the coupling (Figure 3) is between two similar potentially tipping point (e.g. rise of dieback in East Amazon and West Amazon), but I think this is especially novel and important. A potential suggestion for future research might be to compare ESMs (and maybe trends if enough data) of known atmospheric transport against ESM projections. Do raise the profile of this part of the paper, because it starts to answer the spatial connection of tipping points, which is a weak point in understanding what the future will be like as GHGs rise.

Overall, I believe this manuscript can make a substantial contribution to understanding the potential fate of the Amazon rainforest in response to a changing climate. However, I think it needs to be clear that this manuscript does not yet provide process-based equations along with their parameterisation, of the effects considered. Those responsible for developing ESMs are not provided

here with a set of equations ready for numerical discretisation and subsequent mapping onto the land part of ESMs. By the authors' own assessment (Table 1), even after their analysis, certainty in levels of climate to induce tipping points is either "low" or "medium". This honesty is to be commended, and this manuscript will certainly focus the minds of those undertaking future Amazon research.

Despite the points raised above, it needs to be recognised that this manuscript provides a powerful assessment of potential changes to the Amazon rainfall under multiple climate pressures (5 stressors – Table 1). I especially like the way climate stressors are put alongside direct deforestation forcings (Section: "Three alternative ecosystem trajectories"). Too many other papers are either about climate impacts alone or deforestation impacts alone, and never give them equal weighting.

I hope this paper can be published in some form. The paper is in review format and as such, does not determine to high precision the actual dangerous thresholds in climate change to prevent Amazon loss. It does, though, provide one of the most updated assessments of possible drivers and offers a robust roadmap to future research needed. At the risk of repeating the comments above, but with the influential IPCC reports so reliant on ESM projections, I think the paper would gain substantially from a small number of additional sentences that guide model developers on how they can use the findings of this review.

I am happy to see any revised paper version.

Small Points

While I can see from Box 1, first panel, the diagram looks like a typical jump in state that is also present in simple dynamical systems models (e.g. a bifurcation diagram), I am not convinced the paper is really "Grounded in dynamical systems theory" (Abstract).

The authors should maybe note how eventually there may be better integration of ESM projections with contemporary measurements of the physical environment. As an example, in section "Global warming and climatic variability", the same paragraph mentions observed trends and ESM projections, leading to Extended Data Fig 3. It would be good if at some point, similar diagrams to Extended Data Fig 3 are built by a fusion between noted climatic trends and ESMs, then driving ecosystem models.

The Figures are colourful and very informative. However, please pay a little more attention to the presentation. For instance, Figure 2, the legends are small, lat/lon tick marks are impossible to read. Of particular concern is that the colourbar for panel a is difficult to read, and at the minimum should have more levels given (why not make it horizontal, such as under panel d?).

The use of the literature is impressive, and that in itself makes this an especially useful paper.

Please try to avoid emotive language for a scientific paper. I have some sympathy, but somehow it does feel wrong here to describe low emissions scenarios as "optimistic" and high emissions

scenarios as “pessimistic”. The reader will soon come to those conclusions anyway, but it must be written impartially.

Referee #4 (Remarks to the Author):

General Comments:

Reviewer summary: The manuscript presents a review of the state of the science and literature surrounding potential Amazon tipping points. The authors place recommended values on five tipping points with associated confidence and studies supporting these recommendations. Crossing these tipping points are associated with potential ecosystem trajectories that are further presented with associated references and descriptions supported by the literature. This is finished with a discussion of the factors shaping and impacting forest resilience, including climate, disturbance, CO₂ fertilization, environmental site heterogeneity, land atmosphere teleconnections, and diversity for both the natural and human components. The authors conclude with a reflection of the recommendations from a 2012 report and a call to action for our current state. This call to action highlights the importance of global reductions in greenhouse gases, but also local scale action in the Amazon. The authors argue that this local action require local and global collaboration to tackle the challenge of restoring and reversing the degradation of environmental policies in the region, so that we can maintain the Amazon and avoid catastrophic dieback.

Article contribution and overall impact: The Amazon has long been the focus of discussion as an important ecological indicator and tipping point essential to the health of our global system. This paper follows recent IPCC and other reports (Armstrong McKay et al 2022) highlighting the important aspects of the Amazon. The review aligns with these recent reports and studies on potential tipping points for the Amazon. There are multiple factors with the potential for interactions and feedbacks that will impact the Amazon, surrounding region and global system. Previous and recent articles have presented aspects of these potential drivers, interactions and feedbacks. The authors present their review and associated guidance with a wealth of studies from a range of disciplines bringing it together in a concise manner that adds to the discussion. In particular the authors highlight the importance of local communities as a force for conservation in the effort to halt illegal deforestation and degradation. This local voice is an important and essential part of discussion around the importance of the Amazon.

The modeling section needs to be either updated or substantially supplemented to reflect more recent modeling studies and results using more recent CMIP6 models, new data, and updated theory around interactions potential tipping points and increased understanding of feedbacks in the Amazon. Some of these updated studies are included in other sections, but the older modeling studies need to be replaced with new results, or presented in a format that demonstrates the progress made in modeling of the Amazon. Detailed comments are provided below.

Detailed comments:

Line 83: Update this to “to help in creating a safe operating...”

Line 125-127: Use a different example of feedbacks that increase fire. Andrae et al 2008 does not

make a link between smoke and decreased rainfall amount, rather they cite a delay in the initiation of precipitation and increased vigor of the storms. Perhaps Kumar et al 2022 for the interactions of land use with fire sensitivity.

Kumar, S., Getirana, A., Libonati, R., Hain, C., Mahanama, S., and Andela, N.: Changes in land use enhance the sensitivity of tropical ecosystems to fire-climate extremes, *Sci Rep*, 12, 964, <https://doi.org/10.1038/s41598-022-05130-0>, 2022.

Line 159: The impacts on the Amazon are not as clear in this reference, and suggest replacing it with Boulton et al 2022

Boulton, C. A., Lenton, T. M., and Boers, N.: Pronounced loss of Amazon rainforest resilience since the early 2000s, *Nat. Clim. Chang.*, 12, 271–278, <https://doi.org/10.1038/s41558-022-01287-8>, 2022.

Line 161-162: Update or supplement this with more recent modeling studies such as (Brando et al., 2020; Burton et al., 2022; Staal et al., 2020; Zemp et al., 2017)

Brando, P. M., Soares-Filho, B., Rodrigues, L., Assunção, A., Morton, D., Tuchsneider, D., Fernandes, E. C. M., Macedo, M. N., Oliveira, U., and Coe, M. T.: The gathering firestorm in southern Amazonia, *Sci. Adv.*, 6, eaay1632, <https://doi.org/10.1126/sciadv.aay1632>, 2020.

Burton, C., Kelley, D. I., Jones, C. D., Betts, R. A., Cardoso, M., and Anderson, L.: South American fires and their impacts on ecosystems increase with continued emissions, *Climate Resilience*, 1, <https://doi.org/10.1002/cli2.8>, 2022.

Staal, A., Fetzer, I., Wang-Erlandsson, L., Bosmans, J. H. C., Dekker, S. C., van Nes, E. H., Rockström, J., and Tuinenburg, O. A.: Hysteresis of tropical forests in the 21st century, *Nat Commun*, 11, 4978, <https://doi.org/10.1038/s41467-020-18728-7>, 2020.

Zemp, D. C., Schleussner, C.-F., Barbosa, H. M. J., Hirota, M., Montade, V., Sampaio, G., Staal, A., Wang-Erlandsson, L., and Rammig, A.: Self-amplified Amazon forest loss due to vegetation-atmosphere feedbacks, *Nat Commun*, 8, 14681, <https://doi.org/10.1038/ncomms14681>, 2017.

Line 165: Update this to reflect a more recent reference

Line 174: Update to “which is incredibly challenging to capture in a model, for instance...”

Line 171-173: This reference (Chai et al 2021) considers CMIP5 models to explore the impacts of climate on Amazon dieback, and should be replaced with literature for the more recent CMIP6 set of models. Or include a separate sentence highlighting the numerous studies showing that CMIP6 models show agreement and demonstrate increased drying and likelihood of drought in the Amazon.

Cook, B. I., Mankin, J. S., Marvel, K., Williams, A. P., Smerdon, J. E., and Anchukaitis, K. J.: Twenty-First Century Drought Projections in the CMIP6 Forcing Scenarios, *Earth's Future*, 8,

<https://doi.org/10.1029/2019EF001461>, 2020.

Parsons, L. A.: Implications of CMIP6 Projected Drying Trends for 21st Century Amazonian Drought Risk, *Earth's Future*, 8, <https://doi.org/10.1029/2020EF001608>, 2020.

Ukkola, A. M., De Kauwe, M. G., Roderick, M. L., Abramowitz, G., and Pitman, A. J.: Robust Future Changes in Meteorological Drought in CMIP6 Projections Despite Uncertainty in Precipitation, *Geophys. Res. Lett.*, 47, <https://doi.org/10.1029/2020GL087820>, 2020.

Line 178: Update to “can help by projecting more realistic...”

Line 178-181: Supplement this sentence with the more recent modeling study (such as Burton et al 2022) demonstrating that with increased temperature, drying, land-use and fire there is a potential for 30% loss of vegetation C with temperatures of 4 degree C.

Burton, C., Kelley, D. I., Jones, C. D., Betts, R. A., Cardoso, M., and Anderson, L.: South American fires and their impacts on ecosystems increase with continued emissions, *Climate Resilience*, 1, <https://doi.org/10.1002/cli2.8>, 2022.

Line 220-224: This result for Jones et al 2009 is quite outdated, and must be updated with more recent model studies such as those suggested earlier and within Armstrong McKay et al 2022.

Line 223-224: It is unclear why these modeling results are being considered as separate in this section.

Line 290: Update to “forest resilience are ...”

Line 350-351: This is a bold statement. Consider rephrasing to a main driver or provide some qualification on this degradation.

Line 376-377: These are old references (Betts et al 2012 and Malhi et al 2009). Update with more recent studies such as those previously suggested.

Line 468-470: Clarify what tipping points from table 1 that the seasonal forests are close to reaching or exceeding.

Line 505: Update to “to changing stress conditions”

Line 558: Add Hetch et al 2011 to numbered reference list.

Figures

Figure 2c: Consider a different set of colors. It is hard to see the small fine scale changes of fire (blue) against the forest (green).

Figure 2d: Update the color of this figure and add more tick mark labels. It is hard to distinguish

between the gradients and identify where tipping point of 2 is located.

References:

Line 716: Update the citation for ref 50: Armstrong McKay et al 2022

Armstrong McKay, D. I., Staal, A., Abrams, J. F., Winkelmann, R., Sakschewski, B., Loriani, S., Fetzer, I., Cornell, S. E., Rockström, J., and Lenton, T. M.: Exceeding 1.5°C global warming could trigger multiple climate tipping points, *Science*, 377, eabn7950, <https://doi.org/10.1126/science.abn7950>, 2022.

Referee #5 (Remarks to the Author):

Flores and colleagues review the main tipping point elements in Amazonia. The manuscript provides both quantitative and qualitative estimates of how much of the Amazon could be at risk due to changes in precipitation, rainfall seasonality, and air temperature. The manuscript also reviews how deforestation could impact forest stability elsewhere across the Basin. Another important contribution of the review relates to qualifying what type of ecosystems could replace Amazon forests if a tipping point is surpassed, providing a glimpse of likely ecosystem trajectories in the near future. Finally, the review assesses the ongoing environmental and climatological changes in Amazonia, and the options to manage Amazon forest resilience. While the manuscript addresses an important topic for conservation, climate change mitigation, and livelihood of millions of people, more detailed and clarification are needed.

1) The section about the five (four?) potential tipping points in Amazonia could provide a more in-depth understanding of the 1) specific mechanisms driving the potential changes in vegetation stability, 2) nuances related to where and when those changes would likely to occur, and 3) knowledge gaps preventing us from properly predicting where and when those changes would likely occur. Although there is a section about environmental heterogeneity, it is vague and does not address all tipping point elements.

2) The definition of tipping points relies on the assumption that once a threshold is crossed, Amazonian forests in some parts of the Basin should be stuck in an alternative (stable) state, in part, because of feedbacks or intrinsic dynamics. However, there is very little discussion on whether these feedbacks are strong enough in Amazonia to drive long-term changes in vegetation stability. Instead, the review provides evidence that there are two basins of attractions (low and high tree canopy cover) and that Amazonian forests have been under pressure due to several stressors—which is important elements to push forests over the edge but not necessarily to maintain forests into that state.

3) Although the manuscript mentions that one tipping point in Amazonia is 20% of deforestation, there is a large body of work on this topic. I encourage the authors to provide a more balanced review of the current literature here, presenting that number as a hypothesis rather than a consensus. Also, the manuscript should provide more nuanced view about where major transformations in forest structure/composition could occur due to deforestation-driven changes in

climate. For instance, in wetter portions of the Amazon, major changes in forest stability due to indirect effects of deforestation are less likely to happen.

4) The manuscript could be clearer about the likelihood of different regions of the Amazon being impacted by the different types of disturbances (The alternative ecosystem trajectories section). Some regions experience some types of disturbances more more often than others. For instance, the western Amazon is more likely to be impacted by blowdown events, while other regions more likely to burn during dry years. Also, primary forest transitions into different forest types could be presented more clearly, given that not all the transitions are likely to occur with the same probability. How much of the Amazon could become a white- sand savanna? Probably a small area, whereas the entire Amazon could become degraded forests.

5) Disturbance regimes: it would be helpful to separate anthropogenic disturbances from natural ones, as well as provide a balanced view of how they may interact.

Referee #6 (Remarks to the Author):

A. Summary of the key results

This manuscript presents two assertions: (1) that an ecological tipping point of reduced rainfall due to loss of tree canopy cover having the potential to shift the vegetation of the region from stable forest to stable savanna and/or unstable forest exists for the Amazon forest, and (2) that the region is currently approaching that tipping point.

B. Originality and significance: if not novel, please include reference

The assertion is supported mainly by an extensive literature review, as well as a correlation analysis of remotely sensed tree canopy cover against mean annual rainfall. Additional analyses are presented in the Supplementary Information, but the results of these are not well integrated into the paper.

Beyond the seminal ecological literature by Holling and others on systems theory, I am not qualified to review the thoroughness of the literature review on the Amazon region as a subject. However, I am qualified to review the basic ecology and application of remotely sensed tree-cover data, so there is where I will place my focus in the following sections C and D.

C. Data & methodology: validity of approach, quality of data, quality of presentation

The overall assertion and the analysis in Figure 1 rest on the assumption that rainfall is the driver of the tipping point. This is reasonable, but indirect or approximate. More directly, tree cover affects rainfall through changes in evapotranspiration and vapor pressure deficit, as noted on Line 128, which then result in changes in rainfall downwind. Barring a more complete analysis with a coupled

vegetation-atmosphere model, which is understandably beyond the scope of this analysis, it would benefit the paper considerably to include scatterplots for one or both of these variables alongside the one for rainfall in Figure 1.

Additionally, and far more importantly, the use of the MODIS VCF Tree Cover layer relies on the assumption that tree cover is determined by climatological constraints. However, this is not true for the Amazon region. The dominant cause of tree canopy loss in the region is land-use conversion of forest to agriculture. Taking a single temporal snapshot of a dataset and neglecting to distinguish climatological from land-use effects invalidates this pivotal result in the paper. This neglect is aggravated by the exclusion of savanna biomes from the analysis.

D. Appropriate use of statistics and treatment of uncertainties

Being mainly a literature review, this criterion is mainly relevant for the correlation analyses presented in Figure 1. The scatterplot is certainly compelling visually, but little quantitative analysis is presented. The Supplementary Information mentions a potential analysis, but only the specification of parameters was described--not how they were chosen or the sensitivities of the results to those choices.

E. Conclusions: robustness, validity, reliability

The conclusions that the Amazon region has and is crossing an ecological tipping point might be supported by the literature review presented. However, they are not supported by the analysis in Figure 1, which fails to distinguish land-use from climatological drivers on tree-canopy density.

F. Suggested improvements: experiments, data for possible revision

The authors must decide if the empirical test of the tipping point presented in Figure 1 is central to their argument. If not, and the literature review is sufficient, then the analysis should be removed. If it is necessary, it must be strengthened by removing the currently unincorporated effect of land use on tree canopy cover. Additionally, a more thorough treatment would repeat the left panel of Figure 1 for evapotranspiration and/or vapor pressure deficit.

G. References: appropriate credit to previous work?

Yes

H. Clarity and context: lucidity of abstract/summary, appropriateness of abstract, introduction and conclusions

As a review, the main manuscript was clear. However, as an analysis, the methods were inadequately described in the Supplementary Information.

Author Rebuttals to Initial Comments:

Referees' comments:

Referee #1 (Remarks to the Author):

This paper reviews the evidence and immediacy for tipping points being reached in the Amazon, Earth's highest biodiversity ecosystem, and one with climate forcing potentials that make it play an outsized role in the future of the Earth system as a whole. Tipping points have been discussed in the literature for a long time, and became contentious with conclusions at odds from one another. There has been recent work (including this year) on predictions on tipping points from a climatological perspective.

This review goes far beyond climatological perspectives and gives a review of all of the interacting components that conspire to stabilize or destabilize Amazonian forest cover. It is timely, as there has been a lot of work on tipping points and interest in the scientific and general community on the fate of the Amazon, as evidenced in the Science Panel for the Amazon First Assessment Report. It is in large part drawn from information in the SPA report (one would expect this if the SPA report did its job synthesizing information!), and it is the voice of the authors of that same report, and some of the most knowledgeable and well known voices studying the Amazon today. The review is a welcome and important summary, and brings together a large amount of information in one spot and will be of wide interest.

There are areas for improvement and these are discussed in the specific comments below, and a few general comments here.

Thank you for these constructive criticisms, which helped us to improve our manuscript.

In the new analyses presented, this paper treats bistability—the forest being able to tip between two states—as exclusively a function of precipitation. This is an oversimplification, and there is a lot of nuance and geography in the cause of that bistability (discussed below) and where it applies. To my mind this is something that needs much better justification and explanation, particularly as the authors’ analyses and presentation of bistability extends throughout the Amazon Basin, from the lowlands to high Andes (see Amazon biome vs Amazon basin in figure 2). It also assumes no thermal tolerance changes of trees, or internal migrations/portfolio effects leading to increased resilience, etc., of the trees that make up the forest, even though those could be important sources of climate resilience (though almost certainly not against fire).

In Box 1, we now included the reanalyses of tipping points and bistability ranges for MCWD and dry season length (previously as Extended Data Fig. 1), together with mean annual rainfall, showing how these different dimensions of the rainfall regime contribute to shape forest resilience. Together, they show how bistability is related to rainfall conditions, but indeed, alternative tree cover states could be in different contexts related to geography or the local environmental (e.g. soil or altitude). We consider rainfall only as a proxy for water availability, and other factors will play a role, as we explain in Box 1. The Amazon is a highly complex system, and the type of simplification shown in Box 1 is only a way to facilitate our understanding of how water shapes forest resilience. For instance, forests coexist with savannas in peripheral parts of the Amazon where climate is drier, such as the Cerrado savannas along the south or the Gran Savana in the north. Often, these regions of coexistence are relatively drier, when compared with regions where savannas are rare, but factors such as altitude could also contribute to increase savanna abundance. At finer scales, savannas and forests may be distributed in contrasting environments. For example, white-sand savannas are found scattered within the forest biome, mostly on seasonally flooded areas. We discuss these nuances in the subsection ‘Environmental heterogeneity’.

We also discuss how aspects related to regional species pools could affect forest resilience to drought (Feeley et al. 2012; Esquivel-Muelbert et al. 2018) in section ‘Biodiversity’ (lines 380-387). Nonetheless, in parts of the Amazon that cross tipping points in rainfall conditions (Box 1), even if drought-tolerant tree species could migrate, forests would likely collapse into an open vegetation state.

We mention thermal tolerances of trees and associated risks in the sections ‘Regional climatic variability’ (lines 179-185) and introductory paragraph of ‘Five potential tipping points’ (line 416). Although observations of historical temperature changes in the past 40 years indicate widespread warming across the basin (Extended Data Fig. 1), these temperatures are still far from reaching the thermal tolerances of most trees. Hence, we discuss that heat stress could be a factor of concern in the future, but focused on water stress that is already causing impacts on the ecosystem.

Also, early on the paper needs a better discussion of safe operating spaces and the goal of the paper in setting them. This doesn’t have to be a lot longer, but it needs to clearly define safe operating space and then talk about the axes that will be discussed. In general, there is jargon in the paper that could be reduced for precision and understanding. For example, biocultural diversity conflates two very different

mechanisms of forest resilience and points of engagement for policy, as the authors discuss. Conflating them in one term hides more than it reveals.

Indeed, that is an important point. To better define the concept of ‘safe operating space’ in the manuscript, now say in the end of the Introduction “Inspired by the framework of ‘planetary boundaries’, we identify climatic and land-use boundaries that reveal a safe operating space for the Amazon forest system in the Anthropocene ^{15,16}.”

Although at this point the axes have not been defined (only later in section ‘Five potential tipping points’), the sentence above has the purpose of clarifying the point raised by the reviewer.

‘Biocultural diversity’ is now divided into two sections ‘biodiversity’ and ‘Indigenous peoples and local communities’.

The tipping point potential index and discussion needs to be made more transparent, and I have a larger section below on this. It is an interesting approach, though not without controversy.

Thank you for these suggestions. We have tried to improve the ‘Tipping potential’ (now ‘Transition potential’) based on your comments below.

Finally, given the importance of deforestation and fire in driving tipping points in Amazonia, I thought the discussion of these factors and their incorporation into the tipping index and prediction was not as thought out or well presented as it could be. The recent Amazon report (<https://amazon.org.br/imprensa/estradas-cortam-ou-se-aproximam-de-41-da-area-de-floresta-na-amazonia-mostra-mapeamento-inedito>) reinforces this. Some kinds of prediction for future disturbance components or drivers like roads and ignition, though fraught, is important, as by 2050 one does not expect them to be the same as today, even if there was not external climatic reinforcement of them. It is possible that we will deforest the Amazon purely for economic motive. (On this line of reasoning, the restoration activities and potential for the SE Amazon is worth expanding as well.). To turn the point around, fire and deforestation are key drivers of the tipping point and decreased resilience, and they should be given at least as deep a treatment as the climatological aspects.

Thank you for these suggestions. The ‘transition potential’ index was thought to precisely incorporate this combination of climatological and land-use drivers. We agree that road network is likely a better predictor of future deforestation and fires (Laurance et al. 2002; Soares-Filho et al. 2006; Kumar et al. 2014). To calculate the index, we added values for different disturbances overlaying in each pixel (location), and now we replaced both deforestation and fire layers by the road network, adding value 1 to all pixels within 10 km from roads, thus increasing ‘transition potential’ (see Fig. 1f). We also included Indigenous Lands and Protected Areas as well-known buffers against deforestation and fires (e.g. Nepstad et al. 2006), by subtracting value 1 from pixels inside these areas, thus reducing ‘tipping potential’ and compensating for the disturbing effects associated with roads (in Fig. 1f).

We discuss the importance of restoration activities in section ‘Implications for governance’ (line 626).

Line-by-line comments:

L53. Does biocultural diversity really enhance adaptability? The Amazon was a rain forest system for ~45 my. Did the addition of human really make the system more adaptable for the last 10-20kyr compared to the previous 44.98 my? I'd rephrase to talk about the changes in human land use and biodiversity rather than conflate them.

The reviewer correctly argues that the positive effects of Indigenous peoples and local communities (IPLCs) apply only to the period when humans were present. We tried to clarify this point and the mechanisms by which humans may increase forest resilience along the section 'Indigenous peoples and local communities'. Both biological and cultural diversity (through diverse practices and ecological knowledge systems of IPLCs) can enhance Amazon forest adaptability to global changes and the resilience of ecosystem services through mechanisms that are well described in the extensive literature within the frameworks of 'social-ecological systems' and 'complex adaptive systems' (e.g. Biggs et al. 2012), but also in more specific studies of the Amazon forest (e.g. Sakschewski et al. 2016; de Souza et al. 2019).

L75. Explain acronym as it does not follow from description.

We removed the acronym.

L83. What exactly is meant by safe operating space in this context? Avoid jargon and say precisely. Also, generally, be careful about over-use of jargon.

We tried to clarify: "Inspired by the framework of 'planetary boundaries', we identify climatic and land-use boundaries that reveal a safe operating space for the Amazon forest system in the Anthropocene^{15,16}."

L150. What is meant by 'sedimentation dynamics' here? Be precise.

We clarified: "sediment deposition on ancient floodplains".

L158. Need a section in here about increased CO2 effects. In the past it was cooler during droughts, but CO2 was also about ½ current values for much of the time, creating the potential for high water stress at similar levels of drought. Corlett's (2016) suggestion that current forest drought responses have already been buffered by high CO2, meaning that the past may have been even more severe. It is an open question at this point, but one that needs to be presented when thinking about the past. On further reading there is a section deeper in the paper that treats this, but should be listed up front.

This is a good point. We now say: "Although past drought periods were usually associated with much lower atmospheric CO2 concentrations, which may have reduced water-use efficiency of trees⁴², they also coincided with cooler temperatures^{31,32}, which likely reduced water demand by trees."

Box 1. This is a beautiful figure of the bistability of tree cover in the Amazon. It

suggests a tipping point, but what about soils/underlying geological effects? Shallow, nutrient poor, and/or sandy soils can amplify the effects of drought, or produce the appearance of bistability, but when a second axis is added, it becomes just stability. Could this be accounted for here? Also, 1500 mm of relatively stable rainfall, either on an annual or interannual timescale, can be very different for trees from a system with high intra- and interannual variability. Does this (rainfall variability) explain any of the bistability?

The reviewer correctly argues that environmental heterogeneity may reduce hysteresis, causing the bistability range to shrink or disappear. This is probably more important for factors related to groundwater access, such as topography (e.g. Oliveira et al. 2019) or rooting depth (Sakschewski et al. 2021). In the section ‘Environmental heterogeneity’, we explain how heterogeneity can affect the risk of tipping points, including differences between forests under high and low rainfall variability (lines 318-328). In our review, we did not test for bistability using many variables together (which would be a novel approach), but we looked at rainfall variability separately and found bistability in tree cover related to MCWD and in DSL. We have now combined these two plots with Box 1 (they were previously in Extended Data Fig. 1), together with mean annual precipitation, to show in the main text how tipping points and bistability can also be assessed in relation to rainfall seasonality. We agree that future studies should test how various factors together affect bistability.

More than that, Flores et al. 2017 indicates that much of the bistability is due to floodplain forests—if you separate the forests into varzea and terra firme, the bistability in terra firme largely disappears, being mainly in low-rainfall floodplain systems. Worth addressing here.

Indeed, in the 2017 paper, we found that terra-firme areas have basically a forest stable state, whereas floodplains are bistable within a certain range of mean annual rainfall conditions. The difference with this one is because in that paper we analysed only the area of the forest biome, excluding surrounding areas of savanna biome and including only patches of savanna distributed within the forest biome. This approach revealed the role of floodplains in shaping forest resilience within the biome. Now, we use a broader area (the whole Amazon basin area), including savannas of the northern Cerrado and of the Gran Savanna in Venezuela. This approach (similar to Hirota et al. 2011; Staver et al. 2011) is more suitable to assess how rainfall conditions affect tree cover bistability.

Also, in its presentation, the figure is highly misleading—much of the Amazon has been deforested, and that deforestation is concentrated in some of the most interesting places on this map (non-coincidentally). The figure should be modified to show deforested areas and road networks. This will also help with the discussion of anthropogenic fires and tipping-point potential.

To produce the scatter plots in Box 1 we certainly excluded deforestation until 2020. Based on the empirical estimate of tipping points using mean annual rainfall, we then projected tree cover stability classes on the map (Box 1b). The map does not indicate deforested areas precisely to show stability classes for these areas, which could be

indicate for instance reductions in rainfall related to deforestation. We therefore disagree that Fig 1a is misleading.

We agree, however, that showing deforestation and roads in the map of Box 1 would help readers visualise how disturbances and bistability interact. But this is the purpose of Fig. 1, where we show human and climatic disturbances across the Amazon. However, the purpose of Box 1 is mainly to illustrate the concept of tipping point for general readers using the example of rainfall conditions. We expect that from the examples shown in Box 1, empirical researchers (such as ecologists) will improve their understanding of the tipping dynamics, and consequently of our manuscript.

L218. Okay, discussion of things raised above here. Needs to be hinted at in the box.

Please see previous response.

L~230. Feeley et al. 2012 GCB had a discussion of this effect + CO₂ that should be cited somewhere in here in addition to Araujo et al. 2021—former covers the broader Amazon.

Thank you for suggestion. We incorporated the findings of Feeley et al. (2012) in the section ‘Regional climatic variability’ (line 184).

L238. Flores et al. 2017 generated a small amount of controversy, but, again, how do the current results compare with those, particularly given the effects of driest quarter, terra firme vs varzea, etc. And, if you were to remove the giant and interesting areas in the upper Madeira (the Bolivian savannas), what inference would we draw from the remaining areas? How much does that one area influence our conclusions on bistability?

We understand that the text was not sufficiently clear on these issues. As mentioned three responses above, now we use a broader area that includes large parts of savanna biomes surrounding the Amazon forest (in Box 1). In Flores et al. (2017) the focus was understanding the role of seasonal flooding as a factor increasing savanna resilience within the forest biome. Indeed, bimodality in tree cover practically disappeared if we removed the large savanna in Bolivia, but because these floodplain savannas are in one of the driest parts of the Amazon biome, we could not separate the roles of flooding and drought. Here (in Box 1), the focus is the role of rainfall conditions (as a proxy for water stress), and this is why we included a broader area.

L~275. Much of the fire activity in the Amazon, and propagation of fire into natural systems, comes from agricultural land conversion and fire escape. It might be worth stating this explicitly here.

We explain this in the subsection ‘Disturbance regimes’ (lines 218-224).

“Within the remaining forest area, 38 % has been degraded by extreme droughts and human disturbances, such as logging, edge effects and understory fires^{1,59,61}. Road networks (Fig. 1d) facilitate illegal activities throughout the core of the Amazon forest, promoting deforestation and fires^{62–65}. Fire can be a severe disturbance^{27,66}, usually associated with deforestation areas from where it may escape into standing forests in drier years^{67–69}. New fire regimes are burning larger forest areas⁷⁰, emitting more

carbon to the atmosphere^{71,72} and forcing IPLCs to adapt⁷³.”

L308. “liana-infested” might be toned down—lianas are as much a part of Amazonia as the trees—it is just when they pass a tipping point that they become problematic. Liana dominated? Liana hyperdominance?

We changed it to ‘liana forests’.

L311. The large extent of bamboo dominated forest (180K km² fide Nelson et al.) is not degraded forest, but a natural feature on the landscape with mammal, bird, reptile, etc., taxa found only there, and its own internal dynamics. It does escape and, like lianas, shifts degraded lands into alternate states, but the presence of bamboo per se is decidedly not pathological in Amazonia, even vast expanses of it.

We agree that these forests are important native ecosystems. The problem is when they expand continuously, replacing surrounding burnt forests. We now mention this explicitly (lines 533-536).

L347. Another feedback to include here is just human use of the landscape. It opens due to fire but then is grazed or used for agriculture, which is a very strong feedback. I think it is important to distinguish what would happen to the landscape if humans were to disappear tomorrow vs if they are left to business as usual as it informs the scale of possible policy interventions.

Indeed, relevant to raise such implications for different human-related scenarios. Feedbacks related to land use are described in detail in the section ‘Three alternative ecosystem trajectories’, but for each trajectory they involve different components.

Humans disappearing from the system seems unrealistic. Indigenous peoples and local communities occupy or at least use even the remotest forests of the Amazon. If we consider farmers, they are also part of the Amazonian society and their collaboration will be necessary to end deforestation and promote restoration.

L366. I think you can say 55+ million, not 45 thousand. Jaramillo and others have shown that you have rain forest assemblages by the mid-Paleocene, and higher palynological diversity in the South America for rain forest taxa in the Eocene. Obviously no Amazonia per se without the Amazon river, but, still, 10s of millions of years. The ecological interactions that keep rain forest rain forest are enduring and fundamental consequences of biodiversity.

Both papers by Jaramillo et al (2006, 2010) use pollen data from the periphery (northwest) of the basin, thus not representing the Amazon biome. The study by Wang et al. (2017, Nature) uses data from both the eastern and western Amazon and thus provides a broader evidence that the Amazon persisted dominated by forest in the past 45 kyr. Nonetheless, we agree with the reviewer’s critique and now we also mention in the section ‘Past dynamics’ (previously ‘Paleorecords’) the date of 5 million years ago as the approximate time when it is safer to say that the Amazon was dominated by tropical forests (Hoorn et al. 2010).

Figure 2. (a) Color ramp might be reversed, with reddest being hottest. Also, what are the units on the modeled slopes? Text says year, but a low of 0.1/yr still gives almost 4

deg C warming, and a high would give something like 23C! (b) Most of the savanna to bistable is in the high Andes where there now is grassland. Is this worth a few words? Also, it would seem like high elevation fire regimes would prevent this, at least in areas with high enough productivity to carry grass fire. (c) Hard to pick out red for deforestation—different color? (d) can this be stated as a probability?

Thank you for the useful points to improve/clarify Figure 2.

(a,c) We improved the figure colours based on these suggestions.

(a) We also checked the regressions that resulted in the slopes and corrected the map (now slopes range from -0.01 to +0.06).

(b) We now excluded areas outside the forest biome, such as the big bistable area in the Andes, to discuss only changes within the biome.

(d) We produced this index suggesting areas more prone to collapse, but unfortunately we cannot infer probability.

L406. Intensity  intensify

Done.

L429. Great paragraph on CO₂ Fertilization—you might make the summary point up front that the potential effects are myriad and poorly understood and poorly constrained, and changeable depending on spatial and temporal scale of inference (what is “good” at the leaf or plant scale may have a negative consequence at the forest scale).

Thank you. Following the standard structure, we added a summary sentence at the end of the section. “In sum, due to multiple interacting factors, potential responses of Amazonian forests to CO₂ fertilization are still poorly understood. Forest responses are also dependent on scale, with resilience possibly increasing at the local scale on the relatively more fertile soils, but decreasing at the regional scale due to reduced atmospheric moisture flow.”

L545. Even though the climate may be mitigated through increasing rainfall, etc. (if that continues), it would be hard to make a statement about forest resilience increasing given the high rates of deforestation, ag conversion, and road building and the fires that they bring. It raises a larger point about resilience and scale. At the tree scale resilience may increase in W Amazonia, but at larger spatial scales anthropogenic clearing and fire could/would swamp that effect outside of large protected areas.

We now deleted this former text, since this has been properly explained in the sections ‘Regional Climatic variability’ and ‘Disturbance regimes’.

L549. What are the “ancient sources of connectivity”? And, more generally, it is worth talking about fragmentation and land use specifically. There is no mystery about their effects. The bulk of Eastern North America was cleared of forests within a century, and with hand tools.

By ‘ancient sources of connectivity’ we referred to the ones that existed in the past millennia, promoting forest resilience, such as feedbacks between the forest and rainfall, soil fertility and IPLCs. We now deleted this part, which was already explained in section ‘Ecosystem adaptability’.

In section ‘Implications for governance’, we now highlight the importance of various disturbance regimes combined as a mechanism reducing forest resilience, including those related to fragmentation and land use (lines 609-613).

“Forest resilience is changing across the Amazon as disturbance regimes intensify and become pervasive (Fig. 1). Although most recent models agree that a large-scale collapse of the Amazon forest is unlikely within the 21st-Century², our findings suggest that interactions and synergisms among different disturbances (e.g. frequent extreme hot-droughts and forest fires) could trigger unexpected tipping behaviour even in remote and central parts of the system^{15,16}.”

L550. Here and earlier. Biocultural should be split into “biological” and “cultural” diversity like was done in the very nice section starting L499. The mechanisms for their action are different, and mashing them into one word hides the importance of each.

We separated them into ‘biodiversity’ and ‘Indigenous peoples and local communities’.

L558. Reference out of format—should be Hecht 2011?

We corrected the format.

L567. Citation format.

We deleted the link.

L568. If GHG levels returned to preindustrial levels would there be no risk of tipping-points in the Amazonian periphery? Human fire after agricultural and other clearing is a powerful force, particularly when coupled with grazing after several rotations of crops. What about the possibility that fragmentation and shifting population/ag practices can create a runaway process, unabated by climate change.

We agree with the reviewer. As we explained (four points) above, the impacts of various disturbance regimes are accumulating across the Amazon, even in central and remote parts of the biome (see Fig. 1f), increasing the possibility that stable forest areas may also transition into an alternative stable state (of low tree cover or simply into a degraded forest). This could indeed increase the risk of a runaway process.

L571. Are -- > is

Done.

L1097. Tipping Potential. The time frame for tipping potential is 2050, but some of the indicators makes it seem like the latter, but many are based on past (ending 2020) measurements. An argument could be made that the most recent decade is the best indicator we have of what will happen in the next 30 years, but there are more nuanced

ways to look at things like deforestation and fire potential using agricultural and road network trends and predictions. Are those predictions specifically believable—no—but the idea that tipping potential 30 years from now is limited to the current ag clearing and roads doesn't seem to be a valid assumption—there will be much more deforestation. As well, the structure of the tipping potential and its components raises a few questions. First, temperature is taken as dry season temperature, and as the slope of simple linear regressions across the measured time frame. Dry season temperature in the southern Amazon is the coolest time of the year, save for a few months in the late dry season, so the idea that even moderate increases in that per se would be inimical to trees as a growth form is likely not true. It would seem like the interaction with both water stress and clearing practices would be the key harm. Bolivian forests, for example, could take a lot of dry season warming and never come close to the temperatures of a typical central Amazonian forest—the Amazon is huge, and the temperature variability is large. Second, the simple linear extrapolation of rainfall trends to 2050 seems simplistic, particularly when the authors earlier in this paper and other literature have shown the importance of MCWD. Would it make more sense to combine the two and look at actual dry episode predictions? One could think that the linear trends would not be as important as the probability of episodic events, like the megadroughts of 2005 and 2010. The use of deforestation from 2012-2020 as an index of tipping potential through 2050, and the use of current observed fires, seems almost certain to be an underestimate of the impacts of these factors. Road networks will expand, the agricultural network will expand, and ignitions will expand, even if climate were to remain static. At least why not take the same approach of using simple linear extrapolations of road/deforestation/ag activity in any cell through 2050. That would be consistent, at least.

We appreciate these reflections and suggestions, and have now adapted our 'transition potential' (previously 'tipping potential') index to include: (1) the slopes of dry season temperature change, (2) ecosystem stability classes estimated for year 2050 based on mean annual rainfall, (3) accumulated impacts from repeated extreme drought events, based on MCWD time series, (4) road networks (within 10 km buffer), and (5) protected areas and indigenous territories as factor of increased governance. This way we incorporate MCWD in the form of extreme droughts events. Instead of using projections of deforestation and fires, we use current road networks, which are probably good predictors of these land-use disturbances in the future.

The comment about dry season being the coldest time of the year is correct only for a small part of the Amazon in the south of the basin where dry season mean temperature is below 25 °C (see new Extended Data Fig. 1), even though maximum temperatures in those areas are mostly above 30 °C during this season. Moreover, both mean and maximum dry season temperatures have been significantly rising since the 1980s, thus increasing water stress across the basin. Therefore, we consider that for most of the Amazon forest, it is precisely during the dry season when water stress is higher (due to a combination of water deficit and more evaporative demand), and interacting with deforestation and fires.

More generally, the authors need to put more transparent justification for the development of these metrics. The paper reviews and discusses a large literature based

on model predictions and what they reveal about tipping points. (Several of the authors are authors on these other papers.) The interactions and nonlinearities therein among the factors that lead to forest tipping points are difficult, so the argument could be made for a very simple metric based on the things that factor into the authors' conceptual diagram rather than a more complex model, but that in itself is an interesting statement about how the scientific should use and rely on information in policy, and should be explicitly stated. And, it could be clarified further in the main text that the index is really just the sum of the intensity of five different factors, and that the change from 1-2 may not be equal to the change in 3-4, etc.

Thank you for this suggestion. We now say in section 'Disturbance regimes' (lines 241-243): "By exploring only these factors affecting forest resilience and simplifying the enormous Amazonian complexity, we here aimed to produce a simple and comprehensive map that can be useful for guiding future governance."

We also added more details about how we produced the index in the legend of new Fig. 1f. and in the Supplementary Information section 'Transition potential'.

Referee #2 (Remarks to the Author):

Review of Flores et al. Tipping points in the Amazon forest system

Overall, the manuscript attempts to assess whether and how much of the Amazon forest may pass a threshold (tipping point) and become a more open system, which would result in important impacts on global and regional climate, local people, and biodiversity.

This is a difficult manuscript to review – it has a bit of new analysis, I think, and the rest is a review. I think it probably works best as a Perspective/Review as there is not enough that is new to call it original research.

We agree with this evaluation that our manuscript is a review with additional reanalyses based on previous studies. We aimed at gathering and combining interdisciplinary information to paint a comprehensive and up to date and more precise picture of the state and tipping points of the Amazon.

I think the new work is estimating the three states of Amazon vegetation related to water availability (in Fig. 1, and ED fig.), some modelling later on (CMIP6 runs), and Fig. 2d, seems to be a new attempt at quantifying where forests may change to more open systems. But in all cases it is not 100% clear what is a new analysis and what is previously published.

We understand that the structure may have raised doubts on what is new or not. We reorganised and restructured parts of the manuscript and hope that it is now clear that the manuscript is all based on previously published work, with a few reanalyses only to add more precision and integration to the results.

The material is written in a series of discrete sections, without much integration across sections. That makes it hard to get clear messages, and leads to some duplication of material. The lack of clear messages starts with the abstract, which contains no quantification or even partial answer to the key question posed.

Thank you for pointing that out. Our idea was to present the manuscript in sections that connect with each other linearly, building the basis for understanding Amazonian tipping points. Their messages are ordered as follows: (1) we explain the main concepts, (2) we review the evidence of past dynamics from paleorecords, (3) we explain how resilience is changing due to various mechanisms, which increases the potential for tipping points, (4) we present the tipping points for which there is support, (5) we present the most plausible ecosystem trajectories following tipping points, (5) we propose topics for future research and ways to improve Amazonian governance.

Taking into account the reviewer point, we reworked the text to avoid repetition of content across sections.

In the Abstract, we now included more quantitative results (copied below).

“The possibility that the Amazon forest could soon reach a tipping point of large-scale collapse has raised global concern. Here we address this issue by gathering the state-of-the-art knowledge about the various mechanisms that shape forest resilience across this iconic system. For at least 5 million years, most of the Amazon forest remained resilient to climatic variability, but now, the region is increasingly exposed to warming temperatures, extreme droughts, deforestation and fires, even in central and remote parts of the biome. Long existing feedbacks between the forest and environmental conditions are being replaced by novel ones that modify ecosystem dynamics, reducing adaptability and systemic resilience. We review and discuss existing evidence for five major drivers of water stress on the forest, their potential tipping points and boundaries that define a safe operating space for the Amazon system. We estimate that by 2050, 10 - 47 % of Amazonian forests could transition into an alternative ecosystem state due to synergistic disturbance regimes. For forests that transition, we identify the three most plausible ecosystem trajectories involving different feedbacks and environmental conditions. We discuss how the inherent complexity of the Amazon system adds uncertainty about future dynamics, but also reveals opportunities for action. Keeping the Amazon forest resilient in the Anthropocene will depend on local efforts to end deforestation, reduce land-use disturbances and expand forest restoration, as well as on global efforts to stop greenhouse-gas emissions.”

The topic is very important, and there is lots that I think is useful in the manuscript, but it feels to me that it needs an overhaul in structure and some refinement to be really useful to people.

Thank you for pointing this out. We now have revised the logic and structure of the abstract also according to the changes of the whole manuscript. We hope that our explanation above helped to clarify our intentions.

I have one big and perhaps fundamental disagreement with the authors, as I think there is only actually one threshold, i.e. one tipping point. This threshold is a lack of

sufficient water for the maintenance of a closed canopy tropical forest. Three of the five tipping points are annual rainfall, dry season length and water deficit, which are all aspects of the same thing – they are all proxies for a lack of sufficient water for maintenance of the forest. They are also all very highly correlated.

We agree with this interpretation that all five tipping points are directly or indirectly related to water stress. In the revised version, we now call them ‘drivers of water stress’ throughout the text. We also combined Table 1 with a diagram of interactions among these drivers, forming a new Fig. 3. ‘Drivers of water stress’ increase forest loss, which feeds back into the drivers. Although they might be correlated, this depends on the spatial and temporal scales. For instance, some drivers may change faster than others and in different parts of the Amazon. Therefore, we think it is important to mention them separately, because monitoring changes in these drivers is crucial for orienting mitigation and adaptation actions. We now say in the introductory paragraph of the ‘Five potential tipping points’ section:

“Environmental conditions are changing in the Amazon region (Fig. 1), yet only a few studies indicate critical threshold values (‘tipping points’) in these conditions that could result in abrupt ecosystem transition¹³. In this review, we identify five potential tipping points in the Amazon forest system (Fig. 3a), each one related to a threshold value of a different driver of water stress on the vegetation: (1) global warming, (2) annual rainfall, (3) rainfall seasonality intensity, (4) dry season length, and (5) accumulated deforestation. In different ways, these five drivers affect water stress, and their intensification may cause widespread tree mortality, for instance because of hydraulic failure¹⁴⁰ or fires⁶⁸. Water stress controls vegetation productivity and resilience globally^{141–144}, but other stressors may also affect the forest, such as heat stress^{53–55}. In the coming decades, these five drivers could change at different rates, with some approaching a tipping point faster than others, thus monitoring them separately can provide critical information to guide mitigation and adaptation strategies for the region. Moreover, we show how the five drivers may interact and form positive feedbacks (Fig. 3b), which could accelerate a tipping behaviour²³. Hence, to avoid unexpected tipping points, we suggest safe boundaries of these drivers^{14–16}, which define a safe operating space for the Amazon forest.”

Then, I would argue, the other tipping point, forest loss, is just one of two ways that lack of sufficient water can arrive, as forest loss leads to lower evapotranspiration and water recycling and so lower rainfall downwind. The second way of lacking sufficient water is that the climate changes and leads to dry enough conditions so that a forest cannot persist. I think everything beyond that is a modifier of the forest’s ability to resist the change (e.g. disturbance opens the canopy makes droughts more intense) or moves the threshold (e.g. rising CO₂ increases water use efficiency, and so forest can tolerate drier conditions, all else being equal).

In our previous response, we explain how we also considered forest loss and the tipping point in accumulated deforestation as another ‘driver of water stress’. We thank the reviewer for highlighting the importance of this fundamental mechanism.

Making my argument the other way round, to follow the logical of the authors, we could invent even more tipping points related to different aspects of temperature increases

(mean annual temp, or daily max temp), or differing metrics of forest loss. We would end up with huge number of tipping points! I don't think that's scientifically correct, nor very helpful.

These five potential tipping points are the ones for which we found evidence in the literature. We disagree with the standpoint that only because all the tipping points identified in this study are more or less related to water availability or water stress it is not worthwhile listing the different indicators and tipping point values. We agree that the final tree individual based stressor is water stress. But there are many ways, mechanisms and reasons to create water stress. In a complex tipping landscape there is no single hill. But it is crucial for public and policy relevant communication to break this seemingly overwhelming logic cascade down into graspable numbers. We now underline the crucial role of water stress for all drivers and their tipping thresholds.

There is also a theoretical pure temperature tipping point, but I don't think there is much evidence that under hot wet conditions major die-back is likely.

We agree that a temperature tipping point (which we interpret as the breakpoint for thermal leaf stress, according to Tiwari et al. 2020; Araújo et al 2021; Docherty et al. 2023) is currently not so important as the water-related tipping points because we are still far from it. Most trees seem to tolerate temperatures up to 48 - 50 °C, while dry-season maximum temperatures across the Amazon are reaching 40 °C (much cooler than their tolerances), even though some regions are warming. We briefly discuss the evidence about tree thermal tolerances in sections 'Regional climatic variability' (lines 182-184) and 'Five potential tipping points' (line 417).

Lastly, on the major points, some coverage of the papers that show something else, not a tipping point/no tipping point are needed, e.g.

Levine et al 2016. Ecosystem heterogeneity determines the ecological resilience of the Amazon to climate change. PNAS.

In section 'Five potential tipping points', we discuss papers that did not find tipping point (or dieback) (for instance Chai et al. 2021), and papers that show a broad range in tipping point values. In section 'Environmental heterogeneity', we discuss results from Levine et al. 2016 explaining how this and other papers found that environmental heterogeneity can reduce the risk of tipping points.

There can be major impacts on the forest, without a tipping point per se, as Levine et al. 2016 appear to show. Similarly, mention of the huge impacts for carbon, climate and biodiversity of dramatic changes that fall short of flipping to an open system ought to be mentioned. It is not all or nothing, which some readers could come away with thinking on reading the current manuscript.

In the section 'Three alternative ecosystem trajectories' (see 'Degraded forests' in line 511) we explain in detail how impacts on the ecosystem can be more subtle, involving changes in species composition and functioning, due to different types of feedbacks.

One other cross-cutting issue is the treatment of soils. There are essential in terms of modulating the threshold, and shown nicely by the examples in the manuscript of white

sands. But soils need more careful inclusion elsewhere, particularly soil texture, as a proxy for soil water holding capacity.

Thank you for this suggestion. We do not include soils explicitly in our reanalyses, because this would be new research. Nonetheless, we agree that soils are very important, and this is why we discuss their roles in shaping forest resilience in several sections of the manuscript: (1) they are part of the feedbacks in the ‘Three alternative ecosystem trajectories’, (2) they are discussed as interacting factors with the ‘CO₂-fertilization’, (3) they contribute to increase ‘Environmental heterogeneity’.

Overall, I wonder if it would be clearer to first examine if there is a tipping point in the Amazon system, and review the evidence. Then with that established, assess where the threshold is for the lack of water, and whether climate change or forest loss can breach it, and if so over how much of the basin. Then deal with the key uncertainties in assessing those thresholds – which can really guide new research -- then finally have a section on what to do and the ‘safe operating space’. Feel free to ignore this specific structure, others may work, but I think the current structure isn’t working that well.

We thank the reviewer for this suggestion. We did not follow exactly the order suggested by the reviewer, but we reorganized our storyline, including new sections and reordering of content. Because the tipping points come from a literature review, we first present the evidence from paleorecords of ecosystem dynamics in response to climatic variability (section ‘Past dynamics’). Then, we discuss the evidence and uncertainties related to key external conditions affecting forest dynamics today and how they are changing (sections, ‘Regional climatic variability’ and ‘Disturbance regimes’). We also discuss uncertainties in sections ‘Local versus systemic collapse’ and ‘Ecosystem adaptability’. In the section ‘Five potential tipping points’ we focus on the evidence available supporting the tipping points identified, but also discuss uncertainties and confidence levels (Fig. 3). We now also highlight the role of water stress as a key mechanism for the risk of tipping points. As suggested by the reviewer, we conclude with a section on ‘Implications for governance’, where we suggest actions to strengthen Amazonian resilience based on our findings.

I have disagreements, but do think an exploration of the Amazon tipping point would be very useful, as it is such a critical topic, and some disagreements among experts are fine.

We appreciate these constructive criticisms. The reviewer has convinced us to adapt our tipping points section to consider the fundamental role of water. Other suggestions also helped us to clarify the story for different types of readers.

Section by Section comments.

Introduction

Fig. 1. The authors need to present the most robust choice of rainfall, dry season length and water deficit, and not one arbitrarily. Your data can help. Perhaps fitting a logistic regression to each, and the proxy (rainfall, DSL, MWD) with the steepest slope is the

one that best reflects the tipping point.

The choice for using annual rainfall (in the previous version) was based on previous influential work (e.g. Hirota et al. 2011 Science; Staver et al. 2011 Science; Staal et al. 2020 Nature Comms) showing the importance of annual rainfall as a major determinant of water availability for plants across the tropics. Nonetheless, following the reviewer's suggestion of including the other rainfall variables, we also tested for bistability for MCWD and DSL using the same approach with potential analysis (Livina et al. 2010) and included these two plots in Box 1, together with the previous one with mean annual rainfall. This way, instead of having to choose only one variable, we show all three rainfall variables, their tipping points and bistabilities.

Supporting Evidence.

I am not sure why there is a section called this. Supporting what?

We deleted this section and moved their content to the sections 'Past dynamics', 'Regional climatic variability' and 'Disturbance regimes', which improved the flow and structure.

Paleorecords

To me they show forest resistance and/or resilience in the face of major environmental changes (arrival of humans, glacial to interglacial conditions). But you can see impact of disturbance – some expansion of savanna associated with human-induced fires. This isn't what is currently written.

We agree that these are the main findings. We tried to clarify these points in the conclusion of this section (now called 'Past dynamics').

“Paleorecords suggest that a large-scale Amazon forest collapse did not occur within the past 5 million years. Nonetheless, paleorecords indicate that savannas expanded at local scales, particularly along the more seasonal peripheral regions, where human-ignited fires were frequent^{37–39}. Patches of white-sand savanna also expanded inside the forest biome due to geomorphological dynamics and fires^{40,41}. Although past drought periods were usually associated with much lower atmospheric CO₂ concentrations, which may have reduced water-use efficiency of trees⁴², they also coincided with cooler temperatures^{31,32}, which likely reduced water demand by trees. Past drier climatic conditions were therefore very different from the current climatic conditions, in which observed warming trends will likely exacerbate drought impacts on the forest by exposing trees to unprecedented levels of water stress^{1,43,44}.”

Line 143, that humans were 'managing' the Amazon 6,000 yrs ago is highly debatable. Their impact was likely v. modest. Suggest rephrasing.

We now say: "... when humans were already present in the landscape." (line 132)

Modelling

Not sure why this comes before observations (or between paleo and contemporary observations).

The 'potential' for die-back is not what is interesting, but whether that would happen in the near future.

We moved the text from the previous ‘Modelling’ section to the sections ‘Regional climatic variability’ and ‘Global warming’. In the ‘Introduction’, we say: “Large parts of the Amazon forest are projected to experience mass mortality events due to various climate and land-use disturbances in the coming decades ⁶⁻⁸.” (line 65)

Observations

Overall, this section is weak, as it is missing data from remote sensing, including the essential work of Gatti et al. 2021 Nature, using CO₂ collected from aircraft. Limited data cited from satellites, no mention of the inventory plots data, as relevant data from Brienen et al. 2015, Hubau et al 2020 Nature, nor any disaggregation of the forest responses in intact, logged and burnt forests.

We deleted this section and most its content was moved to section ‘Disturbance regimes’, where in the second paragraph, we incorporated the references mentioned by the reviewer and other important references (lines 225-234). See text below:

“Currently, 86 % of the Amazon forest biome may be in a stable forest state (Box 1b), but some parts are showing signs of fragility. A basin-wide satellite analysis suggests that 75 % of the Amazon forest has been losing resilience since the early 2000s, potentially approaching a tipping point ⁴⁵. Field evidence from monitoring forest sites across the Amazon shows that tree mortality rates are increasing in most sites, reducing carbon storage capacity ^{74,75}, while favouring the recruitment of drought-affiliated species ⁷⁶. Carbon flux measurements above the Amazon reveal how south-eastern forests are already emitting more carbon to the atmosphere than they absorb ⁴⁴. Altogether, evidence suggests that large parts of the Amazon forest are gradually shifting their functioning and composition ^{75,76}, while turning into sources of atmospheric carbon ⁴⁴.”

Five tipping points

As above I do not think there are 5 tipping points.

Please, see our response to the other comment. We agree that all five drivers that may cause an Amazon tipping are somehow related to water stress. Listing these tipping points is highly valuable for science but more importantly for public and policy communication to enhance the transformation needed to protect Amazonia.

Global temperature rise of 2C

As currently written, this tipping point is really as much part of the water-related tipping point. The reason why there is a risk of breaching a threshold is because droughts increase in frequency and intensity, and I think only secondarily about the heat.

We now included a new Fig. 3 with a diagram of interactions among the five ‘drivers of water stress’, showing how driver 1 – ‘Global warming’ affects regional Amazonian rainfall. These effects on rainfall as well as also on temperature are explained in detail in the section ‘Regional climatic variability’.

On temperature per se, there is a big literature here, but little is cited.

We now discuss three papers that we found with evidence on tree thermal tolerances and potential impacts related to heat stress (Tiwari et al. 2020; Araujo et al. 2021; Docherty et al. 2023).

Saying warming will make it “likely too warm for most Amazon trees” seems speculation, and I doubt it is true. Are you really claiming that warming will be too high for ~8,000 tree species? Needs evidence. There was a push back on the idea of ‘biotic attrition’ some years ago, see Dick et al. 2013 Neogene origins and implied warmth tolerance of Amazon tree species, Ecology and Evolution.

We agree that heat stress is currently not as concerning as water stress. We deleted this sentence and now explain briefly how heat alone may affect trees in the section ‘Regional climatic variability’.

Furthermore, if even some large group of species that continue to form a closed canopy forest can survive a few degrees higher temperatures, then there is no temperature tipping point to savanna, while all else is equal.

Agreed.

Forest Loss

This section should probably should go first as it’s the most immediate.

Thinking of changes in forest cover due to accumulated deforestation is more intuitive than changes in the other four drivers of water stress. Because the others are climatological and their dynamics occurs larger scales, we kept the tipping point related to drier 5 - ‘Accumulated deforestation’ (previously was ‘accumulated forest loss’) as the last one because it occurs at local scales, so from larger to smaller scales.

The relationship of forest loss to a tipping point, is again via water availability, and the lack of moisture recycling. It should be made clear that losing forest is another way of lowering rainfall and reaching the drought tipping point.

We agree and have now clarified this throughout the text and in a new Fig. 3.

Fig.2. The fig 2d is potentially extremely worrying and no doubt controversial, so some details need to be in the main text. I don’t think you can just sum them.

Thank you for this suggestion. We included more details in the figure caption, which is now Fig. 1. We sum the different layers of disturbance regimes because this is a simple and most intuitive way to quantify their compounding effects on forest resilience. We understand that it is a simplistic approach, but still a first step to integrate (even though linearly) all the factors available that potentially affect tipping points from our review.

Three alternative ecosystem trajectories

I didn’t understand this section. Alternatives to the tipping points being discussed? I was unclear if these were modifiers of the value of the threshold, or something additional. I didn’t understand why disturbance of a specific soil type was in a grouping with degraded forests per se. If something additional, then perhaps differing terminology would help.

Thank you for identifying the need for clarity. This section has additional content describing how forests that cross tipping points may shift into three different types of ecosystems (alternative stable states). Forests may follow these three different

trajectories, depending on the types of disturbances (e.g. including fire or not), the feedbacks and other environmental contexts, such as rainfall and topography. These details are now explained in the introductory paragraph of the section.

Forest resilience is changing

This is a really useful set of sections. However, I think it would be better to use ‘resistance’ to mean the ability of the system to withstand an environmental change/perturbation, and ‘resilience’ to mean the ability to recover from disturbance/an environmental change, as is common in ecology. For example, higher CO₂ levels likely confer resistance, as it increases water use efficiency. Fires likely decrease resilience (ability to recover) and lower resistance (to a second fire).

These definitions of resistance and resilience are frequently used among empirical ecologists, but they do not consider that alternative equilibrium states might exist in the system, such as the concept of ‘ecological resilience’ coined by Holling 1973, 1996. We now clarified in the section ‘Theory and concepts’ that we use this definition of ‘ecological resilience’. Ecological resilience also connects with the other concepts used in the manuscript, including tipping point.

We split the section ‘Forest resilience is changing’ into three relatively more specific sections (‘Global changes’, ‘Local versus systemic transition’, and ‘Ecosystem adaptability’) and brought them earlier in the manuscript to clarify why it is important to assess changes in ecosystem resilience.

It seems like each of these sections are, in terms of breaching a tipping point, modifiers, that either push the system closer to a threshold, or buffer it from reaching the threshold. I wonder if the manuscript works better if you stick to relating it all to tipping points more directly and obviously.

In section ‘Theory and concepts’ we explain (copied below) how the concepts of ‘tipping point’ and ‘resilience’ are part of the same theoretical background and understanding one helps to understand the other.

“... A ‘tipping point’ is the threshold value of an environmental stressing condition at which a little disturbance may cause an abrupt shift in the ecosystem state, accelerated by positive feedbacks ^{3,17} (see Extended Data Table 1). As ecosystems approach a tipping point, they often lose resilience while still remaining close to equilibrium ¹⁵. Hence, monitoring changes in ecosystem resilience and in key environmental conditions may allow societies to avoid or mitigate tipping points. We adopt the concept of ‘ecological resilience’ ¹⁸ (hereafter ‘resilience’), which refers to the ability of an ecosystem to persist somewhere near an equilibrium state (with similar structure, functioning and interactions), despite disturbances that may push it to an alternative equilibrium state ^{13,19}. Knowing when ecosystems have alternative equilibria is crucial because the crossing of tipping points may be irreversible for the time scales that matter to societies ¹³. ...”

The CO₂ fertilisation section needs more care and more balance, the evidence is much stronger than CO₂ ‘may’ affect trees. All models assume it; forest theory and contemporary observations are consistent with it, as are very limited experiments.

Similarly, on temperature, there is a debate, and some evidence on temperature acclimation that should be noted.

We discussed the main evidence in the literature, including papers showing that there is an effect and others showing potential reductions in tree transpiration and rainfall recycling, as well as the effects of nutrient limitations. This is a very complex mechanism that interacts with various other ecosystem processes and which is still poorly understood. For instance, some models assume unconstrained CO₂-fertilization, which could influence their capacity to simulate tipping behaviour. More research is needed for us to properly understand this effect.

In the Connectivity section, you could also cite Cooper et al. 2020 Regime shifts occur disproportionately faster in larger ecosystems. Nature Comms. As it adds a new-ish worrying development.

Thank you for indicating this reference, which is highly relevant. We incorporated their findings in the section ‘Sources of connectivity’ (lines 355-357).

Referee #3 (Remarks to the Author):

Thank you for inviting me to review paper: “Tipping Points in the Amazon Forest System” by Flores et al. This has been an enjoyable paper to assess.

It is important to state at the outset that of all the potential iconic tipping points of the climate system, concern about Amazon “dieback” arguably attracts the most concern. So any technical paper or review paper that gets us nearer to determining dangerous levels of climate change (in the context of “die-back”) is welcome.

The problem with almost all possible Earth System tipping points is that it is especially difficult to determine the level of global warming at which they will be triggered. Hence, now more than anything else, climate and ecological researchers need to map process understanding onto robust process-based model structures. Such simulation frameworks are usually Earth System Models (ESMs), but there are other illustrative dynamical system structures as well. It is in that context that this paper is reviewed – how can the findings be used to advance ESMs?

So despite the need for ESM improvement, what is noticeable is that this paper is in many ways, quite “model-free”. That makes a very refreshing change from the usual reporting of Earth System Model outputs. To say the obvious, more data-led papers provide important benchmarks for future model building. But at the same time, the authors do refer to the climate-ecology-land use couplings as “dynamical systems”, hinting that understanding can be readily mapped to numerical models.

In my view, what is missing in the paper is a paragraph (or even small section) on the

challenges of modelling, and in particular how the paper findings are expected to be influential. Would a Discussion part work that touches on the following:?

Thank you for this suggestion, we included a new section ‘Prospects for modelling Amazon dynamics’ to address the challenges of modelling this complex system and highlight how our findings may contribute to advance ESMs.

(1) We need to constrain ESMs to tell us more accurately links between future GHG levels (or mean global warming) and the climatological thresholds of Table 1. The current situation of simply regarding all models (e.g. those in Extended Data Table 2) as equally plausible is not satisfactory. Methods exist, of course, to help generate a refined multi-model estimate – Emergent Constraints being one example. Or simple observation trend checking for the contemporary period against ESMs in the CMIPx ensembles.

We fully agree that ESMs need to be ‘constrained’ to tell us more accurately links between future GHG levels and the climatological thresholds of Table 1 (now Fig. 3a). From evaluations we know that there is quite a quality range in ESMs. So ensembles should always be handled with caution. Conclusion could be based on smaller sub-sets of what is out there based on quality agreements. Emergent constraints can also help to evaluate ensembles and provide a more accurate picture as well as guide model development to focus on certain variables or mechanisms. Nevertheless this guidance is limited and the scientific community must constantly ask itself if ESMs cover or fully represent the variables and mechanisms needed to reflect or test the potential sensitivity of the Amazon forest to its main stressors. After all it remains important to further develop models towards aspects that the scientific community decides to be important. In the end, emergent constraint approaches could still miss important aspects when certain unidentified mechanisms/variables are not well covered in models, while the same have not been important in current evaluation data but might become important in the future.

Emergent constraints are used to evaluate ensemble runs in post-processing and provide a more accurate picture of a certain change of a variable like rainfall or forest cover. ECs are not directly used to tune models. ECs can help identify variables/mechanisms that determine Amazon forest cover the most and hence are most important to realistically capture changes in the system.

(2) Can land surface models be improved to also help turn some of the Confidence levels of Table 1 to “High”? From this study, what missing processes are urgent to get in to land models?

There is quite a large number of processes that should be improved in land surface models but it remains unclear if they all can help turning confidence levels up to high. Knowing improvements in confidence prior to implementation is impossible. But there are of course good candidates. In our review article, we actually want to set most of the focus on land surface model related improvements regarding the modeling outlook section as e.g. topics on how to improve the location and dynamics of the ITCZ in ESMs goes a little beyond of what we can tackle in this article (even though such aspects are important for Amazon resilience in simulations). We argue that, in order to improve confidence, one should be able to better match current and past Amazon extent

and dynamics. This is often related to acknowledging a higher resolution of environmental heterogeneity and the ability of the biosphere to adapt.

In the new section ‘Prospects for modelling Amazon dynamics’, we address these points. We also suggest improvements regarding representation of vegetation heterogeneity, key feedbacks and synergistic effects among disturbances. Models need to incorporate more of the feedbacks that matter, their strengths and scales they operate. Simple conceptual models are a reasonable start (e.g. van Nes et al, 2014 GBC) to producing valuable information that can be incorporated into ESMs.

(3) Can we get better direct land use descriptions into ESMs? And can we have factorial ESM simulations to confirm how different scales of spatial heterogeneity aggregate to affect overall dieback risks. What is needed to model the points raised in manuscript section “Environmental heterogeneity?”. To what extent can suppression of land use / deforestation alter the dangerous levels of climate change that determine the overall fate of Amazon tree cover.

This is a good, but complex question. There should at least be more tests conducted to check for the sensitivity of the Amazon tipping behaviour on land-use scenarios including e.g. tests for different spatial scales. In Ecology, landscape mosaics, disturbances, degradation, connectivity, as well as plant community dynamics and functional diversity are important topics because they affect tree survivorship/mortality probability in response to environmental changes and disturbances. As laid out in many studies, as in this review, these variables are connected to resilience (see section ‘Environmental heterogeneity’), but it remains elusive what role they play for overall Amazon resilience. For instance degrading agents, such as logging, can make a local forest more vulnerable to burn. It is plausible to assume that this coherence is important basin wide. This in return means ESMs must underestimate this effect since they fully neglect heterogeneity. We do think that land use and its spatial facets can alter the tipping point of e.g. climate. We fully acknowledge that all aspects of potential Amazon tipping are interconnected. Listing all model shortcomings though is a very long list and we decided to point out those points which we think are most important.

(4) Would the authors support recommending developing fast dynamical systems “toy box” models that can explore parameter space and in particular couplings? It would be especially good to quantify what happens if forcings change simultaneously. It is unlikely that the edge of any safe operating space will be approached by just one climate feature. If two or more edges of safe operating space are approached together, then does that potentially bring the tipping point forward? Or make the jump to any new state very fast i.e. highly nonlinear?

There are multiple reasons why using toy models can be an important addition to research. First of all these models are fast and relatively easy to further develop hence the perfect candidate for fast and adaptive screenings in accordance with the knowledge standard. There might also be situations which make toy models and emulators necessary in terms of testing parameter space and couplings. The safe operating space framework and our position towards it does change on all fronts simultaneously, first because all stressors are interconnected (discussed in first paragraph of section ‘Five potential tipping points’). Interconnection and heterogeneity as well as synchronous

increment of stressors will most likely shift the tipping points compared to looking at an isolated part of the Earth system. Isolation is actually the big argument for using toy models as an important supplement in Earth system science, since humanity has not yet understood what it actually means to have a hyperdimensional system with tons of trade-offs and agents across many scales.

(5) Can climate-driven degradation, in conjunction with land use, create an especially dangerous tipping point situation, with strong hysteresis? This means that even with overshooting and attempting to get GHG levels much lower, the system is locked in a new state where the rainforest cannot return.

Thank you again for raising this important point. If climate and land-use changes interact synergistically, this can certainly reduce the threshold of each stressor for tipping. This is also what some studies found (e.g. Nobre et al. 2016) but deserves more research. Fire, for instance, as part of land-use disturbances can make a potential hysteresis much stronger. In terms of overshoots of CO₂ emissions, there might be a 'lock in effect' but it has not sufficiently been tested yet with ESMs.

If the authors like the idea of an extra "roadmap" paragraph, as suggested above, then I think it makes a better ending to the current paper form. In my view, the current paper version ends too abruptly, with a quite specific title "Managing Amazon forest resilience". At the minimum, there needs to be some mention of next steps required for scientific research, based on the findings shown in this review. Arguably there is a possible disconnect in the paper between the data-led findings and the reported ESM projections, and how they may be combined to refine future estimates of Amazon die-back risk.

We now provide a clear roadmap as a 'Prospects for modelling Amazon dynamics' section (line 577), elaborating on what we identified as most important points that should be covered. Again the potential list of things that could be covered here is much longer than what we provide, but we believe that we described, in light of our findings, the main priorities to represent and assess tipping points within the Amazon system.

I like the use of the UTrack atmospheric moisture tracking dataset to characterise connectivity (Figure 3). As the paper authors will know, there is particular interest in the risk of tipping point "cascades", but the coupling between effects is poorly understood. Here, the coupling (Figure 3) is between two similar potentially tipping point (e.g. rise of dieback in East Amazon and West Amazon), but I think this is especially novel and important. A potential suggestion for future research might be to compare ESMs (and maybe trends if enough data) of known atmospheric transport against ESM projections. Do raise the profile of this part of the paper, because it starts to answer the spatial connection of tipping points, which is a weak point in understanding what the future will be like as GHGs rise.

This point is indeed important and deserves more research. We now better underline the importance of potential tipping cascades (section 'Sources of connectivity', line 352) and call for identifying model shortcomings and hence needed development in (model section).

Overall, I believe this manuscript can make a substantial contribution to understanding the potential fate of the Amazon rainforest in response to a changing climate. However, I think it needs to be clear that this manuscript does not yet provide process-based equations along with their parameterisation, of the effects considered. Those responsible for developing ESMs are not provided here with a set of equations ready for numerical discretisation and subsequent mapping onto the land part of ESMs. By the authors' own assessment (Table 1), even after their analysis, certainty in levels of climate to induce tipping points is either "low" or "medium". This honesty is to be commended, and this manuscript will certainly focus the minds of those undertaking future Amazon research.

Thank you for this positive and realistic evaluation. We agree that this is essential work for future modelling studies addressing the Amazon. As suggested by the reviewer, we now indicate in the section 'Prospect for modelling Amazon dynamics' that this manuscript does not deliver a comprehensive set of equations to parameterize models, but reviews and summarizes the state-of-art knowledge on potential Amazon forest dynamics in the Anthropocene.

Despite the points raised above, it needs to be recognised that this manuscript provides a powerful assessment of potential changes to the Amazon rainfall under multiple climate pressures (5 stressors – Table 1). I especially like the way climate stressors are put alongside direct deforestation forcings (Section: "Three alternative ecosystem trajectories"). Too many other papers are either about climate impacts alone or deforestation impacts alone, and never give them equal weighting.

I hope this paper can be published in some form. The paper is in review format and as such, does not determine to high precision the actual dangerous thresholds in climate change to prevent Amazon loss. It does, though, provide one of the most updated assessments of possible drivers and offers a robust roadmap to future research needed. At the risk of repeating the comments above, but with the influential IPCC reports so reliant on ESM projections, I think the paper would gain substantially from a small number of additional sentences that guide model developers on how they can use the findings of this review.

Thank you again for this positive evaluation. Of course model development is never finished and the topics where more model development is possible are countless. We are very happy to provide suggestions of model development. Therefore, we incorporated a section on model development needed to enhance model reliability from our interdisciplinary viewpoint which we think have made the manuscript more powerful.

I am happy to see any revised paper version.

Small Points

While I can see from Box 1, first panel, the diagram looks like a typical jump in state that is also present in simple dynamical systems models (e.g. a bifurcation diagram), I am not convinced the paper is really "Grounded in dynamical systems theory" (Abstract).

Concepts from dynamical systems theory inspired the review and our interpretations of the evidence in the literature. We deleted this part from the Abstract.

The authors should maybe note how eventually there may be better integration of ESM projections with contemporary measurements of the physical environment. As an example, in section “Global warming and climatic variability”, the same paragraph mentions observed trends and ESM projections, leading to Extended Data Fig 3. It would be good if at some point, similar diagrams to Extended Data Fig 3 are built by a fusion between noted climatic trends and ESMs, then driving ecosystem models.

In this study, we only included evidence from the literature, but we agree that this is an important research topic for future research, combining observed trends and ESM projections of climatic conditions.

The Figures are colourful and very informative. However, please pay a little more attention to the presentation. For instance, Figure 2, the legends are small, lat/lon tick marks are impossible to read. Of particular concern is that the colourbar for panel a is difficult to read, and at the minimum should have more levels given (why not make it horizontal, such as under panel d?).

Thank you. We improved the Figures, based on these suggestions.

The use of the literature is impressive, and that in itself makes this an especially useful paper.

Please try to avoid emotive language for a scientific paper. I have some sympathy, but somehow it does feel wrong here to describe low emissions scenarios as “optimistic” and high emissions scenarios as “pessimistic”. The reader will soon come to those conclusions anyway, but it must be written impartially.

We have now changed to ‘low’ and ‘high’ emission scenarios.

Referee #4 (Remarks to the Author):

General Comments:

Reviewer summary: The manuscript presents a review of the state of the science and literature surrounding potential Amazon tipping points. The authors place recommended values on five tipping points with associated confidence and studies supporting these recommendations. Crossing these tipping points are associated with potential ecosystem trajectories that are further presented with associated references and descriptions supported by the literature. This is finished with a discussion of the factors shaping and impacting forest resilience, including climate, disturbance, CO₂ fertilization, environmental site heterogeneity, land atmosphere teleconnections, and diversity for both the natural and human components. The authors conclude with a reflection of the recommendations from a 2012 report and a call to action for our current state. This call to action highlights the importance of global reductions in greenhouse gases, but also local scale action in the Amazon. The authors argue that this local action require local

and global collaboration to tackle the challenge of restoring and reversing the degradation of environmental policies in the region, so that we can maintain the Amazon and avoid catastrophic dieback.

Article contribution and overall impact: The Amazon has long been the focus of discussion as an important ecological indicator and tipping point essential to the health of our global system. This paper follows recent IPCC and other reports (Armstrong McKay et al 2022) highlighting the important aspects of the Amazon. The review aligns with these recent reports and studies on potential tipping points for the Amazon. There are multiple factors with the potential for interactions and feedbacks that will impact the Amazon, surrounding region and global system. Previous and recent articles have presented aspects of these potential drivers, interactions and feedbacks. The authors present their review and associated guidance with a wealth of studies from a range of disciplines bringing it together in a concise manner that adds to the discussion. In particular the authors highlight the importance of local communities as a force for conservation in the effort to halt illegal deforestation and degradation. This local voice is an important and essential part of discussion around the importance of the Amazon.

We appreciate your words about the importance of our article.

The modeling section needs to be either updated or substantially supplemented to reflect more recent modeling studies and results using more recent CMIP6 models, new data, and updated theory around interactions potential tipping points and increased understanding of feedbacks in the Amazon. Some of these updated studies are included in other sections, but the older modeling studies need to be replaced with new results, or presented in a format that demonstrates the progress made in modeling of the Amazon. Detailed comments are provided below.

Thank you for these comments. We have updated the modelling part with the suggested literature. In the revised version, we have deleted the former ‘Supporting evidence’ section and now we discuss modelling studies in sections ‘Five potential tipping points’, ‘Regional climatic variability’, ‘Environmental heterogeneity’ and a new section ‘Prospects for modelling Amazon dynamics’.

Detailed comments:

Line 83: Update this to “to help in creating a safe operating...”

Done.

Line 125-127: Use a different example of feedbacks that increase fire. Andrae et al 2008 does not make a link between smoke and decreased rainfall amount, rather they cite a delay in the initiation of precipitation and increased vigor of the storms. Perhaps Kumar et al 2022 for the interactions of land use with fire sensitivity.

Kumar, S., Getirana, A., Libonati, R., Hain, C., Mahanama, S., and Andela, N.:

Changes in land use enhance the sensitivity of tropical ecosystems to fire-climate extremes, *Sci Rep*, 12, 964, <https://doi.org/10.1038/s41598-022-05130-0>, 2022.

We have deleted the reference to a positive feedback between smoke and fire because the effect of smoke on rainfall is still uncertain (it could either increase or decrease) and also because this particular example was unnecessary. Thank you for indicating Kumar et al. (2022), which addresses the Pantanal wetlands.

Line 159: The impacts on the Amazon are not as clear in this reference, and suggest replacing it with Boulton et al 2022

Boulton, C. A., Lenton, T. M., and Boers, N.: Pronounced loss of Amazon rainforest resilience since the early 2000s, *Nat. Clim. Chang.*, 12, 271–278, <https://doi.org/10.1038/s41558-022-01287-8>, 2022.

Thank you. We removed the citation of Smith et al. 2022 from this sentence, since the point here is about increasing climatic stress, not about forest response. We already cited paper by Boulton et al. (2022) in other parts of the manuscript (e.g. ‘Disturbance regimes’, line 226).

Line 161-162: Update or supplement this with more recent modeling studies such as (Brando et al., 2020; Burton et al., 2022; Staal et al., 2020; Zemp et al., 2017)

Brando, P. M., Soares-Filho, B., Rodrigues, L., Assunção, A., Morton, D., Tuschneider, D., Fernandes, E. C. M., Macedo, M. N., Oliveira, U., and Coe, M. T.: The gathering firestorm in southern Amazonia, *Sci. Adv.*, 6, eaay1632, <https://doi.org/10.1126/sciadv.aay1632>, 2020.

Burton, C., Kelley, D. I., Jones, C. D., Betts, R. A., Cardoso, M., and Anderson, L.: South American fires and their impacts on ecosystems increase with continued emissions, *Climate Resilience*, 1, <https://doi.org/10.1002/cli2.8>, 2022.

Staal, A., Fetzer, I., Wang-Erlandsson, L., Bosmans, J. H. C., Dekker, S. C., van Nes, E. H., Rockström, J., and Tuinenburg, O. A.: Hysteresis of tropical forests in the 21st century, *Nat Commun*, 11, 4978, <https://doi.org/10.1038/s41467-020-18728-7>, 2020.

Zemp, D. C., Schleussner, C.-F., Barbosa, H. M. J., Hirota, M., Montade, V., Sampaio, G., Staal, A., Wang-Erlandsson, L., and Rammig, A.: Self-amplified Amazon forest loss due to vegetation-atmosphere feedbacks, *Nat Commun*, 8, 14681, <https://doi.org/10.1038/ncomms14681>, 2017.

In this sentence (which we deleted in the new version) we referred to the first modelling studies. In the sections ‘Global warming’ (lines 439-444) and ‘Prospects for modelling Amazon dynamics’ (lines 597-604), we incorporated more recent studies that incorporate the role of land-use disturbances.

Line 165: Update this to reflect a more recent reference

We rearranged the text. Section ‘Global warming’ compiles all the main evidence for this specific tipping point.

“Most CMIP6 models agree that a large-scale dieback of the Amazon is unlikely in response to global warming above pre-industrial levels ², but this ecosystem response is based on certain assumptions. For instance, most models assume a large CO₂-fertilization effect that could overestimate forest resilience ⁹⁰. Forests across the Amazon are already responding with increasing tree mortality rates that are not simulated by these models ^{74,75}, possibly because of rising disturbance regimes (Fig. 1). Nonetheless, a few global climate models indicate a broad range for a potential large-scale Amazonian tipping point in global warming between 2 – 6 °C ^{3,46,145–148}.”

Line 174: Update to “which is incredibly challenging to capture in a model, for instance...”

Done.

Line 171-173: This reference (Chai et al 2021) considers CMIP5 models to explore the impacts of climate on Amazon dieback, and should be replaced with literature for the more recent CMIP6 set of models. Or include a separate sentence highlighting the numerous studies showing that CMIP6 models show agreement and demonstrate increased drying and likelihood of drought in the Amazon.

Cook, B. I., Mankin, J. S., Marvel, K., Williams, A. P., Smerdon, J. E., and Anchukaitis, K. J.: Twenty-First Century Drought Projections in the CMIP6 Forcing Scenarios, *Earth’s Future*, 8, <https://doi.org/10.1029/2019EF001461>, 2020.

Parsons, L. A.: Implications of CMIP6 Projected Drying Trends for 21st Century Amazonian Drought Risk, *Earth’s Future*, 8, <https://doi.org/10.1029/2020EF001608>, 2020.

Ukkola, A. M., De Kauwe, M. G., Roderick, M. L., Abramowitz, G., and Pitman, A. J.: Robust Future Changes in Meteorological Drought in CMIP6 Projections Despite Uncertainty in Precipitation, *Geophys. Res. Lett.*, 47, <https://doi.org/10.1029/2020GL087820>, 2020.

We appreciate these reference suggestions. In section ‘Global warming’ we now say:

“... most updated CMIP6 models agree that droughts in the Amazon region will increase in length and intensity ^{50,51}, and that exceptionally hot droughts will become more common ⁴⁹, creating conditions that will likely boost disturbance regimes, including forest fires and deforestation ^{23,59,72,150,151}.”

Line 178: Update to “can help by projecting more realistic...”

Done.

Line 178-181: Supplement this sentence with the more recent modeling study (such as Burton et al 2022) demonstrating that with increased temperature, drying, land-use and

fire there is a potential for 30% loss of vegetation C with temperatures of 4 degree C.

Burton, C., Kelley, D. I., Jones, C. D., Betts, R. A., Cardoso, M., and Anderson, L.: South American fires and their impacts on ecosystems increase with continued emissions, *Climate Resilience*, 1, <https://doi.org/10.1002/cli2.8>, 2022.

In section ‘Accumulated deforestation’ we added this sentence: “Other more recent models incorporating fire disturbances support a potential tipping behaviour of the Amazon forest, simulating a biomass loss of 30 – 40 % under a high emission scenario (SSP5-8.5 at 4 °C)^{7,154}.”

Line 220-224: This result for Jones et al 2009 is quite outdated, and must be updated with more recent model studies such as those suggested earlier and within Armstrong McKay et al 2022.

We deleted that first sentence about Jones et al. 2009 and updated the whole section on ‘Global warming’, citing other studies, including Armstrong-McKay et al. 2022.

Line 223-224: It is unclear why these modeling results are being considered as separate in this section.

We now combined all the evidence in the sentence below copied from the ‘Global warming’ section: “... a few global climate models indicate a broad range for a potential large-scale Amazonian tipping point in global warming between 2 – 6 °C^{3,46,145–148}.”

Line 290: Update to “forest resilience are ...”

Done.

Line 350-351: This is a bold statement. Consider rephrasing to a main driver or provide some qualification on this degradation.

We now say in section ‘Degraded open-canopy ecosystems’ (line 566):

“At the regional scale, positive feedbacks involve interactions between deforestation and atmospheric moisture flow^{24,25}.”

Line 376-377: These are old references (Betts et al 2012 and Malhi et al 2009). Update with more recent studies such as those previously suggested.

We now also cite here Parsons (2020); Ukkola et al. (2020), Cook et al. (2020).

Line 468-470: Clarify what tipping points from table 1 that the seasonal forests are close to reaching or exceeding.

Done. We refer to the ones related to rainfall seasonality, MCWD and DSL.

Line 505: Update to “to changing stress conditions”

We now say: “Tree species diversity increases forest adaptability to climatic changes by offering different possibilities of functioning¹²¹.” (line 380).

Line 558: Add Hetch et al 2011 to numbered reference list.

Done.

Figures

Figure 2c: Consider a different set of colors. It is hard to see the small fine scale changes of fire (blue) against the forest (green).

Now this has changed to Fig. 1d. We improved the figure with new colors.

Figure 2d: Update the color of this figure and add more tick mark labels. It is hard to distinguish between the gradients and identify where tipping point of 2 is located.

We also improved this figure in various ways, including a different colour palette that now shows more clearly how tipping potential varies across the basin. We removed the tick labels, as they are not necessary in this figure.

References:

Line 716: Update the citation for ref 50: Armstrong McKay et al 2022
Armstrong McKay, D. I., Staal, A., Abrams, J. F., Winkelmann, R., Sakschewski, B., Loriani, S., Fetzer, I., Cornell, S. E., Rockström, J., and Lenton, T. M.: Exceeding 1.5°C global warming could trigger multiple climate tipping points, *Science*, 377, eabn7950, <https://doi.org/10.1126/science.abn7950>, 2022.

Done.

Referee #5 (Remarks to the Author):

Flores and colleagues review the main tipping point elements in Amazonia. The manuscript provides both quantitative and qualitative estimates of how much of the Amazon could be at risk due to changes in precipitation, rainfall seasonality, and air temperature. The manuscript also reviews how deforestation could impact forest stability elsewhere across the Basin. Another important contribution of the review relates to qualifying what type of ecosystems could replace Amazon forests if a tipping point is surpassed, providing a glimpse of likely ecosystem trajectories in the near future. Finally, the review assesses the ongoing environmental and climatological changes in Amazonia, and the options to manage Amazon forest resilience. While the manuscript addresses an important topic for conservation, climate change mitigation, and livelihood of millions of people, more detailed and clarification are needed.

Thank you, we provided detailed responses to all points raised by the reviewer.

1) The section about the five (four?) potential tipping points in Amazonia could provide a more in-depth understanding of the 1) specific mechanisms driving the potential changes in vegetation stability, 2) nuances related to where and when those changes would likely to occur, and 3) knowledge gaps preventing us from properly predicting where and when those changes would likely occur. Although there is a section about environmental heterogeneity, it is vague and does not address all tipping point elements.

We appreciate these suggestions. We agree that all these points are very important. The purpose of that section was to briefly describe each of the five tipping points, their drivers and supporting evidence. Now, we have separated both drivers related to rainfall seasonality to have in total five sub-sections. We also now explain in the introductory paragraph of the section how all five drivers are related somehow to water stress as an underlying mechanism for forest loss, which feeds back into the drivers (see new Fig. 3). We explain more details about the feedbacks stabilizing changes in the vegetation and nuances about where they are more likely to occur in the section 'Alternative ecosystem trajectories'. We also discuss a lot of these nuances about where and when tipping events and other types of ecosystem transitions may occur in the sections 'Global changes' (see lines 186-210), 'Disturbance regimes' (lines 235-259), and in Fig. 1. We discuss knowledge gaps a new section on 'Prospects for modelling Amazon dynamics'.

2) The definition of tipping points relies on the assumption that once a threshold is crossed, Amazonian forests in some parts of the Basin should be stuck in an alternative (stable) state, in part, because of feedbacks or intrinsic dynamics. However, there is very little discussion on whether these feedbacks are strong enough in Amazonia to drive long-term changes in vegetation stability. Instead, the review provides evidence that there are two basins of attractions (low and high tree canopy cover) and that Amazonian forests have been under pressure due to several stressors—which is important elements to push forests over the edge but not necessarily to maintain forests into that state.

Thank you for highlighting the need to strengthen the role of feedbacks in the manuscript. Indeed, feedbacks are certainly key mechanisms affecting forest resilience and ecosystem shifts. We now included a new Extended Data Table 1, with examples of important feedbacks in the system. At local scales, feedbacks are discussed in the 'Alternative ecosystem trajectories' section and shown in Fig. 4, to explain how each ecosystem trajectories proposed can be stabilized. Feedbacks operating at large scales are discussed in Box 1 and now shown explicitly in new Fig. 3b, involving interactions among the five drivers of tipping points.

3) Although the manuscript mentions that one tipping point in Amazonia is 20% of deforestation, there is a large body of work on this topic. I encourage the authors to provide a more balanced review of the current literature here, presenting that number as a hypothesis rather than a consensus. Also, the manuscript should provide more nuanced view about where major transformations in forest structure/composition could occur due to deforestation-driven changes in climate. For instance, in wetter portions of

the Amazon, major changes in forest stability due to indirect effects of deforestation are less likely to happen.

We agree that it is far from consensus. This is why we say in Fig. 3a (previously Table 1) that there is ‘low confidence’. In Fig. 3a, we also show a range (20 – 50 %) for the tipping point in ‘Accumulated deforestation’. In the section discussing ‘Accumulated deforestation’, we give more details about this range, including now recent projections that provide a more balanced review of the literature.

We describe nuanced changes in the forest in relation to deforestation-driven (and also emissions-driven) changes in Amazonian regional climate in the section ‘Degraded forests’, where we discuss precisely these more subtle changes in forest composition, functioning and the potentially stabilizing feedbacks. We also show where forests are more exposed to climate and land-use disturbance regimes in Fig. 1, with ‘Transition potential’ varying spatially across the Amazon due to differences in warming, rainfall reductions and extreme drought events (Fig. 1a-c).

4) The manuscript could be clearer about the likelihood of different regions of the Amazon being impacted by the different types of disturbances (The alternative ecosystem trajectories section). Some regions experience some types of disturbances more often than others. For instance, the western Amazon is more likely to be impacted by blowdown events, while other regions more likely to burn during dry years. Also, primary forest transitions into different forest types could be presented more clearly, given that not all the transitions are likely to occur with the same probability. How much of the Amazon could become a white- sand savanna? Probably a small area, whereas the entire Amazon could become degraded forests.

We have adapted Fig. 1 to show more clearly how different regions of the Amazon are impacted by different types of disturbance regimes, including those related to climate and human activities. The resulting map adding the different disturbances shows where ecosystem ‘transition potential’ (Fig. 1f) is highest in the Amazon, but the other maps (Fig. 1a-e) show the spatial variation of potential impact by each disturbance (or stressor) separately. We indirectly represent the risk of forest fires by including roads (see for instance Kumar et al. 2014). We did not include blowdown because these natural disturbances currently cause very low impact on the Amazon forest as a whole (Espírito-Santo et al. 2014 www.nature.com/articles/ncomms4434);, yet we do mention that windthrows (including blowdowns) may gain importance with global warming (lines 216-218).

For each one of the ‘Three alternative ecosystem trajectories’, we mention where across the Amazon they are most likely and provide our best estimate of the area currently covered by them, both determined by annual rainfall conditions (in Box 1b), by the types of disturbances and by other fine-scale environmental conditions (e.g. topography). For instance, in the case of white-sand savannas, we now say that this alternative trajectory is more likely within 14 % of the Amazon that is seasonally flooded.

5) Disturbance regimes: it would be helpful to separate anthropogenic disturbances from natural ones, as well as provide a balanced view of how they may interact.

We consider that frequency and intensity of climate-driven disturbances, such as extreme drought events, are indirectly influenced by anthropogenic greenhouse gas emissions. We tried to clarify this at the start of section ‘Disturbance regimes’.

“In this section, we outline the main changes in disturbance regimes associated with climatic variability and land use, as well as how these changes may reshape forest resilience. Because anthropogenic greenhouse-gas emissions are the primary cause of global warming², we consider that changes in climatic variability are indirectly influenced by human activities.”

Referee #6 (Remarks to the Author):

A. Summary of the key results

This manuscript presents two assertions: (1) that an ecological tipping point of reduced rainfall due to loss of tree canopy cover having the potential to shift the vegetation of the region from stable forest to stable savanna and/or unstable forest exists for the Amazon forest, and (2) that the region is currently approaching that tipping point.

B. Originality and significance: if not novel, please include reference

The assertion is supported mainly by an extensive literature review, as well as a correlation analysis of remotely sensed tree canopy cover against mean annual rainfall. Additional analyses are presented in the Supplementary Information, but the results of these are not well integrated into the paper.

We have tried to better integrate our additional analyses into the article. We restructured the sections, bringing the section ‘Forest resilience is changing’ earlier in the text to clarify the importance of assessing ecosystem resilience for understanding the risk of tipping points. In the end of this section, we now show in Fig. 1 how the distribution of various disturbance regimes across the Amazon can help us foresee potential for ecosystem transitions in the coming decades. We now included the reanalyses of tipping points and bistability ranges for MCWD and dry season length in the main text Box 1 (previously as Extended Data Fig. 1), together with mean annual rainfall, showing how these different dimensions of the rainfall regime contribute to shape forest resilience. We also made a new Fig. 3 integrating the gathered knowledge about how the five drivers of forest loss (five tipping points section) interact, forming a positive (self-reinforcing) feedback loop that may accelerate large-scale dieback of the Amazon. We think that after this revision, the manuscript sends a much clearer message to all types of readers.

Beyond the seminal ecological literature by Holling and others on systems theory, I am not qualified to review the thoroughness of the literature review on the Amazon region as a subject. However, I am qualified to review the basic ecology and application of remotely sensed tree-cover data, so there is where I will place my focus in the following sections C and D.

C. Data & methodology: validity of approach, quality of data, quality of presentation

The overall assertion and the analysis in Figure 1 rest on the assumption that rainfall is the driver of the tipping point. This is reasonable, but indirect or approximate. More directly, tree cover affects rainfall through changes in evapotranspiration and vapor pressure deficit, as noted on Line 128, which then result in changes in rainfall downwind. Barring a more complete analysis with a coupled vegetation-atmosphere model, which is understandably beyond the scope of this analysis, it would benefit the paper considerably to include scatterplots for one or both of these variables alongside the one for rainfall in Figure 1.

The reviewer is correct that ultimately, by changing evapotranspiration and moisture flow, all these five drivers (with tipping points) directly or indirectly increase water stress on trees, thus causing forest loss. We now included scatterplots of MCWD and dry season length versus tree cover in the main text Box 1 (previously as Extended Data Fig. 1), showing how different dimensions of the rainfall regime (MAP, MCWD, DSL) contribute to shape forest resilience, with their own tipping points and bistability ranges.

Thinking of how we could use our findings to create a conceptual model for the Amazon, we now included a diagram with interactions among the five drivers of water stress that cause forest loss, which clearly form a positive feedback loop that helps to explain how a large-scale dieback could occur (see Fig. 3b).

Additionally, and far more importantly, the use of the MODIS VCF Tree Cover layer relies on the assumption that tree cover is determined by climatological constraints. However, this is not true for the Amazon region. The dominant cause of tree canopy loss in the region is land-use conversion of forest to agriculture. Taking a single temporal snapshot of a dataset and neglecting to distinguish climatological from land-use effects invalidates this pivotal result in the paper. This neglect is aggravated by the exclusion of savanna biomes from the analysis.

The reviewer is correct that deforestation completely alters what would be a more 'natural' distribution of tree cover. This is why we did exclude areas of land-use from our reanalyses, using the most recent product by MapBiomas Project 2022. Also, because we wanted to assess tipping points in tree cover, we did include large areas of savanna surrounding the Amazon basin, including savannas along the northern Cerrado, the Gran Savana in Venezuela, the Llanos de Moxos in Bolivia and other smaller savannas across the basin. This is why we use the Amazon basin area for plotting Box 1. We now clarified this in the Box text and in the Supplementary Information.

D. Appropriate use of statistics and treatment of uncertainties

Being mainly a literature review, this criterion is mainly relevant for the correlation analyses presented in Figure 1. The scatterplot is certainly compelling visually, but little quantitative analysis is presented. The Supplementary Information mentions a potential analysis, but only the specification of parameters was described--not how they were chosen or the sensitivities of the results to those choices.

Thank you, we expanded the Supplementary Information to incorporate these suggestions. For the scatterplots in Box 1, we used potential analysis (Livina et al. 2010) to assess tipping (bifurcation) points and bistability ranges. We now included in the Supporting Information a sensitivity analysis that resulted in a range of possible tipping points (see Table S1). We show in the main text the values that mostly agree with the evidence found in the literature (e.g. Staver et al. 2011 in Fig. 3a).

E. Conclusions: robustness, validity, reliability

The conclusions that the Amazon region has and is crossing an ecological tipping point might be supported by the literature review presented. However, they are not supported by the analysis in Figure 1, which fails to distinguish land-use from climatological drivers on tree-canopy density.

Please, see our second response to reviewer's point C. We were able to distinguish land-use from climatological drivers on tree density because we did exclude all deforested areas by 2022 and include savannas in our reanalysis.

F. Suggested improvements: experiments, data for possible revision

The authors must decide if the empirical test of the tipping point presented in Figure 1 is central to their argument. If not, and the literature review is sufficient, then the analysis should be removed. If it is necessary, it must be strengthened by removing the currently unincorporated effect of land use on tree canopy cover. Additionally, a more thorough treatment would repeat the left panel of Figure 1 for evapotranspiration and/or vapor pressure deficit.

We already explained in the previous comments above that the reanalysis in Box 1 excluded the effect of land-use on tree cover density. We agree that it is an interesting and novel idea to test for the roles of evapotranspiration and VPD (preferably combined with groundwater availability) in shaping forest resilience. This is a review manuscript and we only reanalysed data from previous studies. To address the reviewer's request, we sent to the main text in Box 1 two plots showing tree cover vs MCWD and vs dry season length, which represent other dimensions of water stress.

G. References: appropriate credit to previous work?

Yes

H. Clarity and context: lucidity of abstract/summary, appropriateness of abstract, introduction and conclusions

As a review, the main manuscript was clear. However, as an analysis, the methods were inadequately described in the Supplementary Information.

We tried to clarify all the points raised by the reviewer, which helped us to improve the description of our methods and findings for the readers. For example in main text Box 1 we explicitly explain that deforested areas were excluded and savannas were included in the analyses (lines 101-104). In Supplementary Information we describe how we excluded areas deforested until 2020 (line 1160) and we provide sensitivity tests of the potential analysis (lines 1179-1192; Table S1).

Reviewer Reports on the First Revision:

Referees' comments:

Referee #1 (Remarks to the Author):

A&B. The revised manuscript has addressed most of the points raised in the first review by my and the other reviewers. It is improved and provides a summary and prospectus for the future of the Amazon through the lens of tipping points, and importantly includes drivers, and a methodology for assessing their importances, that is not a quantitative ESM. This is a good thing, as there is no assurance that the ESMs are right (or best), or that they give the flexibility in prediction that the authors' approach does. Not just flexibility to include multiple different drivers, but also an honesty in the level of certainty that we have in the outcomes. There are other discussions of tipping points, but they are almost universally model-based. This is also the viewpoint and review of scientists that live and/or have deep roots and ties to Amazonian systems.

C. The methodology for the tipping point potential index might be controversial, but it is justified. The figures could still use some attention. Box one, noted in detail below, still confuses/presents some extra-Amazonian habitats and could use some clarification and graphical improvements.

In terms of other inference, the authors pushed back in the responses on some paleoecological inference, and in the revision have factual errors about the timing of certain events that need to be corrected (see detailed comments). These are easily addressed.

D. Okay

E, F. No suggestions.

G. Good. Additional references on deep-time history provided below.

H. Good.

Specific Comments

Box 1. High Andean grasslands still included as savanna, and sentence at L101 says the analysis is for the Amazon Basin, not the Amazon Biome. In their response the authors said this was corrected, but it is still unclear. In this box it needs to be stated that high Andean grasslands are excluded from the analysis (e.g. bistability assessment). The line for the Amazon Biome could be made much thicker and bolder as well. It remains confusing as presented--the reader would have to look at the small, thin lines in the legend in panel B and then map those back to the distinction in the text between Amazon Basin and Amazon Biome, which most of the world will not be clear on anyway...

L128. This is misleading. There were tropical forests in Pan-Amazonia (roughly equivalent to the

authors' definition of the Amazon Biome) since the Paleogene (see Hoorn et al. 2010, fig 1B). The change at 7 million years ago was not the start of lowland tropical forests in what was Amazonian, but a change in their distribution. It is wrong to say otherwise. The next sentence as well--the paleoecology of Pan-Amazonia, even before the rise of the Andes, is that of a largely forested biome, with largely the same composition. The Amazon Biome, particularly as defined by the authors to include the Orinoco basin, Guiana Highlands, etc., did not originate 7m years ago!

Smaller point, but the authors treat the Amazon as if it were only trees, when all of the mutualisms and trophic interactions that create the forest (dispersal, pollination, soil microbes, etc.), and comprise the spectacular, non-tree biodiversity of the Amazon, have also been largely resilient, save for indigenous populations causing the extinction of the highly diverse megafauna ~12-15 kybp.

At some point in the manuscript defaunation in particular, and the loss of ecological interactions between species, needs to be addressed as a potential contributor to tipping points, even if it is to dismiss it. Large herbivores are key to the presence of savanna in what would be wooded habitats around the world, and results from the Andes suggests that loss of large herbivores led to increased tree cover there as well. What was the effect in the Amazon?

L259. There is a large literature on the importance of local recycling--how does this fit in to the areas susceptible to tipping? How does the prediction in this paragraph compare with other authors' predictions about areas vulnerable to tipping? Okay, returning to this after reading the entire manuscript, recycling is treated in another section. But, it is relevant to the disturbance section and might be mentioned briefly--disrupting the atmospheric flux of water and local recycling is a disturbance.

L317. The timing of geological events and its implication for the Amazon system needs attention in the paper in general, and in particular the date for the Andes (7 my) is unsupported. Boschman 2021 is a good overview [Boschman, L. M. (2021). Andean mountain building since the Late Cretaceous: A paleoelevation reconstruction. *Earth-Science Reviews*, 220, 103640.] The Andes uplift in the areas of interest started before 20-25 mybp (e.g. Sundell, K. E., Saylor, J. E., Lapen, T. J., & Horton, B. K. (2019). Implications of variable late Cenozoic surface uplift across the Peruvian central Andes. *Scientific reports*, 9(1), 4877.), Particularly in a way that would affect hydrology. By 7 mybp much of the action was over, though with exceptions in the N Andes and E Central Andes. But, for these purposes, the final uplift isn't the important part.

L353. Would be good to start new para with sentence that begins on this line--the ideas are distinct.

Referee #2 (Remarks to the Author):

The new version is improved, and I appreciate how hard it is to please 5 reviewers.

But the manuscript still needs a good edit for internal consistency. For example, is it tipping points

plural (in the title) or a tipping point (opening line of the abstract)?

Overall, there seems too little integration of specific points/points of difference amongst reviewers and manuscript authors. This means there is still a lack of clarity about what a tipping point is, how many there are, and what the mechanisms are.

Reading the response to reviewers, I think the manuscript authors agreed with me that there are not five tipping points, there is a lack of sufficient water threshold that can tip a forest into a non-forest system (which will differ in space and time). I think we agree that there are different mechanisms to get there (climate-induced drought and cumulative deforestation), that influence water availability in different ways, so we can estimate the threshold in different ways, overall rainfall vs length of drought vs intensity of drought. But, going over to the new version of the manuscript, there is still a section titled 'Five Tipping Points' at line 408. And still five sections on five tipping point, Global Warming, Overall rainfall, Dry Season length, Rainfall Seasonality Intensity, and Accumulated Deforestation. These are **not** five different 'tipping points', and they mix two different categories of things.

Then there is the new Fig. 3b, which I like, as it starts to bring clarity. It shows that there are two what we might call ultimate 'drivers of water stress' (global climate change, accumulated deforestation, the top and bottom halves of the 3b diagram), rather than five, which can each influence one or more of dry season length / more intense dry seasons / less overall rainfall (which we might call proximate 'drivers of water stress', to adopt the terminology of the classic Geist and Lambin 2002 Bioscience paper on causes of deforestation) . But that schematic is not very well reflected in the text.

Reading the response to reviewers, we seem to agree that a temperature tipping point is currently not so important as the water-related tipping points because we are still far from it. But then going over to the manuscript the first 'Tipping Point' is titled 'Global Warming' at line 425... which then doesn't do more than mention how GHGs drive climate change a key 'driver of water stress'. To me this text section contradicts the message on global warming in Fig 3b.

Also on Fig. 3b, it seems to me that the box 'Amazon regional climate' is not a subset of 'Global Warming' as depicted, but should be half in the local scale (deforestation) half of the diagram – as the combination of the two is a key point of the manuscript.

The second 'Tipping Point' on Rainfall Seasonality, line 458, further illustrates what is missing. There is a bistability at approx. -400 mm MCWD. This is what I would call the threshold (and I think we probably agree). But, the thing we need to know is how much of the Amazon is now greater than -400 mm and forest, and how much would be -400 mm under, say, 2 C warmer. And how much deforestation is required to push some fraction of the Amazon over -400 mm. The paper identified the thresholds (which is important), but doesn't identify the point at which some fraction of the Amazon would go over the threshold, by some level of climate change or deforestation. The 'safe space' is the GHG emissions limit to avoid some fraction of the Amazon going over -350 mm (or some other agreed threshold), not the -350 mm itself.

One further example of the lack of integration, in the new section, Prospects for Modelling, line 577, there is a section saying individual-based modelling is the way forward. But then the text jumps to saying, in the short-term we should make ESMs better. But elsewhere you cite a paper doing exactly as you ask (in response to a reviewer comment), a study modelling individual-level modelling of Amazon forests to test for abrupt changes. See Longo et al. 2018, your ref. 100.

I won't labour the point, but the manuscript still feels to me like it needs attention to get the framework straight (as I said in the initial review), and consistency across the manuscript.

Specifics:

Line 271, on CO₂ fertilization -- this is the wrong way round! Hubau et al. (2020) Nature, and others right back to Phillips et al. 1994 Science on increasing turnover rates, have noted that it is the trees on infertile soils will have a carbon sink for the longest, because the trees last longer, and the fertile soils will see the sink shut down sooner, as the system re-equilibrates faster. You say this earlier in the paragraph. Overall, I would also say you are too pessimistic on CO₂ fertilization impacts (and counter to the IPCC WG1 results).

Related, it is important to separate the CO₂ fertilization effect, which is very well known, and the ecosystem response to a long-term increase of CO₂ on one measure of the ecosystem response, net carbon uptake/release, the duration and magnitude of which are uncertain.

Fig 1a. Temperature. That is a striking bulls-eye in the middle of the Amazon, is it seen in other products. Is it real? And what could cause such a pattern of increasing heat centred on the middle of the Amazon?

Line 310. This section, says logic and the evidence suggest heterogeneity will reduce the risk of an abrupt and large scale transition, but then argues that this may not be the case. The evidence is that heterogeneity does reduce the risk, but that's not really the take-home message of the paragraph. I think the real question and the unknown is how much heterogeneity is there in Amazon forests, and is it enough to mitigate the risk, and I think writing that may be a clearer way to summarise this section.

Line 375. 99% of Amazon tree species are not rare. The statistic in the cited paper is 1% of species are 50% of trees and the other 99% of species are the other 50% of trees. Those 99% are not all rare. Some are pretty common, just no hyperdominant.

Line 374, Biodiversity section. I think summarising this paper, showing amazon tree species composition is tracking the changing climate would be good to summarise in this section. You could cite:

Esquivel-Muelbert, A, Baker, TR, Dexter, KG, et al. Compositional response of Amazon forests to climate change. *Glob Change Biol.* 2019; 25: 39– 56. <https://doi.org/10.1111/gcb.14413>.

line 396. "Through diverse ecosystem management practices, humans transformed the Amazonian

flora." Transformed? I am sure there is not evidence that they 'transformed' the Amazonian flora -- at most it was encouraging a few useful species to grow (not more than 10's to 100's out of 15,000 species), and the evidence for that is highly contested.

line 405. This sentence: "However, consistent loss of Amazonian languages is causing an irreversible disruption of ecological knowledge systems, mostly driven by road construction, which could contribute to further reducing forest resilience." Can you explain what the mechanism is that connects the loss of languages leading to lower forest resilience is.

line 425, section on global warming. It is very model-focused. You could describe the space-for-time study of Sullivan et al. 2020 Science, which show an abrupt loss of carbon above a max temp of 32 deg C.

line 576, Modelling section. You ought to note that is it the mortality in tropical forests and the woody turnover that ESMs perform poorly with, rather than the growth part of the simulation, which they do well at, when compared to forest data.

Koch et al (2021) Earth System Models Are Not Capturing Present-Day Tropical Forest Carbon Dynamics. *Earth's Future*, 9, e2020EF001874.

David Galbraith et al. (2013) Residence times of woody biomass in tropical forests, *Plant Ecology & Diversity*, 6:1, 139-157.

Referee #3 (Remarks to the Author):

Thank you for inviting me to re-reviewing the paper "Tipping points in the Amazon forest system".

I do remain convinced that this manuscript provides a state-of-the-art review of the risk of Amazon dieback, and is highly accessible to a broad set of readers. The quality of the display items is superb.

I want to thank the authors for taking the reviewer comments seriously, and I can see that there are comprehensive answers to both my suggestions, and the suggestions from the other five reviewers. However, in some instances, the responses are just to the reviewers – there are a few suggestions made that I think could support the manuscript if used to enhance the paper itself.

It is my review that has triggered the new paragraph "Prospects for modelling Amazon dynamics", and this addition is appreciated. The new paragraph is good, but it could gain from a little further improvement, and noting that it is currently incomplete. The paragraph describes well the need for explicit modelling, in ESMs, of forest functioning, forest dynamics and key physiological responses. However, it is also important to note the need for refinement of ESMs in projecting climate change, which they were originally designed to achieve (i.e. when they were Global Circulation Models).

This should not take the authors long to achieve, but please improve that paragraph by:

(1) Continue to state the importance of adding to ESMs representation of “plant community diversity and adaptation”.

(2) The paper Abstract mentions climatological concerns – “warming temperature, extreme droughts...”. So although obvious, to really determine when tipping points will occur, and to build on this Flores et al review, we do need accurate predictions of future climate change. Unfortunately, ESMs differ in their projection of Amazon warming rates, and expected changes in rainfall patterns. This needs to be mentioned in this “next steps” Section.

(3) Please mention methods are needed to reduce uncertainty in the projection of climate change. This can build on the Table / Schematic of Figure 3. The two ways are: (1) simply look at the performance of models for the contemporary period and (2) emergent constraints (ECs). In the response, the authors note the key issue that ECs fail if all models do not contain a key process affecting the Amazon. I would state that in this paragraph – so something like: “The emergent constraint method will fail if all ESMs are missing a common feature affecting the Amazon and our review helps to identify these underrepresented or currently ignored processes”.

(4) Check that the paragraph reads tightly, and I think in places could be more positive. For instance, remove the words “not even” (line 589) and lines 605-607, maybe “Required for implementation in ESMs is the need for a set of reliable and comprehensive equations to parameterise processes impacting land dynamics. Our study points to many of these missing modelled processes by providing state-of-the-art.....”

My comment concerning the need for factorial ESMs I would still like to be mentioned as a future aspiration for climate researchers. As is clear throughout the manuscript, tipping points often involved feedbacks between processes. Hence land surface “offline” simulations, which although easier to compute, may not be sufficient to determine the full role of proposed new mechanisms. Please just add a sentence somewhere to note that – doing so will make this a paper cited by those arguing for more computer power to operate ESMs.

I am happy to review any further manuscript iteration.

(One small bit of advice to the authors for any future submission! It helps the reviewer a lot at the second revision stage if it is clear what amendments have been made to the original paper. Either cite new text in the response document or if allowed by the journal, provide a track-changes version).

Referee #4 (Remarks to the Author):

General Comments on revision:

Reviewer summary: The manuscript presents a review of the state of the science and literature surrounding potential Amazon tipping points alongside analysis to visualize tipping points and stable states responding to water conditions of rainfall, water deficit and dry season length. The authors

place recommended values on five tipping points with associated confidence and studies supporting these recommendations. Crossing these tipping points are associated with potential ecosystem trajectories that are further presented with associated references and descriptions supported by the literature. The authors provide markers for safe operating space to maintain resilience of the Amazon forest.

Article contribution and overall impact: This manuscript adds value to an important conversation. The authors have done a strong job of responding to the feedback of such diverse reviews, and the revision addresses the concerns. The challenges facing the Amazon are diverse and of interest to broad groups with very different specializations. The authors provide a manuscript that highlights these challenges by including an extensive list of relevant references across disciplines. The new structure is quite different from the first submission, and would benefit from some tightening of the language in the early sections. Detail comments are provided below.

Detailed comments:

Line 88: Update this to “at which disturbance may cause...”

Line 121-122 (Box 1): This is not clear. Reword to clearly state the mechanism you are referring to for fire. Closed canopy forests depress fire. Suggest adding a phrase identifying the mechanism (canopy loss promotes fire which is a positive feedback on further canopy loss, etc.).

Line 186-210: This is a good section, but a bit hard to follow the flow of ideas and their connections. Try to clear this up.

Line 208-210: Suggest adding a short sentence explaining why bistability is more likely to increase tipping events. (This comes in later in the text, but suggest having some explanation here.)

Line 219: Provide a brief definition of edge effects and their importance. This is referenced in regards to flammability in the later modeling section, but define the importance of edge effects here as well. For instance, edges contribute to degradation through changes in microclimate which increasing drying and vulnerability to fire. There are multiple references in the manuscript that can inform the importance of edges (Brando et al 2014; Alencar et al 2015; Brando et al 2020; Brando et al 2019; Uhl et al)

Line 238-243: Reword this section. It reads as a list which detracts from the impact of combining this information for policy. There is a higher impact statement that can come out of this.

Line 245-248: Add a sentence regarding the uncertainty of using an additive approach. By adding the disturbance you account for the increased likelihood, but you do not account for feedbacks or potential changes that may act as a negative if one disturbance decreases the chance of another. Cumulative and compound disturbances are neglected in this additive approach.

Line 261: Update to “to increase the photosynthetic rates...”

Line 261-279: The CO₂ fertilization section reads well.

Line 326: Suggest rewording this to “drought due to historical pressures, which allows them...” and consider providing a time context (decades, centuries, etc)

Line 339: Add details about why the system is increasingly homogenous. Even stand age of forests recovering?

Line 358: Add illegal logging to this list with fire and overhunting

Line 402-403: Update to “Amazonian regions with the highest linguistic diversity...”

Line 423: Update this to include the purpose of this safe operating space. Such as: “...safe operating space for the continued resilience of the Amazon forest.”

Line 433: Update to “...results about tipping behavior...”

Line 505: Update to “we identify the three most plausible...”

Line 548-549: Consider updating the sentence to indicate that it is unlikely for these forests to shift back to forest once they transition to white-sand savanna. Or add a short sentence clarifying this point.

Line 566-67: Update to include a phrase identifying that deforestation is associated with reduced moisture.

Figures:

Figure 3: Line 498-500: Suggest updating caption to include the name of the drivers rather than just the number.

Referee #6 (Remarks to the Author):

This manuscript is much improved from its previous submission. It is more clearly presented as a review, and the avoidance of its analyses of the possible confounding effects of land use have been clarified. I only have three remaining points to address:

(1) The MODIS VCF has nonuniform accuracy across the range of tree cover, with tree-cover values in the "savanna" range being among the weakest. Note also the saturation evident in Box 1 of values above 80% cover, which is due to a persistent artifact in the original response design that may or may not have remained unresolved.

See Figure 4 in Sexton, J.O., X.-P. Song, M. Feng, P. Noojipady, A. Anand, C. Huang, D.-H. Kim, K.M. Collins, S. Channan, C. DiMiceli, J.R. Townshend. 2013. Global, 30-m resolution continuous fields of tree cover: Landsat-based rescaling of MODIS continuous fields and lidar-based estimates of error. *International Journal of Digital Earth* 6(5): 427-448.

The authors should find a more recent validation of the MODIS VCF if one exists and provide at least one statement in the paper explaining the resulting possibility of spurious effects, as a caveat on their analyses.

2. The word "significant" is used multiple times in reference to Figure 1. To better match the text to the Figure, and to better convey the statistical certainty of the findings, all pixels whose linear regression slopes are not significant at some chosen (and stated) p-value should be masked from the figure. This can be done simply by making those pixels transparent in color.

3. Anthropogenic conversion of forest to agriculture is currently a far more powerful driver of forest loss in the Amazon biome than climate change. The MapBiomas project as well as several papers have come out recently on the rates of forest loss in the Amazon. To give a complete accounting of the region, and to put the climate and land use effects in perspective, the review must add a paragraph simply giving the rates of forest loss in the region due to land-use change.

Author Rebuttals to First Revision:

Referees' comments:

Referee #1 (Remarks to the Author):

A&B. The revised manuscript has addressed most of the points raised in the first review by my and the other reviewers. It is improved and provides a summary and prospectus for the future of the Amazon through the lens of tipping points, and importantly includes drivers, and a methodology for assessing their importances, that is not a quantitative ESM. This is a good thing, as there is no assurance that the ESMs are right (or best), or that they give the flexibility in prediction that the authors' approach does. Not just flexibility to include multiple different drivers, but also an honesty in the level of certainty that we have in the outcomes. There are other discussions of tipping points, but they are almost universally model-based. This is also the viewpoint and review of scientists that live and/or have deep roots and ties to Amazonian systems.

We appreciate the reviewer's comment and agree that our manuscript addresses this topic from a broader perspective considering theoretical and empirical evidence.

C. The methodology for the tipping point potential index might be controversial, but it is justified. The figures could still use some attention. Box one, noted in detail below, still confuses/presents some extra-Amazonian habitats and could use some clarification and graphical improvements.

Thank you for calling attention to this lack of clarity in relation to the purpose of Box 1. We have tried to further clarify in the Box (copied below) that the map

and analyses use the area of the Amazon basin, which includes some large areas of savanna ecosystems in the periphery of the basin. This approach was chosen precisely to allow an assessment of forest and savanna distributions within the range of rainfall conditions using potential analysis (Livina et al. 2010).

“By excluding accumulated deforestation until 2020²¹ and including large areas of tropical savanna biomes in the periphery of the Amazon basin, we are able to assess the stability of forests and savannas within rainfall gradients (see Supplementary Information for details” (lines 108-111)

In terms of other inference, the authors pushed back in the responses on some paleoecological inference, and in the revision have factual errors about the timing of certain events that need to be corrected (see detailed comments). These are easily addressed.

We have taken careful attention to these comments regarding our paleoecological inference, and have incorporated the reviewer’s suggestions (please see below each specific comment).

D. Okay

E, F. No suggestions.

G. Good. Additional references on deep-time history provided below.

H. Good.

Specific Comments

Box 1. High Andean grasslands still included as savanna, and sentence at L101 says the analysis is for the Amazon Basin, not the Amazon Biome. In their response the authors said this was corrected, but it is still unclear. In this box it needs to be stated that high Andean grasslands are excluded from the analysis (e.g. bistability assessment). The line for the Amazon Biome could be made much thicker and bolder as well. It remains confusing as presented--the reader would have to look at the small, thin lines in the legend in panel B and then map those back to the distinction in the text between Amazon Basin and Amazon Biome, which most of the world will not be clear on anyway...

Thank you for these suggestions.

For the potential analysis that reveals alternative states of equilibrium in tree cover, we included these savannas and grasslands in peripheral parts of the Amazon basin, such as the Andean grasslands, precisely because they represent the low tree cover areas within the rainfall gradient. This is why we used the basin area for the potential analyses, instead of the biome area that excludes these peripheral savannas and grasslands.

We have improved the figure by showing the Amazon biome in thicker black bold line, which indeed is much more evident now.

L128. This is misleading. There were tropical forests in Pan-Amazonia (roughly equivalent to the authors' definition of the Amazon Biome) since the Paleogene (see Hoorn et al. 2010, fig 1B). The change at 7 million years ago was not the

start of lowland tropical forests in what was Amazonian, but a change in their distribution. It is wrong to say otherwise. The next sentence as well--the paleoecology of Pan-Amazonia, even before the rise of the Andes, is that of a largely forested biome, with largely the same composition. The Amazon Biome, particularly as defined by the authors to include the Orinoco basin, Guiana Highlands, etc., did not originate 7m years ago!

Thank you for these clarifications. We checked again Hoorn et al. 2010 and the reason why we had considered 5 Ma as a conservative age for the Amazon forest biome is because a large part in the west was wetland before 7 Ma. This wetland was mostly covered by water or grassland ecosystems. When these wetlands retreated, forests expanded, as described in several parts of their paper (see below). The legend of Fig 1 (in Hoorn et al. 2010) says: "(E) The megawetland disappeared and terra firme rainforests expanded ..." corresponding to the phase from 7 – 2.5 Ma. Later they also provide more clues about the timing of forest expansion in the western Amazon (see sentences copied below):

- "This so-called "Acre" system harbored a very rich aquatic vertebrate fauna that included mega-sized gharials, caimanines, and side-neck turtles (39), which eventually declined with the disappearance of megawetlands in Western Amazonia at ~7 Ma (Fig. 2A)".
- "The floodplains of this system were dominated by grasses (51) and were inhabited by a more diverse xenarthran fauna than at present (52)."
- "Preliminary palynological evidence indicates a ~10 to 15% increase of plant diversity between ~7 and 5 Ma, shortly after the wetlands were replaced by forested habitats (Fig. 2A)."

For precision, we now say "Seven million years ago, the Amazon river began to drain the massive wetlands that covered most of the western Amazon, allowing forests to expand over grasslands in that region, but the Amazon system as a whole has been mostly covered by forest for 65 million years ²⁷." (line 132)

Smaller point, but the authors treat the Amazon as if it were only trees, when all of the mutualisms and trophic interactions that create the forest (dispersal, pollination, soil microbes, etc.), and comprise the spectacular, non-tree biodiversity of the Amazon, have also been largely resilient, save for indigenous populations causing the extinction of the highly diverse megafauna ~12-15 kybp. At some point in the manuscript defaunation in particular, and the loss of ecological interactions between species, needs to be addressed as a potential contributor to tipping points, even if it is to dismiss it. Large herbivores are key to the presence of savanna in what would be wooded habitats around the world, and results from the Andes suggests that loss of large herbivores led to increased tree cover there as well. What was the effect in the Amazon?

This is a very interesting point that leads to an original question of whether the megafauna extinction around 12 Ka contributed to increase forest resilience by reducing herbivory. It certainly is a promising topic for future research, but seems speculative for our review, since we are unaware of any study testing this hypothesis. Nonetheless, we do briefly discuss the role of interactions with animals in section 'Biodiversity'. In a recent review (<https://onlinelibrary.wiley.com/doi/10.1111/gcb.16293>), we discuss in more detail how interactions with animals in tropical forests could affect forest resilience, including the extinction of the megafauna. Unfortunately, we do not have space here to expand on this fascinating topic.

L259. There is a large literature on the importance of local recycling--how does this fit in to the areas susceptible to tipping? How does the prediction in this paragraph compare with other authors' predictions about areas vulnerable to tipping? Okay, returning to this after reading the entire manuscript, recycling is treated in another section. But, it is relevant to the disturbance section and might be mentioned briefly--disrupting the atmospheric flux of water and local recycling is a disturbance

We agree that this is a key disturbing mechanism that weakens the forest capacity to recycle rainfall and may increase the system's vulnerability to tipping. For the transition potential (Fig. 1f), we only considered local disturbance regimes, but as the reviewer correctly argues, the effects of forest loss elsewhere in the Amazon will likely reflect on disturbance regimes in another area through changes in local rainfall and temperature. In the section 'Sources of connectivity' we provide more details on this mechanism. We also mention disturbances on the rainfall recycling in several parts of the manuscript (see examples copied below).

"This difference between both ensembles is possibly related to the forest-rainfall feedback, included only in the five coupled models, which increases total annual rainfall along the southern Amazon and hence the stable forest area when deforestation is not included in the simulations ^{4,51}." (line 205)

"Of the remaining forests, at least 38 % have been degraded by extreme droughts and other land-use disturbances ⁵³, with impacts on moisture recycling that are still uncertain." (line 494)

"At the local scale, two practical and effective actions need to be addressed to re-strengthen forest-rainfall feedbacks that are crucial for the resilience of the Amazon forest ^{4,51,177}: (1) ending deforestation and forest degradation (99 % of which is illegal ¹⁷⁸), and (2) promoting forest restoration in degraded areas." (line 641)

L317. The timing of geological events and its implication for the Amazon system needs attention in the paper in general, and in particular the date for the Andes (7 my) is unsupported. Boschman 2021 is a good overview [Boschman, L. M. (2021). Andean mountain building since the Late Cretaceous: A paleoelevation reconstruction. *Earth-Science Reviews*, 220, 103640.] The Andes uplift in the areas of interest started before 20-25 mybp (e.g. Sundell, K. E., Saylor, J. E., Lapen, T. J., & Horton, B. K. (2019). Implications of variable late Cenozoic surface uplift across the Peruvian central Andes. *Scientific reports*, 9(1), 4877.), Particularly in a way that would affect hydrology. By 7 mybp much of the action was over, though with exceptions in the N Andes and E Central Andes. But, for these purposes, the final uplift isn't the important part.

Thank you. We deleted this sentence to improve the flow and avoid confusion.

L353. Would be good to start new para with sentence that begins on this line--the ideas are distinct.

Now paragraph included.

Referee #2 (Remarks to the Author):

The new version is improved, and I appreciate how hard it is to please 5 reviewers.

But the manuscript still needs a good edit for internal consistency. For example, is it tipping points plural (in the title) or a tipping point (opening line of the abstract)?

We appreciate the reviewer's constructive criticisms and efforts to help us improving the manuscript. We have taken care when dealing with all suggestions. We agree that there is one single tipping point, and now we modified consistently throughout the manuscript that we identified five critical thresholds related to an underlying tipping point in water stress. Please see more details below.

Inspired by these reflections on the concept of tipping point, we have changed our title to "Tipping point in the Amazon forest system", which now refers to the tipping behaviour of the system and its underlying mechanisms. We hope to have reconciled our views in a way that will also reach an even broader audience.

Overall, there seems too little integration of specific points/points of difference amongst reviewers and manuscript authors. This means there is still a lack of clarity about what a tipping point is, how many there are, and what the mechanisms are.

We tried to clarify our refined definition that is in agreement with the reviewer and also among authors. In addition to changing the title, we now say:

In the Abstract - "We review and discuss existing evidence for five major drivers of water stress on the Amazon forest, as well as potential critical thresholds of those drivers that, if crossed, could trigger local, regional or even a systemic tipping point."

In the Introduction - "It remains uncertain whether a large-scale collapse of the Amazon could actually happen within the 21st Century, and if this would be associated with a particular tipping point. Understanding the risk of such catastrophic behaviour requires addressing complex factors that shape forest resilience¹⁴. In this review, we gather and discuss the evidence from paleorecords, observational data and modelling studies for the existence of critical thresholds in the system, as well as the main feedbacks that could push the Amazon forest towards a tipping point."

Reading the response to reviewers, I think the manuscript authors agreed with me that are not five tipping points, there is a lack of sufficient water threshold that can tip a forest into a non-forest system (which will differ in space and time). I think we agree that there are different mechanisms to get there (climate-induced drought and cumulative deforestation), that influence water availability in different ways, so we can estimate the threshold in different ways, overall rainfall vs length of drought vs intensity of drought. But, going over to the new version of the manuscript, there is still a section titled 'Five Tipping Points' at line 408. And still five sections on five tipping point, Global Warming, Overall rainfall, Dry Season length, Rainfall Seasonality Intensity, and Accumulated Deforestation. These are *not* five different 'tipping points', and they mix two different categories of things.

We agree with this view, and as explained above, we now use the term ‘critical thresholds’, except when we refer to a single tipping point of the system related to water stress. We also changed the name of this section to “Five critical drivers of water stress”. We believe the manuscript is now consistent with the different comprehensions of these concepts.

Then there is the new Fig. 3b, which I like, as it starts to bring clarity. It shows that there are two what we might call ultimate ‘drivers of water stress’ (global climate change, accumulated deforestation, the top and bottom halves of the 3b diagram), rather than five, which can each influence one or more of dry season length / more intense dry seasons / less overall rainfall (which we might call proximate ‘drivers of water stress’, to adopt the terminology of the classic Geist and Lambin 2002 Bioscience paper on causes of deforestation) . But that schematic is not very well reflected in the text.

We now tried to keep consistency throughout the manuscript regarding the message of Fig. 3, as explained in response to the comments above.

Reading the response to reviewers, we seem to agree that a temperature tipping point is currently not so important as the water-related tipping points because we are still far from it. But then going over to the manuscript the first ‘Tipping Point’ is titled ‘Global Warming’ at line 425... which then doesn’t do more than mention how GHGs drive climate change a key ‘driver of water stress’. To me this text section contradicts the message on global warming in Fig 3b.

According to the evidence we gathered in this review, global warming increases water stress in most parts of the Amazon forest indirectly by altering climatic conditions in the region. Global warming is widely projected to cause reductions in annual rainfall and increases in rainfall seasonality, as well as increases in regional temperature (both of which are being confirmed by observations, see ED Fig. 1 and ED Fig. 2). As a result, VPD is increasing, accentuating water stress. We explain this link in section “Regional climatic conditions” (see text copied below). Fig. 3b is aligned with the exact same message. In the section “Global warming” we mainly focus on the evidence that a few models suggest a critical threshold in global warming, as a result of these indirect effects. The main advantage of focusing on global warming, rather than global climate change, is because the first is changing gradually and clearly in response to GHG emissions, whereas the latter is a highly complex process, much more difficult to relate with Amazonian climatic conditions. Focusing on global warming is therefore much more useful for thinking on governance strategies to address the problem. Moreover, in Fig. 3b, we could have included many other variables that affect water stress on the system, such as temperature and VPD, but for simplicity we preferred to focus on the five drivers with evidence of critical thresholds.

“For instance, by 2050 (compared to pre-industrial conditions), models project a significant increase in the number of consecutive dry days by 10 - 30 days, and in annual maximum temperatures by 2 – 4 °C, depending on the greenhouse-gas emission scenario ². These climatic conditions could expose the forest to high vapour pressure deficits ⁴¹, threatening forest persistence through unprecedented levels of water stress ⁴⁰.” (line 170)

Also on Fig. 3b, it seems to me that the box ‘Amazon regional climate’ is not a

subset of 'Global Warming' as depicted, but should be half in the local scale (deforestation) half of the diagram – as the combination of the two is a key point of the manuscript.

Thank you. We agree with this interpretation and have adapted the figure to place the box in the centre of the diagram.

The second 'Tipping Point' on Rainfall Seasonality, line 458, further illustrates what is missing. There is a bistability at approx. -400 mm MCWD. This is what I would call the threshold (and I think we probably agree). But, the thing we need to know is how much of the Amazon is now greater than -400 mm and forest, and how much would be -400 mm under, say, 2 C warmer. And how much deforestation is required to push some fraction of the Amazon over -400 mm. The paper identified the thresholds (which is important), but doesn't identify the point at which some fraction of the Amazon would go over the threshold, by some level of climate change or deforestation. The 'safe space' is the GHG emissions limit to avoid some fraction of the Amazon going over -350 mm (or some other agreed threshold), not the -350 mm itself.

In our reanalyses (Extended Data Fig. 2b; Extended Data Fig. 3), we estimated how much of the Amazon may pass a critical threshold by 2050 using mean annual precipitation (MAP), instead of MCWD, because most studies using this approach also used MAP as a measure of total amount of water availability (e.g. Hirota, et al. 2011; Staver et al. 2011; Staal et al. 2020). Changes in rainfall conditions are available from historical observations and CMIP6 models, but indeed, assessing the effect of deforestation on these changes can be challenging. Zemp et al. (2017), for instance, used moisture tracking and deforestation scenarios to quantify these effects and estimated that deforestation may cause up to 20 % of reductions in the dry-season rainfall in the Bolivian and Peruvian Amazon. Smith et al. (2023, Nature) used correlations between observed rainfall and deforestation dynamics to estimate that for each 1% of accumulated deforestation within a 200 km pixel, annual rainfall would decrease by 2,76 mm. This effect is mostly local or regional, thus not accounting for longer-range effects, which we know (for instance from moisture tracking) exist as well (as shown by Zemp et al. 2017). In sum, future studies are needed to disentangle the effects of global warming and deforestation on rainfall conditions across the Amazon and other regions.

Zemp, D. C., C.-F. Schleussner, H. M. J. Barbosa, and A. Rammig. Deforestation effects on Amazon forest resilience, *Geophys. Res. Lett.*, 44, 6182–6190. (2017).

One further example of the lack of integration, in the new section, Prospects for Modelling, line 577, there is a section saying individual-based modelling is the way forward. But then the text jumps to saying, in the short-term we should make ESMs better. But elsewhere you cite a paper doing exactly as you ask (in response to a reviewer comment), a study modelling individual-level modelling of Amazon forests to test for abrupt changes. See Longo et al. 2018, your ref. 100.

Thank you for highlighting this confusion. We mean that ESMs can improve by incorporating individual based simulations, even though this may take some time to become feasible. We now also cite Longo et al. 2018 and say: "Forest functioning and dynamics start and scale up from the tree individual level, implying that this spatial and mechanistic resolution is desirable as an

improvement for ESMs to truly test for Amazon forest tipping behaviour (e.g. ^{98,99,117}).” (line 589)

I won't labour the point, but the manuscript still feels to me like it needs attention to get the framework straight (as I said in the initial review), and consistency across the manuscript.

We have reworked the text very carefully to incorporate our adapted framework, now considering a single tipping point in water stress, and to keep consistency with the use of the terms 'critical thresholds' and 'tipping point'.

Specifics:

Line 271, on CO₂ fertilization -- this is the wrong way round! Hubau et al. (2020) Nature, and others right back to Phillips et al. 1994 Science on increasing turnover rates, have noted that it is the trees on infertile soils will have a carbon sink for the longest, because the trees last longer, and the fertile soils will see the sink shut down sooner, as the system re-equilibrates faster. You say this earlier in the paragraph. Overall, I would also say you are too pessimistic on CO₂ fertilization impacts (and counter to the IPCC WG1 results).

We agree that trees on infertile soils usually live longer and therefore may have a longer carbon sink, as stated early in this paragraph. What we refer to here is the CO₂ fertilization effect, which most recent field experiments indicate will be limited on infertile soils (e.g. Ellsworth et al. 2017; Terrer et al. 2019). This has consequences that are relatively complex, and which we have tried to disentangle in this paragraph. We now rephrased to acknowledge this aspect highlighted by the reviewer: “Hence, it is possible that in the fertile soils of the western Amazon and varzea floodplains ⁹⁰, forests may gain resilience from increasing atmospheric CO₂ (depending on how it affects tree mortality rates) ...” (line 276)

Related, it is important to separate the CO₂ fertilization effect, which is very well known, and the ecosystem response to a long-term increase of CO₂ on one measure of the ecosystem response, net carbon uptake/release, the duration and magnitude of which are uncertain.

We agree that the CO₂ fertilization effect on ecosystem carbon balance is a key process for understanding feedbacks between the forests and carbon emissions, as well as changes in forest functioning. However, in this review, our focus is on forest ecological resilience. This is why we try to integrate in this section the potential outcome of CO₂ fertilization on the risk of a tipping point. Of course, if the forest shifts from source to sink, this will feed back to the system through the indirect effects of global warming, but the forest may still be there. Our direct interest in this section is how CO₂ fertilization affects forest persistence.

Fig 1a. Temperature. That is a striking bulls-eye in the middle of the Amazon, is it seen in other products. Is it real? And what could cause such a pattern of increasing heat centred on the middle of the Amazon?

We agree. It seems to be consistent with Fig. 1a of Marengo et al. (2022) showing the same pattern using ERA5 data. One possibility is that this pattern is caused by deforestation and urbanization in the central Amazon, but it needs to be studied in much more detail.

Marengo, et al. (2022). Increased climate pressure on the agricultural frontier in the Eastern Amazonia–Cerrado transition zone. *Scientific reports*, 12(1), 457.

Line 310. This section, says logic and the evidence suggest heterogeneity will reduce the risk of an abrupt and large scale transition, but then argues that this may not be the case. The evidence is that heterogeneity does reduce the risk, but that's not really the take-home message of the paragraph. I think the real question and the unknown is how much heterogeneity is there in Amazon forests, and is it enough to mitigate the risk, and I think writing that may be a clearer way to summarise this section.

Thank you for this suggestion. The reviewer is right regarding the take home message. We now say in the last sentence: "In sum, the effects of heterogeneity on Amazon forest resilience have been poorly investigated so far (but see ^{51,98,99,107}) and many questions remain open, such as how much heterogeneity exists in the system and whether it can mitigate a systemic transition." (line 342)

Line 375. 99% of Amazon tree species are not rare. The statistic in the cited paper is 1% of species are 50% of trees and the other 99% of species are the other 50% of trees. Those 99% are not all rare. Some are pretty common, just no hyperdominant.

Indeed, they are not all rare, but most of them are. We rephrased to say: "Amazonian forests are home to more than 15,000 tree species, of which 1 % are dominant and the other 99 % are mostly rare."

Line 374, Biodiversity section. I think summarising this paper, showing amazon tree species composition is tracking the changing climate would be good to summarise in this section. You could cite:

Esquivel-Muelbert, A, Baker, TR, Dexter, KG, et al. Compositional response of Amazon forests to climate change. *Glob Change Biol*. 2019; 25: 39–56. <https://doi.org/10.1111/gcb.14413>.

The purpose of this section is to explain the mechanisms by which biodiversity increases ecosystem adaptability to global changes. We cite another paper from the same authors (Esquivel-Muelbert et al. 2017) referring to the nested pattern of species distributions with different drought tolerances, which could increase forest resilience to drought. This 2019 paper (indicated by the reviewer) is mentioned in another section 'Disturbance regimes', where we discuss changes in the forest: "For instance, field evidence from long-term monitoring sites across the Amazon forest shows that tree mortality rates are increasing in most sites, reducing carbon storage capacity ^{74,75}, while favouring the recruitment of drought-affiliated species ²³." (line 231)

Esquivel-Muelbert et al.. (2017). Seasonal drought limits tree species across the Neotropics. *Ecography*, 40(5), 618-629.

line 396. "Through diverse ecosystem management practices, humans transformed the Amazonian flora." Transformed? I am sure there is not evidence that they 'transformed' the Amazonian flora -- at most it was encouraging a few useful species to grow (not more than 10's to 100's out of 15,000 species), and the evidence for that is highly contested.

We agree that humans did not transform the forest everywhere and influenced all species. We now say "... humans partly modified the Amazonian flora".

line 405. This sentence: "However, consistent loss of Amazonian languages is causing an irreversible disruption of ecological knowledge systems, mostly driven by road construction, which could contribute to further reducing forest resilience." Can you explain what the mechanism is that connects the loss of languages leading to lower forest resilience is.

We explain this mechanism in the previous sentences (copied below). Linguistic diversity is correlated with ecological knowledge diversity, which is a critical mechanism for enhancing forest resilience, because only through their ecological knowledge, IPLCs are able to manage and protect the forests.

"Today, IPLCs have diverse ecological knowledge about Amazonian plants, animals and landscapes, which allows them to quickly identify and respond to environmental changes with mitigation and adaptation practices ^{109,110}. IPLCs defend their territories against illegal deforestation and land-use disturbances ^{77,125}, and they also promote forest restoration by expanding diverse agroforestry systems ^{134,135}." (line 408)

line 425, section on global warming. It is very model-focused. You could describe the space-for-time study of Sullivan et al. 2020 Science, which show an abrupt loss of carbon above a max temp of 32 deg C.

Thank you for this suggestion. Sullivan et al. 2020 show that 32.2 °C may be a threshold for carbon storage capacity. This is why we discuss their findings (ref 47) in the section 'Regional climatic conditions', where we say: "Maximum temperatures during the dry season follow a similar trend, rising across most of the biome (Extended Data Fig. 1), exposing the forest ⁴⁷ and local peoples ⁴⁸ to potentially unbearable heat. For instance, rising temperatures will increase thermal stress, potentially reducing forest productivity and carbon storage capacity ⁴⁹ and causing widespread leaf damage ⁴⁷, particularly if combined with drier conditions ⁵⁰." (line 181). In the section 'Global warming', we focus on the evidence for a critical threshold for forest persistence that comes basically from modelling studies.

line 576, Modelling section. You ought to note that is it the mortality in tropical forests and the woody turnover that ESMs perform poorly with, rather than the growth part of the simulation, which they do well at, when compared to forest data.

Koch et al (2021) Earth System Models Are Not Capturing Present-Day Tropical Forest Carbon Dynamics. *Earth's Future*, 9, e2020EF001874.

David Galbraith et al. (2013) Residence times of woody biomass in tropical forests, *Plant Ecology & Diversity*, 6:1, 139-157.

Indeed, this needed to be better highlighted. We now say: "Forest functioning and dynamics start and scale up from the tree individual level, implying that this spatial and mechanistic resolution is desirable as an improvement for ESMs to truly test for Amazon forest tipping behaviour (e.g. ^{98,99,117}). Simulating tree individuals will improve the representation of growth and mortality dynamics, related for instance to species competition and physiological mechanisms, which strongly affect forest dynamics. In particular, individual tree mortality

dynamics needs to be accounted for as this stands at the beginning of any potential forest dieback.” (line 589).

Referee #3 (Remarks to the Author):

Thank you for inviting me to re-reviewing the paper “Tipping points in the Amazon forest system”.

I do remain convinced that this manuscript provides a state-of-the-art review of the risk of Amazon dieback, and is highly accessible to a broad set of readers. The quality of the display items is superb.

I want to thank the authors for taking the reviewer comments seriously, and I can see that there are comprehensive answers to both my suggestions, and the suggestions from the other five reviewers. However, in some instances, the responses are just to the reviewers – there are a few suggestions made that I think could support the manuscript if used to enhance the paper itself.

Thank you for another careful review and constructive comments. We tried to incorporate these suggestions in section ‘Prospects for modelling Amazon forest dynamics’.

It is my review that has triggered the new paragraph “Prospects for modelling Amazon dynamics”, and this addition is appreciated. The new paragraph is good, but it could gain from a little further improvement, and noting that it is currently incomplete. The paragraph describes well the need for explicit modelling, in ESMs, of forest functioning, forest dynamics and key physiological responses. However, it is also important to note the need for refinement of ESMs in projecting climate change, which they were originally designed to achieve (i.e. when they were Global Circulation Models).

We agree and have adapted the section.

This should not take the authors long to achieve, but please improve that paragraph by:

(1) Continue to state the importance of adding to ESMs representation of “plant community diversity and adaptation”.

We currently say: “While individual based simulations with ESMs will most likely be possible in the future, short term development should focus on how to incorporate the lessons learned from tree individual based models. Increasing the amount of functional diversity, representation of key plant physiological trade-offs, and allowing for community adjustment to environmental heterogeneity and global change will have significant impact on simulation results^{84,99,117,171}.” (line 596)

(2) The paper Abstract mentions climatological concerns – “warming temperature, extreme droughts...”. So although obvious, to really determine when tipping points will occur, and to build on this Flores et al review, we do need accurate predictions of future climate change. Unfortunately, ESMs differ in their projection of Amazon warming rates, and expected changes in rainfall patterns. This needs to be mentioned in this “next steps” Section.

That is a very good point. Please see response below about ESMs improving their predictions of future climatic conditions.

(3) Please mention methods are needed to reduce uncertainty in the projection of climate change. This can build on the Table / Schematic of Figure 3. The two ways are: (1) simply look at the performance of models for the contemporary period and (2) emergent constraints (ECs). In the response, the authors note the key issue that ECs fail if all models do not contain a key process affecting the Amazon. I would state that in this paragraph – so something like: “The emergent constraint method will fail if all ESMs are missing a common feature affecting the Amazon and our review helps to identify these underrepresented or currently ignored processes”.

Thank you for suggesting this extension, building on what we have incorporated for ESMs. We have now included the following sentences: “This requires improvements in the ability of ESMs to predict future climatic conditions, reducing uncertainties in the Amazon region. One way of doing so is to constrain ESM ensembles ¹⁷² to more accurately link future greenhouse-gas levels with the critical thresholds and feedbacks of Fig. 3. Other methods include the application of factorial simulations to account for the interactions between the various factors that shape Amazon forest resilience. However, none of these methods will improve ESMs if they are still missing key features affecting the Amazon forest. In sum, although our study may not deliver a set of reliable and comprehensive equations to parameterise processes impacting Amazon forest dynamics, required for implementation in ESMs, we highlight many of the missing modelled processes.” (line 616).

(4) Check that the paragraph reads tightly, and I think in places could be more positive. For instance, remove the words “not even” (line 589) and lines 605-607, maybe “Required for implementation in ESMs is the need for a set of reliable and comprehensive equations to parameterise processes impacting land dynamics. Our study points to many of these missing modelled processes by providing state-of-the-art.....”

We deleted the ‘even’ from that sentence. For the last sentence, please see our response to comment (3).

My comment concerning the need for factorial ESMs I would still like to be mentioned as a future aspiration for climate researchers. As is clear throughout the manuscript, tipping points often involved feedbacks between processes. Hence land surface “offline” simulations, which although easier to compute, may not be sufficient to determine the full role of proposed new mechanisms. Please just add a sentence somewhere to note that – doing so will make this a paper cited by those arguing for more computer power to operate ESMs.

We included a sentence addressing this point. Please see our response to comment (3).

I am happy to review any further manuscript iteration.

(One small bit of advice to the authors for any future submission! It helps the reviewer a lot at the second revision stage if it is clear what amendments have been made to the original paper. Either cite new text in the response document or if allowed by the journal, provide a track-changes version).

Thank you. We will follow the suggestion when submitting our revision.

Referee #4 (Remarks to the Author):

General Comments on revision:

Reviewer summary: The manuscript presents a review of the state of the science and literature surrounding potential Amazon tipping points alongside analysis to visualize tipping points and stable states responding to water conditions of rainfall, water deficit and dry season length. The authors place recommended values on five tipping points with associated confidence and studies supporting these recommendations. Crossing these tipping points are associated with potential ecosystem trajectories that are further presented with associated references and descriptions supported by the literature. The authors provide markers for safe operating space to maintain resilience of the Amazon forest.

Article contribution and overall impact: This manuscript adds value to an important conversation. The authors have done a strong job of responding to the feedback of such diverse reviews, and the revision addresses the concerns. The challenges facing the Amazon are diverse and of interest to broad groups with very different specializations. The authors provide a manuscript that highlights these challenges by including an extensive list of relevant references across disciplines. The new structure is quite different from the first submission, and would benefit from some tightening of the language in the early sections. Detail comments are provided below.

Thank you. We tried to tighten the language, reduce length and improve the flow following the detailed suggestions below.

Detailed comments:

Line 88: Update this to “at which disturbance may cause...”

We now say: “a small disturbance may cause an abrupt shift in the ecosystem”, to be clear that it doesn’t need a severe disturbance to cause a shift at the tipping point, because the ecosystem has very low resilience. This is how Lenton et al. (2008) define tipping point: “The term “tipping point” commonly refers to a critical threshold at which a tiny perturbation can qualitatively alter the state or development of a system.”

Line 121-122 (Box 1): This is not clear. Reword to clearly state the mechanism you are referring to for fire. Closed canopy forests depress fire. Suggest adding a phrase identifying the mechanism (canopy loss promotes fire which is a positive feedback on further canopy loss, etc.).

We deleted these sentences describing examples of feedbacks, because they are already described in Extended Data Table 1 (copied below), very similarly to the suggestion above.

“Forest cover supresses fire. Fire kills trees, decreasing forest cover and increasing ecosystem flammability.”

Line 186-210: This is a good section, but a bit hard to follow the flow of ideas and their connections. Try to clear this up.

Thank you. We agree that this part has dense information and tried to improve the flow. Please see new version copied below.

“Since the early 1980s, rainfall conditions have also changed. Peripheral and central parts of the Amazon forest are drying significantly ⁴¹, such as in the southern Bolivian Amazon, where annual rainfall reduced by up to 20 mm/year (Extended Data Fig. 2a). In contrast, parts of the western and eastern Amazon forest are becoming wetter, with annual rainfall increasing by up to + 20 mm/year. If these trends continue, ecosystem stability (as in Box 1) will likely change in parts of the Amazon by 2050, reshaping forest resilience to disturbances (Fig. 1b; Extended Data Fig. 2b). For example, 6 % of the biome may change from stable forest to a bistable regime in parts of the southern and central Amazon. Another 3 % of the biome may pass the critical threshold in annual rainfall into stable savanna in the southern Bolivian Amazon. Bistable areas covering 8 % of the biome may turn into stable forest in the western Amazon (Peru and Bolivia), thus becoming more resilient to disturbances. For comparison with satellite observations, we used projections of ecosystem stability by 2050 based on CMIP6 model ensembles for a low (SSP2-4.5) and a high (SSP5-8.5) greenhouse-gas emission scenario (Extended Data Table 2; Extended Data Fig. 3). An ensemble with the five coupled models that include a dynamic vegetation module indicates that 18 – 27 % of the biome may transition from stable forest to bistable and that 2 – 6 % may transition to stable savanna (depending on the scenario), mostly in the north-eastern Amazon. However, an ensemble with all 33 models suggests that 35 – 41 % of the biome could become bistable, including large areas of the southern Amazon. This difference between both ensembles is possibly related to the forest-rainfall feedback, included only in the five coupled models, which increases total annual rainfall along the southern Amazon and hence the stable forest area when deforestation is not included in the simulations ^{4,51}. Nonetheless, both model ensembles agree that bistable regions will expand deeper into the Amazon, increasing the risk of tipping events due to disturbances (as implied by the existence of alternative stable states; see Box 1).” (line 187).

Line 208-210: Suggest adding a short sentence explaining why bistability is more likely to increase tipping events. (This comes in later in the text, but suggest having some explanation here.)

We now say: “Nonetheless, both model ensembles agree that bistable regions will expand deeper into the Amazon, increasing the risk of tipping events due to disturbances (as implied by the existence of alternative stable states; see Box 1).”

Line 219: Provide a brief definition of edge effects and their importance. This is referenced in regards to flammability in the later modeling section, but define the importance of edge effects here as well. For instance, edges contribute to degradation through changes in microclimate which increasing drying and vulnerability to fire. There are multiple references in the manuscript that can inform the importance of edges (Brando et al 2014; Alencar et al 2015; Brando et al 2020; Brando et al 2019; Uhl et al)

Thank you for this suggestion. We now say: Within the remaining Amazon forest area, 17 % has been degraded by human disturbances ⁵⁷, such as logging, edge effects and understory fires, but considering also the impacts from repeated extreme drought events in the past decades, 38 % of the

Amazon could be degraded⁵³. Logging and edge effects increase forest flammability through changes in fuel loads and microclimate⁵⁸⁻⁶⁰, while road networks (Fig. 1d) facilitate illegal activities, promoting deforestation and fire spread throughout the core of the Amazon forest⁶¹⁻⁶⁴". (line 220)

Line 238-243: Reword this section. It reads as a list which detracts from the impact of combining this information for policy. There is a higher impact statement that can come out of this.

We tried to clarify the purpose of each layer used. We now say: "As bistable forests expand deeper into the system (Fig. 1b; Extended Data Fig. 3), the distribution of compounding disturbances indicates where ecosystem transitions are more likely to occur in the coming decades (as the possibility of forest shifting into an alternative structural or compositional state) (Fig. 1f). For this, we combined spatial information on warming and drying trends, repeated extreme drought events, together with road networks, as proxy for future deforestation and degradation^{61,62,64}. We also included protected areas and Indigenous territories as areas with high forest governance, where deforestation and fire regimes are among the lowest within the Amazon⁷⁷ (Fig. 1e). This simple additive approach does not consider synergisms between compounding disturbances that could result in positive (destabilizing) feedbacks, potentially leading to abrupt ecosystem transitions. However, by exploring only these factors affecting forest resilience and simplifying the enormous Amazonian complexity, we here aimed to produce a simple and comprehensive map that can be useful for guiding future governance." (line 237)

Line 245-248: Add a sentence regarding the uncertainty of using an additive approach. By adding the disturbance you account for the increased likelihood, but you do not account for feedbacks or potential changes that may act as a negative if one disturbance decreases the chance of another. Cumulative and compound disturbances are neglected in this additive approach.

Indeed, we use this very simple additive approach because we still lack information on the synergisms between the disturbances that could result in positive feedbacks, with non-linear responses, leading to abrupt ecosystem shifts. The strength of the feedbacks will naturally vary across the Amazon, which makes it particularly challenging to incorporate in this type of approach. So due to this tremendous complexity, we opted for a simple and more straightforward approach, that can be later sophisticated when more information is available.

We now say: "This simple additive approach does not consider synergisms between compounding disturbances that could result in positive (destabilizing) feedbacks, potentially leading to abrupt ecosystem transitions. However, by exploring only these factors affecting forest resilience and simplifying the enormous Amazonian complexity, we here aimed to produce a simple and comprehensive map that can be useful for guiding future governance."

Line 261: Update to "to increase the photosynthetic rates..."

Done

Line 261-279: The CO2 fertilization section reads well.

Thank you.

Line 326: Suggest rewording this to “drought due to historical pressures, which allows them...” and consider providing a time context (decades, centuries, etc)

We now say: “In contrast, most seasonal forests have various strategies to cope with water stress due to evolutionary and adaptive responses to historical drought events over the past centuries or millennia ^{100–103}. These strategies may allow seasonal forests to resist current levels of rainfall fluctuations ⁹⁶, but seasonal forests are also closer to the critical rainfall thresholds (Box 1) and may experience unprecedented water stress in the coming decades (Fig. 1).”

Line 339: Add details about why the system is increasingly homogenous. Even stand age of forests recovering?

We now say; “Moreover, as human disturbances intensify throughout the Amazon (Fig. 1), the spread of invasive grasses and fires can make the system increasingly homogeneous.” (line 341)

Line 358: Add illegal logging to this list with fire and overhunting

Done.

Line 402-403: Update to “Amazonian regions with the highest linguistic diversity...”

Done.

Line 423: Update this to include the purpose of this safe operating space. Such as: “...safe operating space for the continued resilience of the Amazon forest.”

We adapted to: “...we propose specific boundaries of these drivers ^{15,16} that define a safe operating space for keeping the Amazon forest resilient.”

Line 433: Update to “...results about tipping behavior...”

Done.

Line 505: Update to “we identify the three most plausible...”

Done.

Line 548-549: Consider updating the sentence to indicate that it is unlikely for these forests to shift back to forest once they transition to white-sand savanna. Or add a short sentence clarifying this point.

We adapted the sentence to say: “Shifts from forest to white-sand savanna are likely stable (i.e. the ecosystem is unlikely to recover back to forest within centuries or millennia), based on the relatively long persistence of these savannas in the landscape ¹⁶⁷.”.

Line 566-67: Update to include a phrase identifying that deforestation is associated with reduced moisture.

We now say: “At the regional scale, interaction between deforestation and atmospheric moisture flow may result in less rainfall and reduced forest resilience ⁵⁵.”

Figures:

Figure 3: Line 498-500: Suggest updating caption to include the name of the drivers rather than just the number.

Done.

Referee #6 (Remarks to the Author):

This manuscript is much improved from its previous submission. It is more clearly presented as a review, and the avoidance of its analyses of the possible confounding effects of land use have been clarified. I only have three remaining points to address:

(1) The MODIS VCF has nonuniform accuracy across the range of tree cover, with tree-cover values in the "savanna" range being among the weakest. Note also the saturation evident in Box 1 of values above 80% cover, which is due to a persistent artifact in the original response design that may or may not have remained unresolved.

See Figure 4 in Sexton, J.O., X.-P. Song, M. Feng, P. Noojipady, A. Anand, C. Huang, D.-H. Kim, K.M. Collins, S. Channan, C. DiMiceli, J.R. Townshend. 2013. Global, 30-m resolution continuous fields of tree cover: Landsat-based rescaling of MODIS continuous fields and lidar-based estimates of error. *International Journal of Digital Earth* 6(5): 427-448.

The authors should find a more recent validation of the MODIS VCF if one exists and provide at least one statement in the paper explaining the resulting possibility of spurious effects, as a caveat on their analyses.

We have carefully read the suggested paper by Sexton et al. (2013), as well as an exchange of other papers (Hanan et al. 2014; Staver and Hansen, 2015; Gerard et al. 2017) debating the usefulness of MODIS VCF for assessing forest-savanna bistability. Indeed, MODIS VCF has caveats in the data that can limit its utility, but this will depend on the research question. For instance, Staver and Hansen (2015) conclude that "MODIS VCF – which has facilitated major steps in our ability to examine ecological phenomena at global scales – remains a useful tool for well informed ecological analysis." They further detail: "(1) the MODIS VCF product may not be useful for differentiating over small ranges of tree cover (less than c. 10%); (2) that the bimodality of low and high tree cover, with a frequency minimum at intermediate tree cover, is not attributable to bias in MODIS VCF tree-cover calibrations; and (3) that the MODIS VCF is not well-resolved below c. 20–30% tree cover, such that MODIS cannot be used with any confidence to evaluate multimodality in tree cover in that range." As pointed out by Staver and Hansen (2015) – "Comparison of frequency distributions of tree cover from MODIS products versus validation data again suggest no systematic bias over intermediate tree cover (50–75% tree cover, at the boundary between savanna and forest), where frequency minima have been observed globally." Their interpretation is rather similar to the one provided by Sexton et al. (2013): "The MODIS VCF tree layer, now in version 5, is among the most useful and reliable global datasets representing Earth's woody vegetation. However, it suffers from a coarse spatial scale relative to many land cover patterns and changes ... Much of this uncertainty is

systematic, due to over-estimation in sparsely treed (e.g. agricultural) regions and under-estimation in dense forests.”

Other studies provide support to this interpretation. For example, reanalysis of field data on total canopy cover collected from forest-savanna transition zones across the tropics, including Africa, South America and Australia (Figure 2 below; from Staal and Flores, 2015) tested for multimodality using latent class analysis (as in Hirota et al. 2011, Science), and also found a clear bimodality in this field dataset on canopy cover.

Figure 2. The probability density of total canopy cover measured in field sites (n = 41) are significantly bimodal, indicating bistability of forest and savanna (from Staal and Flores, 2015).

Given the potential caveats and debate around MODIS VCF, we now explain in the Supplementary Information section (lines 1173 ...): “Although MODIS VCF can contain errors within lower tree cover ranges and should not be used to test for bistability between grasslands and savannas, the dataset is relatively robust for assessing bistability within the tree cover range of forests and savannas (Staver and Hansen 2015).” We also included a new section ‘Validation of the MODIS VCF’ in the Supporting Information, and a Fig. S1 (copied below) showing the validation of the MODIS VCF data using a distinct dataset, the Landsat forest cover data from Hansen et al. (2013, Science). The validation confirmed the same bistability pattern and even the critical thresholds, which were remarkably similar the ones estimated using MODIS VCF.

[This has been redacted.]

Hanan et al. (2014). Analysis of stable states in global savannas: is the CART pulling the horse? *Global Ecology and Biogeography*, 23, 259–263

Staal, A., & Flores, B. M. (2015). Sharp ecotones spark sharp ideas: comment on "Structural, physiognomic and above-ground biomass variation in savanna–forest transition zones on three continents—how different are co-occurring savanna and forest formations?" by Veenendaal et al. (2015). *Biogeosciences*, 12(18), 5563–5566.

Staver and Hansen. (2015). Analysis of stable states in global savannas: is the CART pulling the horse?—a comment. *Global Ecology and Biogeography*, 24: 985–987.

2. The word "significant" is used multiple times in reference to Figure 1. To better match the text to the Figure, and to better convey the statistical certainty of the findings, all pixels whose linear regression slopes are not significant at some chosen (and stated) p-value should be masked from the figure. This can be done simply by making those pixels transparent in color.

Thank you for calling our attention that the explanation was not clear enough in the text and figure legend. We tried to clarify (please see below).

In Figure 1a (Dry season warming trend) we show in a blue-red palette only pixels with significant slopes, whereas the non-significant slopes are indeed transparent. In the figure legend, we state our chosen p-values: "(a) Changes in the dry season (July-October) mean temperature reveal widespread warming, estimated using simple regressions between time and temperature observed between 1981 and 2020 (with $p < 0.1$)."

Indeed, the map in Figure 1b (Stability classes by year 2050) is not as straightforward as in (a). We now improved the legend explanation: "(b) Ecosystem stability classes estimated for year 2050, adapted from current stability classes (Box 1b) by incorporating only the changes in areas with significant slopes, using simple regressions between time and annual rainfall observed between 1981 and 2020 (with $p < 0.1$) (see Extended Data Fig. 2 for areas with significant changes)." In the Extended Data Fig. 2 (copied below), we also show non-significant slopes in grey.

3. Anthropogenic conversion of forest to agriculture is currently a far more powerful driver of forest loss in the Amazon biome than climate change. The MapBiomas project as well as several papers have come out recently on the rates of forest loss in the Amazon. To give a complete accounting of the region, and to put the climate and land use effects in perspective, the review must add a paragraph simply giving the rates of forest loss in the region due to land-use change.

We agree with the reviewer that land-use changes are causing more severe disturbances on the forest, compared to climatic changes (although both drivers strongly interact). We describe this process in a few places throughout the manuscript (please see example copied below), but we agree that it would be ideal to provide more details. Unfortunately, we are now already above the word limit for the manuscript and we are trying to reduce its size. Including a paragraph at this stage is complicated. Instead, we included a new Extended Data Fig. 4 (below) showing the rates of forest loss in the Amazon biome and where they are occurring in the most recent decade.

Line 218 (section 'Disturbance regimes'): "Rainfall is decreasing in southwestern and central parts of the Amazon possibly due to deforestation^{55,56}, which continues to expand into core areas of the system (Extended Data Fig. 4). Within the remaining Amazon forest area, 17 % has been degraded by human disturbances⁵⁷, such as logging, edge effects and understory fires, but considering also the impacts from repeated extreme drought events in the past decades, 38 % of the Amazon could be degraded⁵³. Logging and edge effects increase forest flammability through changes in fuel loads and microclimate⁵⁸⁻⁶⁰, while road networks (Fig. 1d) facilitate illegal activities, promoting deforestation and fire spread throughout the core of the Amazon forest⁶¹⁻⁶⁴. Fire can be a severe disturbance^{65,66}, usually associated with deforestation and cattle production areas from where it may escape into standing forests in drier years^{67,68}. New fire regimes are burning larger forest areas^{69,70}, emitting more carbon to the atmosphere^{71,72} and forcing IPLCs to readapt⁷³."

Extended Data Fig. 4. Deforestation continues to expand within the Amazon forest system. (a) Map highlighting deforestation and fire activity between 2012 and 2021, a period when environmental governance began to weaken again, as indicated by increasing rates of annual deforestation (b).

Reviewer Reports on the Second Revision:

Referees' comments:

Referee #1 (Remarks to the Author):

The authors have done a good job of addressing the points raised in previous reviews. I've enjoyed the iterations of this manuscript and seeing its evolution--it is an important contribution, highly informative, and brings fresh and comprehensive view to the topic of tipping points.

There are two small areas that I think require some thought, easily addressed, that are raised below in the specific comments. The first is the inclusion of high Andean puna, paramo and forests as Amazonian without clearly telling the reader (it has consequences for interpretation), and the second is not excluding already deforested and agricultural lands from figures talking about the stable ecosystem classes, but rather labeling those as forest or savanna.

L43: Rather than get that specific with a number in the abstract, perhaps, "Throughout the Cenozoic...."?

L63: It would be good here to explicitly tell the reader what the definition of Amazonia is for this paper, and that you are going through the entire Amazon drainage from the high Andean paramo through the lowlands, and including the Orinoco drainage and forests of the Guyanan highlands. When I think of the Amazon forest I tend to think of the lowlands, below 500m, and when I think of the Basin it excludes the non-Amazonian drainages, or explicitly includes them in greater Amazonia. As used here it is a bit of a mash-up and needs a bit more than just the few words at the top of box 1. Even expanding that would be good.

L121: To this end, the stable savanna is almost exclusively found in the high Andes, outside of the Amazon biome. Within the Amazon biome stable savanna would be a fraction of a percent--basically nonexistent. That is a neat result!

L156, section: I had raised this in an earlier review, but I think it is worth saying that perhaps the biggest tipping point in the Amazon is governance as some of the authors in this manuscript showed in Nature last month. Really, the entire biome is bi-stable depending on what policy is, and social tipping points are every bit as powerful as climate tipping points. Not something to put here, but a point to make in the discussion.

L285, Figure 1 B, etc. I still have a problem with a figure showing that the stable ecosystem class for much of the Amazon is "forest" when a large proportion of that has already been deforested. The stable state for agriculture in what used to be southern Amazonian forest is not forest, but agriculture. Should this be re-labeled "Potential Ecosystem Stability Classes" or something like that? Or, better still, I would mask out all land uses that are not currently forest or savanna. It is deceptive to see what looks like vast unbroken habitat, when the reality, as the authors state, is that much of those habitats have been converted or degraded. Soybean and pasture are large ecosystem states in Amazonia.

L417: I like this section, but might add, "...in the face of anthropogenic global change." The Amazon has been around for 65MY, and for ~99.98% of those years there were no humans and the forest was highly resilient. For reasons having nothing to do with humans. IPLCs are centrally important now and into the future, but not because they have improved Amazonian forest per se, but rather provided human solutions to human problems and will do so in even more important ways in the near-term future.

Referee #2 (Remarks to the Author):

The new version of the manuscript is further improved. I appreciate the efforts of the authors in responding to reviewer comments. This should be published.

I only have one further comment. The title reads slightly oddly in the singular, and given the changes to the manuscript, improved terminology, and that concerns some scientists have about 'tipping points' as a potentially loaded term, I wonder if "Critical water thresholds for the Amazon forest system" is a better title, and one that might age better.

Referee #3 (Remarks to the Author):

I really like this paper, Nature is an appropriate journal, and I think it is now very near to being accepted.

The paper revisions are good, but don't quite capture what I was suggesting when discussing ESM development. However that might have been in part my fault with a lack of clarity in the request from me.

To get round this, below is text that tightens the current version and the authors might like to consider as a revision (or something very similar). Other than that, I am happy if the paper is now accepted.

"Besides enhancing simulations of Amazon forest response, improvements in the ability of ESMs to predict future climatic conditions are also required. One way is to identify emergent constraints [Ref 172] applicable to the Amazonia climate to lower inter-ESM differences in projections. Eventually, fully coupled ESM simulations are needed that include an accurate depiction of the land surface. Yet introducing new land surface descriptions to ESMs will revise estimates of land-atmosphere feedbacks, which may adjust calculated near-surface meteorology and ecosystem response. Hence, to fully understand the thresholds for Amazonian resilience, how these link to different meteorological changes and, in turn, link to future raised atmospheric GHG concentrations may require some form of factorial simulations with ESMs. Undertaking many simulations with ESMs remains a challenge compared to "offline" calculations due to their computational requirement but have the advantage of modelling explicitly evolving land-atmosphere feedbacks. However, none of these methods will improve ESM-based projections of Amazon forest changes if they are still missing key features affecting the land surface."

Referee #4 (Remarks to the Author):

Revision of Title: Tipping points in the Amazon forest system

Authors: Bernardo M. Flores, Encarni Montoya , Boris Sakschewski, Richard Betts, Nathália Nascimento, David M. Lapola, Adriane Esquivel-Muelbert, Catarina Jakovac, Carlos A. Nobre, Arie Staal, Carolina Levis, Rafael S. Oliveira, Laura S. Borma, Da Nian, Niklas Boers, Susanna B. Hecht Hans ter Steege Julia Arieira, Isabella L. Lucas, Erika Berenguer, José A. Marengo, Luciana V. Gatti & Marina Hirota

General Comments on revision:

I appreciate that the authors have been given time to address the many and diverse reviewer comments. It has been a lot of effort on their part. The authors have made extensive efforts to address the reviewer comments. I feel they have been addressed, and only have minor suggested changes to this version. The manuscript offers an extensive review and detailed safe space operating suggestions that are valuable to the policy debate around the future of the Amazon.

Detailed comments:

Line 222-223 Update this to “but, if the impacts from repeated extreme drought events in the past decades are also considered, 38% of the Amazon...”

Line 489-490: Update this to “found that with accumulated deforestation of only 30-50%, rainfall in non-deforested...”

Line 590: Update to “scale up from the individual tree level...”

Line 592: Update to “Simulating individual trees will improve...”

Line 598 Update to “individual tree-based models” or “individual based models of trees” The current wording is awkward. I understand that these models are typically abbreviated as IBMs (individual based models) so I see why the latter suggestion may make sense here.

Line 604: Remove instantly. Update to “it also becomes essential to account...”

Referee #6 (Remarks to the Author):

The manuscript has been much improved in both content and clarity. Masking the insignificant areas from the trends maps greatly improved the interpretability of the maps especially. The writing still meanders in several places; I have made comments to clarify it wherever obvious (and possibly opening room for more explicit inclusion of the relative importance of deforestation as a driver of forest loss, but I will leave the rest to the copy editors.

I recommend publication in Nature after a few points are addressed.

Given the comments by the other referees, I recommend the authors change the title to “An ecological tipping point in the Amazon forest system”.

Much of the conversation with the other referees has focused on whether there are one or multiple tipping points. Taken as a whole, the complete review points to the conclusion that there is one tipping point, its primary eco-physiological mechanism is water balance, and it might be approached by the Amazon ecosystem from multiple dimensions. The title cannot contain all the information of the rest of the article, and the addition of the indefinite article “a” and adjective “ecological” should suffice for the purpose of a title.

Remove the new MODIS/Landsat VCF validation section and replace it with a statement about the imprecision/uncertainty of the MODIS VCF and its potential impact on the findings and either (a) a map of the uncertainty (posterior standard deviation) of the MODIS VCF tree-cover estimates across the region, or (b) a true independent validation against GEDI or other high-quality reference data. The authors validate the MODIS-based VCF Tree Cover dataset (DiMiceli et al.) using the Landsat-based Tree Cover dataset from the University of Maryland (UMD) Global Land Analysis and Discovery (GLAD) dataset (Hansen et al. 2013), which the authors have named after the Global Forest Watch platform of World Resources Institute, the nonprofit organization which distributes it. However, these datasets are not independent. The original MODIS VCF dataset was developed by Matt Hansen, Charlene DiMiceli, and several others at the University of Maryland in the early 2000’s, when it was published under Hansen as lead author. Production and ongoing development were subsequently taken over by Dimiceli, who has done so up to the current version (6.1). When the USGS Landsat archive became free in 2009, the algorithm for Hansen’s (2013) “Landsat VCF” was statistically trained using the MODIS VCF as reference data. That is, although they are both correlated to actual tree cover, the extremely high correlation the authors found between the Dimiceli MODIS and Hansen Landsat tree-cover datasets is either spurious or tautological—the latter is mostly a statistical downscaling of the former.

Unfortunately, identification of the problem does not provide a solution. The Sexton et al. (2013) dataset is also of the same family, coming from the University of Maryland at that same period. Although it greatly increased the statistical rigor of its family of data products with independent validation against lidar samples and mapping of posterior uncertainty, it too was originally trained on the MODIS VCF as the primary source of reference. The Sexton et al. (2013) paper shows that triad of correlations between MODIS-, lidar-, and their own Landsat-based tree cover maps, and its approach for communicating uncertainty in satellite data products was adopted in later version of the MODIS VCF.

The interdependence and evolution of global datasets is a challenge, but they are immensely informative, and their shortcomings (which are small compared to scattered, inconsistent samples of in situ data) cannot be held as an absolute barrier to inference. I recommend the authors include, along with a statement about potential impact on results, a map of the posterior uncertainty (RMSE, i.e., standard deviation of the estimate) available at (<https://lpdaac.usgs.gov/products/mod44bv061/>).

If the authors wish to surpass this and provide a true independent validation of the MODIS VCF, to my knowledge, the most independent and reasonably accurate global dataset on tree cover currently available is the GEDI Canopy Cover data (https://lpdaac.usgs.gov/products/gedi02_bv002/), of which gridded versions might have become available. These are based on lidar measurements, which suffer from uncertainty due to cloud cover

in the tropics—which is itself not stationary against tree cover for exactly the ecological reasons the authors have elucidated. Although valuable, this is no small task, and I am not recommending the authors become remote sensing scientists. Again, the most reasonable option, should the authors accept, it is a map of the VCF's posterior uncertainty with a statement explaining the sensitivity of their results on it.

Also, I do not see a reference to the version of the MODIS VCF used by the authors. They cite Version 5, but the current version is 6.1.

The main findings of the paper depend on this dataset. Please document the version used.

Whereas most of the safe boundaries are justified, the 1.5 degree Celsius boundary for global warming is not. Why this value?

Line 451.

Line-by-line comments:

41: delete “address this issue by gathering state of the art knowledge about the various” and replace it with “review”

43-35: Non-sequitur: The first part of the sentence is about the biological components of the ecosystem, but the second part of the sentence is about environment.

50: Find a better word than “systemic”. Systemic is not a gradation of local and regional. Consider “biome-wide”.

61: The first sentence of the Introduction can be removed without loss of information from the section.

73 and thereafter: “indigenous” is an adjective, not a proper noun. Its arbitrary capitalization is a contrivance that is disrespectful to the actual (proper) tribes, cultures, and civilizations indigenous to regions around the world.

78: Start new paragraph at “Understanding...”.

88: Delete “constantly”.

90: Redundant. Delete “between (two or more) equilibrium states (also referred to as”.

109: Replace “are able to assess” with “assessed”.

126: The last sentence of this paragraph has been said previously. It can be deleted here.

133: The second clause of the sentence, beginning “but the...” should be moved to the first sentence of the paragraph as an independent sentence.

142: Change “on” to “by”.

160: Remove the hyphen from “CO2-fertilization”.

170: Remove “For instance” and (compared to pre-industrial conditions).

171: Delete the comma.

212: The first two sentences of this paragraph need not be said and should be deleted.

217: Delete “together”, and replace “that result” with “resulting”.

225: Delete “illegal activities, promoting”. This is about deforestation, not its legality.

233: Delete “capacity”. This is about carbon storage, not the capacity to do so.

239: Delete the parenthetical clause. This has been said many times already in the manuscript.

245: Change “positive (destabilizing)” to “destabilizing positive”.

267: Change “buffer the effect of” to “reduce”.

270: Delete “rates (trees grow faster but also die earlier)”.

279: Delete “even”.

283: Delete the comma and “flow”.

309: This section confuses species or biological diversity with environmental diversity or heterogeneity. Biodiversity is a mechanism of resilience, but environmental heterogeneity can either accelerate or decelerate ecosystem responses—depending again on the mix of species or functional types.

310: Change “systemic” to “biome-wide” or equivalent. In this context, systems are not scale-specific.

313: Delete “, in particular,”. (Or at least delete the commas.)

317: Delete “In theory”.

329: Forests do not have strategies. Change to “responses” or “mechanisms”.

338: Replace “plateau” with “upland”?

348: This section gets the biodiversity heterogeneity distinction right. However, it feels wedged in and might not be worth its own section. Consider merging with another section(s).

376: Move this section to later in the paper, between “Five drivers of water stress” and “Three alternative ecosystem trajectories”.

379: Delete “enhancing”.

420: Delete the first sentence of this paragraph and section.

423: Replace “global” with “atmospheric”.

425: Delete “, for instance”.

426: Replace “controls” with “limits”.

437: Delete “For instance”.

440: Delete “a few”.

443: Delete “undoubtedly”.

467: Replace “Rainfall seasonal intensity” with “Seasonal rainfall intensity”.

468 (and elsewhere): There is no such thing as satellite tree cover, nor of observations of satellite tree cover. Replace “Observations of satellite tree cover” with “satellite observations of tree cover” here and throughout the manuscript.

483: Replace “simulated” with “simulating”; delete “different”, “fire”, “events”, and “. The result”; insert a comma in “deforestation beyond”; delete “consequently” and “depending on the emissions scenario”.

491: Delete “disturbances”.

503: Delete “We identified”.

514: Delete the parentheses and “that are”.

516: Delete the comma after “feedbacks” and “to”.

518: “depending on the outcome of the novel interactions” is tautological. Delete the phrase.

527: Delete the parentheses.

561: Delete “(low tree cover)”.

569: Delete the comma after “state”.

584: Delete “We identified” and “that”. Make the sentence about the subject, not about the author.

586: Over-explaining. Delete “the” and “that could lead to a tipping point related to water stress”.

488: Replace “need to” with “must” or “should”.

595: Delete “dynamics”.

598: Delete “the amount of”.

601: Delete “a “.

630: Delete "hot-".

634: Delete "forms of".

644: Replace the parentheses with commas in "(99% of which is illegal)".

Author Rebuttals to Second Revision:

Referees' comments:

Referee #1 (Remarks to the Author):

The authors have done a good job of addressing the points raised in previous reviews. I've enjoyed the iterations of this manuscript and seeing its evolution--it is an important contribution, highly informative, and brings fresh and comprehensive view to the topic of tipping points.

There are two small areas that I think require some thought, easily addressed, that are raised below in the specific comments. The first is the inclusion of high Andean puna, paramo and forests as Amazonian without clearly telling the reader (it has consequences for interpretation), and the second is not excluding already deforested and agricultural lands from figures talking about the stable ecosystem classes, but rather labeling those as forest or savanna.

Thank you for the constructive iterations and many inputs to the manuscript.

We now explicitly say in the Methods, section 'Datasets': "Our study site was the area of the Amazon basin, considering large areas of tropical savanna biome along the northern portion of the Brazilian Cerrado, the Gran Savana in Venezuela and the Llanos de Moxos in Bolivia, as well as the Orinoco basin to the north, and eastern parts of the Andes to the west. **The area includes also high Andean landscapes with puna and paramo ecosystems.**"

In Box 1b and Figure 1, as we explained in a previous exchange, we do not exclude deforested areas from the map because we want to show potential ecosystem stability classes for these areas. However, for plotting the scatterplots with alternative equilibria in Box 1, we did exclude deforested areas. Only when translating the stability classes to the map from empirical rainfall data, we ignore these deforested areas, for instance, to see if the 'arc-of-deforestation' has relatively less rainfall and more bistable area, compared to other parts of the Amazon. Also, when planning for restoration, it is useful to know the stability classification of the areas. Stable forests are expected to recover passively, whereas bistable forests may need intervention to recover. If we excluded deforested areas, we would miss this type of information.

L43: Rather than get that specific with a number in the abstract, perhaps, "Throughout the Cenozoic...."?

Because we are communicating with a broad range of non-specialist readers that may not understand the meaning of Cenozoic, we opted to avoid this terminology in the Abstract that can be easily replaced by plain language as it is in this case (For 65 million years). In the first sentence of the section 'Past dynamics' we included this suggestion: "The Amazon system has been mostly covered by forest throughout the Cenozoic (for 65 million years)."

L63: It would be good here to explicitly tell the reader what the definition of Amazonia is for this paper, and that you are going through the entire Amazon drainage from the high Andean paramo through the lowlands, and including the Orinoco drainage and forests of the Guyanan highlands. When I think of the Amazon forest I tend to think of the lowlands, below 500m, and

when I think of the Basin it excludes the non-Amazonian drainages, or explicitly includes them in greater Amazonia. As used here it is a bit of a mash-up and needs a bit more than just the few words at the top of box 1. Even expanding that would be good.

We agree with the reviewer that it is important to clarify this information. Since we were already a lot over the word limit, we maintained this detailed information in the Methods. In Box 1 we state only the information necessary for readers to understand that we included forest and savanna ecosystems within and in peripheral parts of the Amazon.

L121: To this end, the stable savanna is almost exclusively found in the high Andes, outside of the Amazon biome. Within the Amazon biome stable savanna would be a fraction of a percent--basically nonexistent. That is a neat result!

The reviewer is correct. In Box 1 this is what we would expect, because from current forest and savanna distributions, we obtain an estimate that savannas are stable basically where they occur, thus outside of the Amazon forest biome. It becomes more interesting in Fig. 1b, where we show that stable savannas expand inside the forest biome, suggesting that forests will collapse in those areas.

L156, section: I had raised this in an earlier review, but I think it is worth saying that perhaps the biggest tipping point in the Amazon is governance as some of the authors in this manuscript showed in Nature last month. Really, the entire biome is bi-stable depending on what policy is, and social tipping points are every bit as powerful as climate tipping points. Not something to put here, but a point to make in the discussion.

This is an interesting political perspective. There are studies investigating tipping points in social behaviour that could lead to sustainable practices. Here we focus on the drivers of stress on the system, and discuss how governance is critical for keeping them in a safe operating space. We certainly welcome studies addressing tipping points in the societal interest in supporting a stronger governance for the Amazon. Unfortunately, we do not have space to expand our text, but in the final section we discuss the importance of governance.

L285, Figure 1 B, etc. I still have a problem with a figure showing that the stable ecosystem class for much of the Amazon is "forest" when a large proportion of that has already been deforested. The stable state for agriculture in what used to be southern Amazonian forest is not forest, but agriculture. Should this be re-labeled "Potential Ecosystem Stability Classes" or something like that? Or, better still, I would mask out all land uses that are not currently forest or savanna. It is deceptive to see what looks like vast unbroken habitat, when the reality, as the authors state, is that much of those habitats have been converted or degraded. Soybean and pasture are large ecosystem states in Amazonia.

As we explained above, these areas may have been converted to agriculture, but the message here is that if they were abandoned from land use, these stable forest areas would likely recover without need for intervention, and this type of information can be extremely useful for restoration projects. We now say in the figure legend: 'Potential ecosystem stability classes...'

L417: I like this section, but might add, "...in the face of anthropogenic global change." The Amazon has been around for 65MY, and for ~99.98% of those years there were no humans and the forest was highly resilient. For reasons having nothing to do with humans. IPLCs are centrally important now and into the future, but not because they have improved Amazonian forest per se, but rather provided human solutions to human problems and will do so in even more important ways in the near-term future.

We now say: “Continued loss of ecological knowledge will undermine the capacity of IPLCs to manage and protect Amazonian forests, further reducing the ecosystem resilience to global changes.”

Referee #2 (Remarks to the Author):

The new version of the manuscript is further improved. I appreciate the efforts of the authors in responding to reviewer comments. This should be published.

I only have one further comment. The title reads slightly oddly in the singular, and given the changes to the manuscript, improved terminology, and that concerns some scientists have about 'tipping points' as a potentially loaded term, I wonder if "Critical water thresholds for the Amazon forest system" is a better title, and one that might age better.

We decided to maintain our title to “Tipping point in the Amazon forest system”, which we already changed to singular, addressing a comment from the same reviewer in a previous exchange. The term tipping point is being widely used by experts on the topic and other fields.

Referee #3 (Remarks to the Author):

I really like this paper, Nature is an appropriate journal, and I think it is now very near to being accepted.

The paper revisions are good, but don't quite capture what I was suggesting when discussing ESM development. However that might have been in part my fault with a lack of clarity in the request from me.

To get round this, below is text that tightens the current version and the authors might like to consider as a revision (or something very similar). Other than that, I am happy if the paper is now accepted.

"Besides enhancing simulations of Amazon forest response, improvements in the ability of ESMs to predict future climatic conditions are also required. One way is to identify emergent constraints [Ref 172] applicable to the Amazonia climate to lower inter-ESM differences in projections. Eventually, fully coupled ESM simulations are needed that include an accurate depiction of the land surface. Yet introducing new land surface descriptions to ESMs will revise estimates of land-atmosphere feedbacks, which may adjust calculated near-surface meteorology and ecosystem response. Hence, to fully understand the thresholds for Amazonian resilience, how these link to different meteorological changes and, in turn, link to future raised atmospheric GHG concentrations may require some form of factorial simulations with ESMs. Undertaking many simulations with ESMs remains a challenge compared to “offline” calculations due to their computational requirement but have the advantage of modelling explicitly evolving land-atmosphere feedbacks. However, none of these methods will improve ESM-based projections of Amazon forest changes if they are still missing key features affecting the land surface."

Thank you. We incorporated this text into the modelling section, replacing the last lines for the ones suggested, with a few adaptations to maintain word-count within the limit.

Referee #4 (Remarks to the Author):

Revision of Title: Tipping points in the Amazon forest system

Authors: Bernardo M. Flores, Encarni Montoya , Boris Sakschewski, Richard Betts, Nathália Nascimento, David M. Lapola, Adriane Esquivel-Muelbert, Catarina Jakovac, Carlos A. Nobre, Arie Staal, Carolina Levis, Rafael S. Oliveira, Laura S. Borma, Da Nian, Niklas Boers, Susanna B. Hecht Hans ter Steege Julia Arieira, Isabella L. Lucas, Erika Berenguer, José A. Marengo, Luciana V. Gatti & Marina Hirota

General Comments on revision:

I appreciate that the authors have been given time to address the many and diverse reviewer comments. It has been a lot of effort on their part. The authors have made extensive efforts to address the reviewer comments. I feel they have been addressed, and only have minor suggested changes to this version. The manuscript offers an extensive review and detailed safe space operating suggestions that are valuable to the policy debate around the future of the Amazon.

Thank you for the positive general comment and for these suggestions below.

Detailed comments:

Line 222-223 Update this to “but, if the impacts from repeated extreme drought events in the past decades are also considered, 38% of the Amazon...”

We adapted the sentence as suggested.

Line 489-490: Update this to “found that with accumulated deforestation of only 30-50%, rainfall in non-deforested...”

We do not mean to say that 30-50% is a small number, but actually high, compared to the study mentioned in the previous sentences. To avoid confusion, we deleted the word ‘only’.

Line 590: Update to “scale up from the individual tree level...”

We deleted this sentence to improve the flow of the section and to reduce word-count.

Line 592: Update to “Simulating individual trees will improve...”

Done.

Line 598 Update to “individual tree-based models” or “individual based models of trees” The current wording is awkward. I understand that these models are typically abbreviated as IBMs (individual based models) so I see why the latter suggestion may make sense here.

We also deleted this sentence for the reasons explained above.

Line 604: Remove instantly. Update to “it also becomes essential to account...”

We also deleted this sentence for the reasons explained above.

Referee #6 (Remarks to the Author):

The manuscript has been much improved in both content and clarity. Masking the insignificant areas from the trends maps greatly improved the interpretability of the maps especially. The writing still meanders in several places; I have made comments to clarify it wherever obvious (and possibly opening room for more explicit inclusion of the relative importance of deforestation as a driver of forest loss, but I will leave the rest to the copy editors. I recommend publication in Nature after a few points are addressed.

Thank you for these suggestions and for supporting the publication of our manuscript.

Given the comments by the other referees, I recommend the authors change the title to “An ecological tipping point in the Amazon forest system”.

Much of the conversation with the other referees has focused on whether there are one or multiple tipping points. Taken as a whole, the complete review points to the conclusion that there is one tipping point, its primary eco-physiological mechanism is water balance, and it might be approached by the Amazon ecosystem from multiple dimensions. The title cannot contain all the information of the rest of the article, and the addition of the indefinite article “a” and adjective “ecological” should suffice for the purpose of a title.

Thank you for this title suggestion. Although we agree with the interpretation that the tipping point is related to water stress for plants, we decided to maintain our original title “Tipping point in the Amazon forest system”. The tipping point we refer to is not only an ecological process, but involves also climatological and social processes as part of this complex system.

Remove the new MODIS/Landsat VCF validation section and replace it with a statement about the imprecision/uncertainty of the MODIS VCF and its potential impact on the findings and either (a) a map of the uncertainty (posterior standard deviation) of the MODIS VCF tree-cover estimates across the region, or (b) a true independent validation against GEDI or other high-quality reference data.

The authors validate the MODIS-based VCF Tree Cover dataset (DiMiceli et al.) using the Landsat-based Tree Cover dataset from the University of Maryland (UMD) Global Land Analysis and Discovery (GLAD) dataset (Hansen et al. 2013), which the authors have named after the Global Forest Watch platform of World Resources Institute, the nonprofit organization which distributes it. However, these datasets are not independent. The original MODIS VCF dataset was developed by Matt Hansen, Charlene DiMiceli, and several others at the University of Maryland in the early 2000’s, when it was published under Hansen as lead author. Production and ongoing development were subsequently taken over by Dimiceli, who has done so up to the current version (6.1). When the USGS Landsat archive became free in 2009, the algorithm for Hansen’s (2013) “Landsat VCF” was statistically trained using the MODIS VCF as reference data. That is, although they are both correlated to actual tree cover, the extremely high correlation the authors found between the Dimiceli MODIS and Hansen Landsat tree-cover datasets is either spurious or tautological—the latter is mostly a statistical downscaling of the former.

Unfortunately, identification of the problem does not provide a solution. The Sexton et al. (2013) dataset is also of the same family, coming from the University of Maryland at that same period. Although it greatly increased the statistical rigor of its family of data products with independent validation against lidar samples and mapping of posterior uncertainty, it too was originally trained on the MODIS VCF as the primary source of reference. The Sexton et al. (2013) paper shows that triad of correlations between MODIS-, lidar-, and their own Landsat-based tree cover maps, and its approach for communicating uncertainty in satellite data products was adopted in later version of the MODIS VCF.

The interdependence and evolution of global datasets is a challenge, but they are immensely informative, and their shortcomings (which are small compared to scattered, inconsistent samples of in situ data) cannot be held as an absolute barrier to inference. I recommend the authors include, along with a statement about potential impact on results, a map of the posterior uncertainty (RMSE, i.e., standard deviation of the estimate) available at (<https://lpdaac.usgs.gov/products/mod44bv061/>).

If the authors wish to surpass this and provide a true independent validation of the MODIS VCF, to my knowledge, the most independent and reasonably accurate global dataset on tree cover currently available is the GEDI Canopy Cover data (https://lpdaac.usgs.gov/products/gedi02_bv002/), of which gridded versions might have become available. These are based on lidar measurements, which suffer from uncertainty due to cloud cover in the tropics—which is itself not stationary against tree cover for exactly the ecological reasons the authors have elucidated. Although valuable, this is no small task, and I am not recommending the authors become remote sensing scientists. Again, the most reasonable option, should the authors accept, it is a map of the VCF's posterior uncertainty with a statement explaining the sensitivity of their results on it.

Thank you for providing this detailed explanation for why Landsat tree cover is not really a validation. We removed the validation using Landsat and now, as suggested, we provide a new Extended Data Fig. 7 with a map of the uncertainty (standard deviation) of the MODIS VCF tree-cover estimates across the region, obtained from the product website (<https://lpdaac.usgs.gov/products/mod44bv061/>).

Also, I do not see a reference to the version of the MODIS VCF used by the authors. They cite Version 5, but the current version is 6.1.

The main findings of the paper depend on this dataset. Please document the version used.

The original purpose of Box 1 was to illustrate the concepts of tipping point for general readers, but now it also shows how we estimated the three critical thresholds in rainfall conditions.

In the Methods, we now describe the dataset with reference and version: “We used the Moderate Resolution Imaging Spectroradiometer (MODIS) Vegetation Continuous Fields (VCF) data (MOD44B Version 6, <https://lpdaac.usgs.gov/products/mod44bv006/>) for the year 2001 at 250-m resolution ¹¹⁵ to reanalyse tree cover distributions within the Amazon basin, refining estimates of bistability ranges and critical thresholds in rainfall conditions from previous studies. Although MODIS VCF can contain errors within lower tree cover ranges and should not be used to test for bistability between grasslands and savannas ¹¹⁶, the dataset is relatively robust for assessing bistability within the tree cover range of forests and savannas ¹¹⁷, as also shown by low uncertainty (standard deviation of tree cover estimates) across the Amazon (Extended Data Fig. 7).”

Whereas most of the safe boundaries are justified, the 1.5 degree Celsius boundary for global warming is not. Why this value?

Line 451.

Thank you for calling attention to this. We now say: “To avoid broad-scale critical transitions due to synergisms between climatic and land-use disturbances, we suggest a safe boundary for the Amazon forest at 1.5 °C for global warming above pre-industrial levels, in concert with the Paris Agreement goals. ”

Line-by-line comments:

41: delete “address this issue by gathering state of the art knowledge about the various” and replace it with “review”

We now say: “Here we synthesize existing evidence ...”

43-35: Non-sequitur: The first part of the sentence is about the biological components of the ecosystem, but the second part of the sentence is about environment.

We understand the point. We now say: “For 65 million years, Amazonian forests remained relatively resilient to climatic variability. Now, the region is increasingly exposed to unprecedented stress from warming temperatures, extreme droughts, deforestation and fires, even in central and remote parts of the system.”

50: Find a better word than “systemic”. Systemic is not a gradation of local and regional. Consider “biome-wide”.

We replaced by ‘biome-wide’.

61: The first sentence of the Introduction can be removed without loss of information from the section.

We disagree. The first sentence introduces (very briefly) the complex nature of the Amazonian system, before entering its importance for people and what could happen in a tipping point scenario.

73 and thereafter: “indigenous” is an adjective, not a proper noun. Its arbitrary capitalization is a contrivance that is disrespectful to the actual (proper) tribes, cultures, and civilizations indigenous to regions around the world.

We follow a global convention for capitalizing Indigenous, recommended by the Chicago Manual of Style and the United Nations. We will follow Nature’s convention if it differs.

<https://www.un.org/dgacm/en/content/editorial-manual/updates>

78: Start new paragraph at “Understanding...”.

We started a new paragraph and rephrased a bit: “Understanding the risk of such catastrophic behaviour requires addressing complex factors that shape ecosystem resilience¹⁰. A major question is whether a large-scale collapse of the Amazon forest system could actually happen within the 21st Century, and if this would be associated with a particular tipping point.”.

88: Delete “constantly”.

Done.

90: Redundant. Delete “between (two or more) equilibrium states (also referred to as”.

Done.

109: Replace “are able to assess” with “assessed”.

We replaced by ‘we can assess’ because the purpose here is not to present a result, but to illustrate the approach used by different studies to assess alternative stable states and critical thresholds in the system.

126: The last sentence of this paragraph has been said previously. It can be deleted here.

Thank you. We deleted the sentence.

133: The second clause of the sentence, beginning “but the...” should be moved to the first sentence of the paragraph as an independent sentence.

Thank you for this suggestion. We now say: “The Amazon system has been mostly covered by forest throughout the Cenozoic (for 65 million years). Seven million years ago, the Amazon

river began to drain the massive wetlands that covered most of the western Amazon, allowing forests to expand over grasslands in that region.”

142: Change “on” to “by”.

We rephrased to “... driven by sediment deposition along ancient rivers.”

160: Remove the hyphen from “CO₂-fertilization”.

We deleted this sentence, and removed the hyphen from this term throughout the manuscript.

170: Remove “For instance” and (compared to pre-industrial conditions).

Done.

171: Delete the comma.

Done.

212: The first two sentences of this paragraph need not be said and should be deleted.

We agree with the reviewer and deleted both sentences.

217: Delete “together”, and replace “that result” with “resulting”.

We prefer our original sentence structure, highlighting both sides of the rainfall variability “Increasing rainfall variability across the Amazon is causing extreme drought events to become more widespread and frequent (Fig. 1c), together with extreme wet events and convective storms that result in more windthrow disturbances.”.

225: Delete “illegal activities, promoting”. This is about deforestation, not its legality.

We maintained our original sentence because it is crucial to consider the illegal aspect of human activities that cause deforestation and degradation.

233: Delete “capacity”. This is about carbon storage, not the capacity to do so.

Done.

239: Delete the parenthetical clause. This has been said many times already in the manuscript.

As requested, we deleted the text inside the parenthesis “(as the possibility of forest shifting into an alternative structural or compositional state)”.

245: Change “positive (destabilizing)” to “destabilizing positive”.

We are saying that disturbance synergisms could result in positive feedbacks, which can destabilize the ecosystem, not that they will destabilize positive feedbacks.

267: Change “buffer the effect of” to “reduce”.

Done.

270: Delete “rates (trees grow faster but also die earlier)”.

We added this part as a request from Ref #2. It can be useful for other readers.

279: Delete “even”.

Done.

283: Delete the comma and “flow”.

We maintained the comma because a short break is useful here, and also maintained the flow because we are talking about the flow of moisture downwind across the Amazon.

309: This section confuses species or biological diversity with environmental diversity or heterogeneity. Biodiversity is a mechanism of resilience, but environmental heterogeneity can either accelerate or decelerate ecosystem responses—depending again on the mix of species or functional types.

The reviewer correctly describes the roles of biodiversity and heterogeneity. In this section, we synthesize the role of heterogeneity, which is related both to environmental conditions and to forest composition. Heterogeneous environments shape heterogeneous tree communities. The outcome of this process will influence whether ecosystem transitions will become systemic or not. The roles of biodiversity are synthesized in another section.

310: Change “systemic” to “biome-wide” or equivalent. In this context, systems are not scale-specific.

We now rephrased to: “Ecosystem transitions may occur at the local scale, but they can also become systemic, resulting in large-scale transitions across the Amazon forest”. By systemic we mean that they spread across the system. We say ‘large-scale’ because transitions may become systemic but not necessarily spread across the entire biome.

313: Delete “, in particular,”. (Or at least delete the commas.)

We deleted ‘in particular’.

317: Delete “In theory”.

Done.

329: Forests do not have strategies. Change to “responses” or “mechanisms”.

We now say ‘seasonal forest trees’.

338: Replace “plateau” with “upland”?

By plateau we mean hilltop. In the Amazon, upland is often opposed to floodplain, and includes both plateaus and valleys.

348: This section gets the biodiversity heterogeneity distinction right. However, it feels wedged in and might not be worth its own section. Consider merging with another section(s).

The section indicated by line 348 is ‘Sources of connectivity’ that does not discuss biodiversity. Perhaps the reviewer is referring to the section ‘Biodiversity’. We created a Box 2 with the section ‘Ecosystem adaptability’ (including sub-sections ‘Biodiversity’ and ‘Indigenous peoples and local communities’), highlighting these mechanisms.

376: Move this section to later in the paper, between “Five drivers of water stress” and “Three alternative ecosystem trajectories”.

It is unclear why the reviewer requested this change. We consider that the section ‘Ecosystem adaptability’ follows the logic of explaining the mechanisms that affect forest resilience, hence it is in an appropriate place.

379: Delete “enhancing”.

Done.

420: Delete the first sentence of this paragraph and section.

Done.

423: Replace “global” with “atmospheric”.

We actually mean ‘global warming’.

425: Delete “, for instance”.

We also moved this sentence to Fig. 3 legend, but kept ‘for instance’ because we do not want to seem to say that these are the only tree death causes.

426: Replace “controls” with “limits”.

We changed to ‘affects’ because it is weird to say something ‘limits resilience’.

437: Delete “For instance”.

We deleted and connected both sentences: “Most CMIP6 models agree that a large-scale dieback of the Amazon is unlikely in response to global warming above pre-industrial levels, but this ecosystem response is based on certain assumptions, such as a large CO₂-fertilization effect.”

440: Delete “a few”.

Only a few models simulate a large-scale collapse of the Amazon among many that do not.

443: Delete “undoubtedly”.

Done.

467: Replace “Rainfall seasonal intensity” with “Seasonal rainfall intensity”.

We maintained as it was because we talk about rainfall seasonality intensity and use this term in several places, including Figure 3.

468 (and elsewhere): There is no such thing as satellite tree cover, nor of observations of satellite tree cover. Replace “Observations of satellite tree cover” with “satellite observations of tree cover” here and throughout the manuscript.

Done.

483: Replace “simulated” with “simulating”; delete “different”, “fire”, “events”, and “. The result”; insert a comma in “deforestation beyond”; delete “consequently” and “depending on the emissions scenario”.

We like our sentence and its elements, but tried to improve the flow and clarity. We now say: “A potential vegetation model ¹⁴⁸ found a critical threshold at 20 % of accumulated deforestation by simulating Amazon forest responses to different scenarios of accumulated deforestation (with associated fire events) and of greenhouse-gas emissions, and by considering a CO₂ fertilization effect of 25 % of the maximum photosynthetic assimilation rate. Beyond 20 % deforestation, forest mortality accelerated, causing large reductions in regional rainfall and consequently an ecosystem transition of 50 - 60 % of the Amazon, depending on the emissions scenario.”

491: Delete “disturbances”.

We refer to fire disturbances.

503: Delete “We identified”.

Done.

514: Delete the parentheses and “that are”.

We deleted the text inside the parenthesis and moved this sentence to the Fig. 4 legend.

516: Delete the comma after “feedbacks” and “to”.

Done.

518: “depending on the outcome of the novel interactions” is tautological. Delete the phrase.

We now say: “These alternative trajectories may be irreversible or transient depending on the strength of the novel interactions”.

527: Delete the parentheses.

Done.

561: Delete “(low tree cover)”.

Done.

569: Delete the comma after “state”.

Done.

584: Delete “We identified” and “that”. Make the sentence about the subject, not about the author.

Done.

586: Over-explaining. Delete “the” and “that could lead to a tipping point related to water stress”.

We simplified this sentence: “Several aspects of the Amazon forest system may help improve Earth System Models (ESMs) to more accurately simulate ecosystem dynamics and feedbacks with the climate system.”

488: Replace “need to” with “must” or “should”.

We deleted this sentence.

595: Delete “dynamics”.

We deleted this sentence.

598: Delete “the amount of”.

Done.

601: Delete “a “.

This ‘a’ is necessary.

630: Delete “hot-“.

The ‘hot-’ is another disturbance, together with extreme drought and fires.

634: Delete “forms of”.

Saying only 'due to new governance' sounds strange. We maintained 'new forms of governance'.

644: Replace the parentheses with commas in "(99% of which is illegal)".

Done.